



# Multi-variable parameter estimation for a global hydrological model: Comparison and evaluation of three ensemble-based calibration methods for the Mississippi River basin

Petra Döll[1,2], H.M. Mehedi Hasan[3], Kerstin Schulze[4], Helena Gerdener[4], Lara Börger[4],
Somayeh Shadkam[1], Sebastian Ackermann[1], Seyed-Mohammad Hosseini-Moghari[1], Hannes
Müller Schmied[1,2], Andreas Güntner[3,5], Jürgen Kusche[4]

[1]Institute of Physical Geography, Goethe University Frankfurt, Frankfurt am Main, Germany
[2]Senckenberg Leibniz Biodiversity and Climate Research Centre Frankfurt (SBiK-F), Frankfurt am
Main, Germany
[3]Helmholtz Centre Potsdam GFZ German Research Centre for Geosciences, Potsdam, Germany
[4]Institute of Geodesy and Geoinformation, University of Bonn, Bonn, Germany
[5]Institute of Environmental Science and Geography, University of Potsdam, Potsdam, Germany

*Correspondence to:* Petra Döll (p.doell@em.uni-frankfurt.de)

**Abstract.** Global hydrological models enhance our understanding of the Earth system and support the sustainable management of water, food and energy in a globalized world. They integrate process knowledge with a multitude of model input data (e.g., precipitation, land cover and soil properties and location and extent of surface water bodies) that describe the state of the Earth. However, they do not fully utilize observations of model output
variables (e.g., streamflow and water storage) to decrease model output uncertainty by, e.g., parameter estimation. For the pilot region Mississippi River basin, we assessed the suitability of three ensemble-based multi-variable calibration approaches for identifying both optimal and behavioral parameter sets for the global hydrological model WaterGAP, utilizing observations of streamflow (Q) and total water storage anomaly (TWSA). The common first steps in all approaches are 1) the definition of spatial units for which calibration parameters are
uniformly adjusted (CDA units), combined with the selection of observation data, 2) the identification of potential calibration parameters and their a-priori probability distributions and 3) sensitivity analyses to select the most influential model parameters per CDA unit that will be adjusted by calibration. In the estimation of model output uncertainty, we considered the uncertainties of the Q and TWSA observations. We found that the Pareto-optimal calibration (POC) approach, which utilizes the Borg multi-objective evolutionary search algorithm to find Pareto-
optimal parameter sets, is best suited for identifying a single "optimal" parameter set for each CDA unit. This parameter set leads to an improved fit to the monthly time series of both Q and TWSA as compared to the standard WaterGAP variant, which is only calibrated against mean annual Q, and can be used to compute the best estimate of WaterGAP output. The Generalized Likelihood Uncertainty Estimation (GLUE) approach is less suitable than POC to identify the optimal parameter set but enables the estimation of model output uncertainties that are due to
the equifinality of parameter sets and the observation uncertainty. The potential advantages of the ensemble Kalman filter calibration and data assimilation (EnCDA) approach, in which both parameter sets and water storages are updated, could not be realized, likely due to the high computational burden of this approach, This limited the EnCDA ensemble size to 32, while 20,000 ensemble members could be evaluated in the case of POC


and GLUE. Partitioning the whole Mississippi River basin into five CDA units (sub-basins) instead of only one improved model performance during the calibration and validation periods. Very diverse parameter sets were found to lead to similarly good fits to observations, but the range of values of three parameters could be narrowed by calibration. Model structure uncertainties, in particular regarding the operation of man-made reservoirs, the location and extent of small wetlands, and the (lacking) representation of losing river conditions in WaterGAP, are suspected to be the main reasons for the low coverage of the observation uncertainty bands by the GLUE-

derived model output uncertainty bands. Model structure uncertainties are also the likely reason for major trade-offs between optimal fit to Q and TWSA. Calibration against GRACE TWSA only, in regions without Q observations, may worsen the Q simulation as compared to the uncalibrated model variant. We plan to add additional remotely-sensed observations in the multi-variable calibration of WaterGAP and suggest considering parameter uncertainty in multi-model ensemble studies of the global freshwater system.

**1 Introduction**

By quantifying water flows and storages on the Earth's continents, global hydrological models (GHMs) contribute to our understanding of the functioning of the Earth system. GHMs are indispensable for assessing past and future impacts of human activities on the global freshwater system in the Anthropocene, including water abstractions, dam construction and greenhouse gas emissions. In our globalized world, where local decisions affect freshwater

systems worldwide, GHMs support sustainable water use by enabling the globally consistent computation of indicators of water availability and water stress. These indicators can inform the decisions of private or corporate water consumers (e.g., by water footprints and life cycle analysis) and water managers (e.g., by drought monitoring of climate change risk assessments).

GHMs integrate a large amount of spatially distributed physiographic data including data on soils and the

deeper subsurface, land cover, surface water bodies and human water use. Like all hydrological models, GHMs (including land surface models) suffer from uncertainty due to uncertain model structure, model input (in particular climate forcing) and model parameters (Döll et al., 2016). To reduce uncertainty, models are calibrated by adjusting the model in a way that simulated values of a model output variable optimally match observations of this variable. In basin-scale hydrological modeling, the estimation of model parameters by calibration against

streamflow time series is standard. This is not the case for GHMs, which is not only due to the limited availability of suitable observation data with global coverage and the large effort required to exploit them but also to a lack of methodological knowledge about how to best use observations for GHM adjustment.

So how can observations of model output variables be best used for estimating parameters of a GHM, given model complexity, the large number of model parameters, the large spatial extent and heterogeneity of the model

domain, the interdependence of the calibration of sub-basins due to their connection by lateral streamflow as well as the problem of equifinality? And how can such observations be best used for determining not only the best estimate but also the uncertainty of GHM model output?

Equifinality or its synonym non-uniqueness means that different combinations of model input, structure and parameters may lead to a similarly good agreement between simulated and observed values of a model output

variable so that it is not possible to determine an optimal (unique) combination (Beven, 1993). It is related to



epistemic uncertainty about model structure, input and parameters (Beven and Smith, 2015), which can be expected to be larger in global-scale modeling than in basin-scale modeling. Equifinality is exacerbated by observation errors, which are difficult to quantify comprehensively, and include non-random errors and biases (e.g., see Di Baldassarre and Montanari, 2009, regarding errors of in-situ streamflow observations derived via the commonly applied rating curve method). In addition, comparisons between simulated and observed values often suffer from incommensurability due to different spatial (e.g., groundwater head from groundwater wells, Reinecke et al., 2020) or temporal (e.g., instantaneous water table elevation of surface water bodies from radar altimetry, Berry et al., 2005) scales. Equifinality implies that multiple model simulations, generated by, e.g., running the model with multiple parameter sets, are acceptable and informative for the model user if they 1) cannot be easily rejected as infeasible representations of the system given the level of the diverse uncertainties (in particular regarding climate forcing, model structure and observations) and 2) support the specific modeling purpose, e.g., to project either low flows or floods (Beven and Smith, 2015). The ensemble of such model runs or parameter sets is referred to as "behavioral" (Beven and Binley, 1992).

Equifinality increases with the number of parameters to be estimated. The identifiability of parameters can be increased by utilizing more than one calibration objective, e.g., minimizing both the root mean square error of 1) all streamflow observations and 2) the low flow observations. It has been suggested from experience that no more than 5-6 parameters can be estimated for each calibration objective (Efstratiadis and Koutsoyiannis, 2010). While the equifinality problem can be reduced by utilizing various streamflow signatures (Gupta et al., 1998; Arheimer et al., 2020) for parameter estimation, it is preferable to utilize, in addition to streamflow observations, observations of one or more additional model output variables (multi-variable calibration, Yassin et al., 2017; Stisen et al., 2018; Dembélé et al., 2020). The additional observation variable of choice for GHM parameter estimation is total water storage anomaly (TWSA) over the continents from GRACE satellites, as they provide spatially uninterrupted global coverage and almost uninterrupted monthly time series since 2003 (some missing months before 2016 and a gap until the start of GRACE-Follow-on mission in May 2018). TWSA observations integrate over all water storage compartments on the continents (glacier, snow, soil, groundwater and surface water bodies) and thus also depend on all water flows on the continents. This is similar to streamflow (Q), which is the integrative result of upstream flow and storage processes. Thus, TWSA observations complement Q observations. The coarse spatial resolution of TWSA observations of about 100,000 km$^2$ is less problematic for GHMs than for basin-scale hydrological models.

Currently, most GHMs do not use observed Q (or any other observations) to estimate parameters in the upstream basin (Bierkens, 2015). One exception is the GHM WaterGAP (Alcamo et al., 2003; Döll et al., 2003), which is calibrated in a simple manner by adjusting one to three parameters in each of 1319 large drainage basins (Müller Schmied et al., 2014, 2021) such that simulated long-term average annual Q is close to observations. For the standard version of WaterGAP, adjustment of a larger set of model parameters is currently not done due to the equifinality problem and computational simplicity. While this limited calibration leads to a reduction of the Q bias and thus more realistic estimates of renewable water resources as compared to the uncalibrated version (and the results of other GHMs that are not calibrated in a basin-specific manner), it does not significantly improve simulated seasonality and interannual variability of Q (Hunger and Döll, 2008). Discrepancies with time series of observed monthly Q (Müller Schmied et al., 2014) or TWSA (Döll et al., 2014; Scanlon et al., 2019) can be high





even after the standard WaterGAP calibration. It is therefore desirable to adjust parameters that affect the seasonality of simulated Q or TWSA as well as their interannual variability and potential trends. Multi-variable parameter estimation based on observations of both Q and TWSA may enable the adjustment of such parameters. In the following, "calibration" is used synonymously with "parameter estimation".

Multi-variable parameter estimation can be achieved by various approaches such as 1) Pareto-optimal calibration using an optimization algorithm (POC) (Werth and Güntner, 2010), 2) the Generalized Likelihood Uncertainty Estimation (GLUE) approach for identifying behavioral parameter sets (Beven and Binley, 1992) and 3) data assimilation with the ensemble Kalman filter in which model states and parameters are jointly updated (Eicker et al., 2014), hereafter called EnCDA. With each of these approaches, an ensemble of optimized parameter sets is generated. Werth and Güntner (2010) developed a multi-variable POC scheme for WaterGAP and applied it to adjust six to eight parameters homogeneously in each of 28 large river basins (e.g., Amazon, Mississippi and Lena), using both Q and TWSA observations. A similar approach was applied by Xie et al. (2012) to calibrate the SWAT model for 10 large basins in Sub-Sahara Africa, using observed TWSA time series and monthly mean Q values. The GLUE approach has not yet been applied with WaterGAP or other GHMs. First EnCDA efforts of assimilating GRACE TWSA into WaterGAP for the Mississippi River basin in the US and the Murray-Darling basin in Australia were made by Eicker et al. (2014) and Schumacher et al. (2016a, b, 2018). While EnCDA with more than one observation variable (Q and remote-sensed soil moisture) has already been done in large-scale hydrological modeling of the Upper Danube basin (Wanders et al., 2014), joint EnCDA of Q and TWSA has not yet been reported.

The objective of this paper is to assess the suitability of the three multi-variable calibration approaches POC, GLUE and EnCDA for identifying ensembles of optimal and behavioral parameter sets of the GHM WaterGAP by model calibration against observations of Q and TWSA, taking into account observation uncertainties. In addition, an approach for taking into account the observations errors for the definition of performance thresholds for behavioral parameter sets is presented. In each calibration approach, model parameters of all WaterGAP grid cells within so-called calibration-data assimilation (CDA) units were uniformly adjusted. Based on calibration exercises either for the whole Mississippi River basin (MRB) as one CDA unit or for its five sub-basins (four upstream basins and one downstream basin) as alternative CDA units, we will answer the following research questions:

1. What are the strength and weaknesses of each of the three multi-variable calibration approaches?

2. What is the added value of the multi-variable calibration as compared to the standard WaterGAP calibration for identifying one "optimal" parameter set?

3. How and how well can WaterGAP model output uncertainty be quantified?

4. How large are the trade-offs between the optimal simulation of Q and TWSA? As TWSA observations are available with a global coverage while Q observations are not: To what extent is Q simulation improved by calibration against TWSA only?

5. What is the added value of individually calibrating sub-basin CDAs instead of one basin CDA?

6. What are the characteristics of the identified optimal and behavioral parameter sets? How large is the equifinality of parameter sets? Can optimal values be identified for some parameters?


The paper is structured as follows. Section 2 describes and compares the three calibration approaches. Section 3 provides a short description of the GHM WaterGAP and explains the setup of the calibration study, including the calibration parameter selection by an initial sensitivity analysis. In Section 4, we present the results of our calibration study, and in Section 5, results are discussed and conclusions are drawn.

## 2 Approaches for multivariable calibration of global hydrological models

While model calibration can encompass adjustments of model structure, initial conditions, input variables and parameters, model calibration in hydrology focuses on the identification of optimal or suitable parameter sets. The focus on parameter adjustment in hydrological modeling is justified by the necessity of using many parameters that cannot be measured independently or derived from first principles. Water flows in the hydrology domain are largely dominated by the local geometry and local boundary resistances of the individual flow pathways, different from the water flows in the meteorology and oceanography domain (Beven, 2002). In hydrological models, water flows are expressed as a function of water storage or potential gradients as well as parameters that represent the highly uncertain average effects of local geometries and boundary resistances. In comprehensive hydrological models that distinguish various compartments, about 10-50 model parameters result per spatial unit. In the case of distributed models in which spatial heterogeneity of land and water is accounted for by distinguishing a large number of spatial units such as sub-basins or grid cells, each computational unit is described by its parameters set, leading to a very large number of model parameters. GHMs covering the whole land area of the globe typically represent spatial heterogeneity on the continents by distinguishing more than 60,000 0.5° grid cells, with more than 1 million model parameters whose values need to be set to enable computation.

In the GLUE approach, an ensemble of behavioral parameter sets is derived, each of which leads to an acceptable model performance given uncertainties and model purpose; the ensemble is in most studies defined by model simulations exceeding certain performance thresholds. In the POC approach, an ensemble of Pareto-optimal parameter sets is generated that does not take into account model or observation uncertainties but the trade-off that occurs between the fit to various performance metrics. A parameter set is called Pareto-optimal or non-dominated if it results in a better simulation performance than any other Pareto parameter set for at least one of the objectives; none of the objective functions can be further improved without degradation of some of the other objective functions (Khu and Madsen, 2005; Werth and Güntner, 2010). In EnCDA, an initial ensemble of parameter sets is updated at each intake of observations, and parameters ideally converge with increasing intake of observations; there is a single objective function, in which multiple objectives are implicitly weighted by considering model and observation uncertainties.

It is computationally challenging to work with an ensemble of parameter sets, e.g., in the context of climate impact studies or seasonal forecasting. Therefore, we also identified (pseudo-)optimal parameter sets for each CDA unit. In this section, the three multi-variable calibration approaches POC, GLUE and EnCDA are first described and then compared to each other.



### 2.1 POC

POC aims at identifying Pareto-optimal parameter sets. While the ensemble of Pareto-optimal parameter sets
determined by POC is optimal only under the assumption that there are no observation, input and model structure
uncertainties, they take into account that there is rarely a parameter set that leads to a simulation of different output
variable that is equally optimal with respect to all observational variables. POC as applied in this study implements
an optimization algorithm such as the Borg multi-objective evolutionary search algorithm (Hadka and Reed, 2013).
Based on an initial small ensemble of parameter sets derived from a-priori parameter distributions, the parameter
sets are updated according to the value of the objective functions (performance metrics) to achieve improved
performance. Then, the model is re-run; based on the new values of the objective function, parameter sets are
updated again in an iterative fashion for a pre-selected number of iterations and thus model runs to identify Pareto-
optimal parameter sets. Due to model, input and observation errors, it is unlikely that any parameter set will lead
to the highest values of all objective functions. Without additional subjective preference information on what
objective function is most important, all Pareto-optimal parameter sets are considered to be equally good. From
the often large number of Pareto-optimal parameter sets, a "preferred" set can be selected using a variety of
approaches (Khu and Madsen, 2005). The so-called "compromise parameter set" leads to values of the applied
objective functions OF (or performance metrics) such that the overall performance deficit $D_p$ regarding all OF is
minimized (Yu, 1973). $D_p$ is the distance between the utopia point, where all OF values are at their optimal values
OF*, and the OF values of the Pareto-optimal parameters sets x. According to Yu (1973),

$$D_p\big(OF(x)\big) = \left[\sum_{i=1}^{n}\big(OF_i^* - OF_i(x)\big)^p\right]^{1/p} \tag{1}$$

where n is the number of objective functions and p is a parameter that is larger or equal to 1 and needs to be
selected. By minimizing $D_p$ with p=2, the Euclidean distance is selected to determine the compromise parameter
set.

Applying a POC approach, Werth and Güntner (2010) used monthly time series of in-situ observed Q and
GRACE for 28 large river basins to adjust WaterGAP parameters individually for each basin, after first
determining the most influential basin-specific parameters. Calibration parameters included multipliers of cell-
specific parameters such as rooting depth as well as parameters that were assigned to each cell in the basin such
as a groundwater outflow coefficient. Werth and Güntner (2010) found that improved simulations of TWSA and
Q were achieved for most basins after calibration, but calibrated Q was still poor compared to the observed values
in some basins; a better fit to GRACE TWSA did not necessarily lead to a better fit of simulated to observed Q.
The disadvantage of the POC approach is that it is computationally much more expensive than the simple
calibration approach for standard WaterGAP such that PCO was only performed for 28 instead of 1319 CDA units.
Thus, spatial variability of calibration parameters within the large basins could not be taken into account by Werth
and Güntner (2010), and differences in model performance after calibration by either POC or the standard
calibration approach were not analyzed.


### 2.2 GLUE

In the GLUE approach, a large number of different model parameter sets is generated first, based on assumed a-
priori distributions of parameter values. In the next step, a subset of so-called behavioral parameter sets is identified
from this initial set. This is done by running the model alternatively with each parameter set and then computing
the values of a model performance metric using observations of model output variables, which is called likelihood
measure in GLUE (Beven and Binley, 2014). In the next step, a threshold for the performance metric is identified
below which model performance is so low that these parameter sets are considered to have a likelihood of zero.
Likelihood measures and thresholds for behavioral parameter sets are subjectively selected also based on the
expertise of the modeler and should take into account the uncertainty of model structure, climate forcing and
observations as well as the specific modeling purpose.

Multiple observation variables can be combined for determining behavioral parameter sets, by selecting the
subset of parameter sets for which all performance metrics are better than their different thresholds. The selection
of the metric-specific thresholds implies a type of weighting between fits to the different variables. As a subset of
all behavioral parameter sets, Pareto-optimal parameter sets can be identified; the pseudo-optimal parameter set
can be determined using Eq. 1. Furthermore, the likelihood of each behavioral parameter set can be derived from
the performance metric such that a probability distribution of model output can be quantified.

### 2.3 EnCDA

In the EnCDA approach, parameter sets of each CDA unit are optimized together with water storages in the various
storage compartments and grid cells by data assimilation with the ensemble Kalman filter (EnKF; Evenson, 1994),
by including both the water storages and the parameters in the state vector. The basic idea of data assimilation with
the Kalman filter approach, as done in EnCDA, is to optimally combine observations with simulation results at the
time of the observations according to estimates of model and observation errors (Clark et al., 2008). In EnCDA,
an ensemble of model runs with different parameter sets and perturbed climate inputs serves to estimate the model
error. The higher the ratio of model error to observation error, the more weight is given to the observations and
the larger is the adjustment of water storages and model parameters. Water volumes and parameters, all of which
are state variables, are updated in each ensemble member whenever observations are available (e.g., once per
month). State update depends on the information contained in the covariance matrices of simulated states (water
storages and parameters), simulated Q and observations. Covariance matrices of states and simulated Q are derived
from differences between the estimates of each ensemble member and the ensemble mean. The ensemble mean of
all updated water storages and Q is assumed to be the best estimator (Evensen, 2003). In the case of models with
many grid cells and various storage compartments (10 in WaterGAP), the number of updated states strongly
exceeds the number of observations.

EnCDA has a high potential for improving parameter estimation as the stepwise updates of water storage in
the diverse storage compartments can help to compensate for model structure and input uncertainties, e.g., for
underestimation of the precipitation input by adding mass/water to the system during the update. EnCDA was
applied in some studies (e.g., DeChant and Moradkhani, 2012; Wanders et al., 2014), also for assimilating GRACE
TWSA into WaterGAP (Eicker et al., 2014; Schumacher et al.,2016a,b, 2018) for the Mississippi River basin in





the US and the Murray-Darling basin in Australia. However, EnCDA for GHMs using GRACE TWSA is still in an exploratory phase. Updating parameters in addition to updating water storages might increase the chance of spurious Q simulation due to the highly non-linear relations between Q and storages as well as parameters (De Chant and Moradkhani, 2012; Xie and Zhang, 2013; Schumacher et al., 2018). To achieve plausible and stable EnCDA results regarding parameters and model output variables in complex distributed hydrological models in

which the number of states exceeds by far the number of observations, the degrees of freedom may have to be reduced and rapid changes in parameters from one time step to the next need to be avoided (Xie and Zhang, 2013). Schumacher et al. (2018) found that EnCDA with only TWSA observations is limited in constraining individual model parameters even if the number of calibration parameters is very small as the calibration/data assimilation system is highly underdetermined. This is why adding Q observations is promising.

The output of EnCDA regarding parameters can be viewed as a time series of recursive estimates for the parameter sets for each ensemble member, even if these parameters are modeled as stationary in time (as in this study). The parameter sets of each ensemble member at the end of the calibration/data assimilation (CDA) period can then be used to generate ensemble predictions. The studies of Wanders et al. (2014) and Eicker et al. (2014), which did not utilize Q observations, showed that with this approach, the ensemble means of model output values

during the validation period fit better to observations of Q and TWSA than uncalibrated model output.

### 2.4 Comparison of the three calibration approaches

POC, GLUE and EnCDA approaches share some characteristics and differ in others (Table 1). All three start with a large number of parameter sets that are derived from a-priori assumptions on the probability distribution of calibration parameters and generate an ensemble of optimized parameter sets. EnCDA differs from POC and

GLUE by simultaneously modifying model parameters and model states. EnCDA and GLUE are regarded as Bayesian approaches as they aim at deriving probability distributions of parameter sets and thus model output. In POC, the ensemble of Pareto-optimal parameter sets represents the uncertainty that is caused by the fact that due to model structure and input uncertainty, different parameter sets lead to optimal performance for different calibration objectives. Information from observations is used in all three approaches to update an a-priori belief

about the probability distribution of parameters. However, parameter set selection is done in very different ways and based on different assumptions. Both POC and GLUE compare the model output over the complete calibration period with all observations to determine performance metrics. While the evolutionary search algorithm of POC starts with a small number of parameters sets, runs the model, and then generates new parameter sets with ever-improved performance metrics, in GLUE the large initial ensemble generated from a-priori parameter distributions

is evaluated regarding performance metrics and the behavioral members among the initial ensemble are identified. In POC and GLUE, parameters are temporally constant. In EnCDA, an ensemble of model runs is performed in a stepwise fashion from the time of one observation to the time of the next. EnCDA updates the parameters sequentially (in our study each month) such that time series of recursive parameter estimates are computed. It is assumed that updates are informed by an ever-increasing amount of information from observations so that the

parameter sets after the last update, i.e., at the end of the calibration period, are the best estimate. However, this can be disputed. A study on EnCDA using GRACE TSWA for the Australian Murray-Darling basin showed that





parameter values vary in time with changes in climatic conditions within the river basin, probably due to an inappropriate model structure that does not allow the correct translation of precipitation variability into model output variability (Schumacher et al., 2018). The capability to reveal such dynamics may be advantageous for
improving our understanding of model deficiencies. It needs to be investigated whether and how EnCDA can be used to determine optimal parameter sets that are suitable for model runs without adjustment of states.

In EnCDA, quantified errors of both the model and the observations are required to update water storages and parameters in each of the ensemble members (Table 1). The ensemble serves to estimate the model error, which includes parameter and climate forcing uncertainty and is calculated as the variance of the differences between
each ensemble member and the ensemble mean. The EnKF applied in EnCDA represents an optimal and unbiased estimator only under the assumption that errors are Gaussian, unbiased and well-known, neither of which is the case (Wang et al., 2020; Moradkhani et al., 2005; Beven and Binley, 1992). In GLUE, the model error due to parameter uncertainty (but not due to climate forcing uncertainty) is indirectly taken into account as the a-priori ensemble depends on assumptions of parameter distribution, similar to POC. Observation errors may be considered
quantitatively but in most applications they are not (Beven and Binley, 2014). In Section 3.4.2, we describe a way for taking into account the observation uncertainties in GLUE. Werth and Güntner (2010) suggested a way to include observation errors in POC. First, they determined an error ellipse around the compromise solution (defined in Eq. 1) by first generating an ensemble of observations from perturbing the observation time series with the observation errors and then determining the range of performance values of the compromise solution for this
ensemble of perturbed observations. By considering all the non-dominated and dominated parameter sets inside the error ellipse, they identified an ensemble of likely parameter sets that was informed by both observations and observation uncertainty. In this case, POC can, like EnCDA and GLUE, be used to estimate uncertainties of parameter sets and model outputs. Nevertheless, it should be noted that this approach does not incorporate observational uncertainty directly into multi-objective parameter calibration in a rigorous way. Therefore, we did
not take this approach in our study.



**Table 1.** Comparison of the main characteristics of the calibration approaches POC, GLUE and EnCDA as applied in this study.

| | POC<br>Pareto-optimal calibration | GLUE<br>Generalized likelihood uncertainty estimation | EnCDA<br>Ensemble Kalman filter calibration and data assimilation |
|---|---|---|---|
| Use of a-priori parameter ensembles | Yes | Yes | Yes |
| Direct modification of water storages | No | No | Yes |
| Bayesian approach | No | Yes | Yes |
| Estimation of model output uncertainty | Uncertainty only due to multiple objectives | Yes | Yes |
| Selection of parameter sets | Once, based on all observations | Once, based on all observations at once | Recursive, parameter sets updated at each observation time step |
| Quantitative information on parameter uncertainties considered | Indirectly via an a-priori range of parameter values | Indirectly via an a-priori ensemble of parameter sets | Directly as a factor of model uncertainty |
| Quantitative information on climate forcing uncertainties considered | No | No | Yes, as a factor of model uncertainty |
| Quantitative information on observation uncertainties considered | Possible in post-processing, by limiting Pareto-optimal parameter sets to thresholds selected using GLUE ensemble | Possible, by selecting thresholds for behavioral solutions according to observation uncertainties | Yes |
| Rigorous consideration of uncertainty | No | No | Partly |
| Various objective functions including signatures can be selected | Yes | Yes | No |
| Weighting between different objective functions | Subjective weighting to identify a parameter set that is optimal in a specific context | Subjective weighting to identify parameter set(s) that is (are) optimal in a specific context | Implicit weighting based on model and observation uncertainties |
| Determination of Pareto-optimal parameter sets under the assumption that there is only parameter uncertainty | Yes, determined by search algorithm | Yes, selected from a-priori ensemble | No (due to the small ensemble size) |
| Complexity | Medium | Low-medium | High |
| Computational effort for a specific objective function | Medium | Medium | Very high |
| Computational effort for analyzing alternative objective functions | High | Medium | Not applicable |
| Risk of spurious model behavior | Low | Low | High due to modifying water volume in multiple storage compartments |

Different from EnCDA with its rigorous handling of uncertainties, GLUE is an informal Bayesian approach that is much simpler than EnCDA (Table 1). Likelihood is here understood in a very general sense, as a fuzzy measure of belief of how well the model conforms to the observed behavior of the system, and not in the sense of maximum likelihood theory which is the basis of EnCDA (Beven and Binley, 1992). In EnCDA, the likelihood of a parameter set is a product of model errors, observation errors and the differences between observed and simulated
variables (and other factors) (Section 3.2 in Schumacher, 2016). The informal and subjective treatment of



uncertainty in the GLUE approach has caused controversy because the different error sources are not distinguished (Vrugt et al., 2008). This can mean that non-maximum likelihood solutions might be accepted as parameter estimates. However, the GLUE approach can be defended against formal Bayesian methods as these require a-priori knowledge about errors that is lacking in most hydrological modeling applications (Beven and Binley, 2014).

In addition, formal Bayesian methods (e.g., DREAM) are difficult to implement and much less computationally efficient but may lead to similar outcomes (Vrugt et al., 2008). In GLUE, the likelihood measure can be freely chosen by the modeler. She could choose a formal likelihood measure like the one applied for EnCDA, a measure that relates the deviation of model output from observations to the observation error or just any model performance metric for comparing observations to simulations (Beven and Binley, 2014, their Table 3). Given the large

epistemic uncertainty about hydrological systems, GLUE relies on the subjective expertise of the modeler to define a suitable likelihood measure given her often only qualitative knowledge about uncertainties of model structure, model input, model parameters and observations. There is a multitude of likelihood measures that can be used to identify parameter sets that fit better to observations than the a-priori ensemble (or the standard deterministic parameter set) and are therefore more likely than others. A likelihood of zero is assigned to all parameter sets that

are not "behavioral", i.e., if the likelihood measure is below a threshold that is set subjectively by the modeler. For the example of the popular likelihood measure Nash-Sutcliffe efficiency (NSE), behavioral parameter sets may be defined as those that result in an NSE larger than 0.7 if the behavior of the hydrological system can be easily simulated; if not, the threshold will have to be lowered to get any behavioral parameter sets. To obtain the a-posteriori probability distribution of parameter sets, only the behavioral parameter sets are considered and their

probability is derived from the NSE obtained with them.

Objective functions (= likelihood measures = performance metrics) can be freely chosen in the case of POC and GLUE. This allows the selection of diverse hydrological signatures of the observables, e.g., those that focus on high or low flows in the case of streamflow. EnCDA minimizes the root mean squared error, and it is very difficult to apply another objective function (Table 1). In addition, the likelihood function in EnCDA considers

only the deviations between the model output and observations at one point in time as the ensemble Kalman filter and not the ensemble Kalman smoother was applied in this study. In contrast, performance measures used in POC and GLUE evaluate model performance (and calibrate model parameters) over the whole calibration period. EnCDA differs from POC and GLUE in that weighting between the performance metrics for the multiple objectives/variables is implicitly done given the model and observation errors (Table 1). In POC and GLUE,

subjective weighting needs to be done for selecting one "optimal" parameter set. POC and GLUE also have in common that they can serve to identify Pareto-optimal parameter sets or one compromise parameter set that can then be used to quantify in a computationally efficient way in, e.g., climate change studies or seasonal forecasting, where hydrological models are driven by an ensemble of climate data sets.

The complexity of the three calibration approaches differs (Table 1). The computational burden is much higher

for EnCDA than for POC and GLUE. Therefore, only a very small number of ensemble members can be used in the analysis; ensemble sizes typically are between 30 and 100. These low-rank ensembles may fail to correctly convey the covariance information between model states and parameters or between different parameters. Localization techniques can be applied to mitigate this effect but with the trade-off that long-distance covariance information is neglected or down-weighted. For the same number of evaluated parameter sets, the computational



effort of POC and GLUE is approximately the same, for evaluating a specific objective function. However, as the parameter ensemble generated by the search algorithm in POC depends on the objective function (unlike in the case of GLUE), the computational burden of POC becomes, for example, twice as high as that of GLUE if one alternative objective function is taken into account. Finally, EnCDA is prone to spurious results as the modification of water storages to improve the fit to TWSA observations might lead to little-constrained changes in individual

storages, with impacts on simulated water flows. In the EnCDA study of Schumacher et al. (2018), river storage was adjusted in WaterGAP based on TWSA observations, leading to spurious increases in Q not seen in WaterGAP runs without water storage updates or in the observations.

### 3 Methods and data

#### 3.1 The global water resources and use model WaterGAP

In this study, we applied WaterGAP 2.2d, which is comprehensively described in Müller Schmied et al. (2021). With a spatial resolution of 0.5° latitude by 0.5° longitude (55 km by 55 km at the equator), WaterGAP computes both water resources, i.e., water flows and storages, and human water use on all land areas of the globe except Antarctica. Water withdrawals and consumptive water use in the sectors households, manufacturing, cooling of thermal power plants, livestock, and irrigation are computed by five water use models. From the output of the

water use models, the linking model GWSWUSE computes potential net water abstractions from groundwater (NAg) and surface water (NAs) as the difference between all withdrawals from and all return flows to groundwater and surface water, respectively. Time series of monthly NAg and NAs are inputs of the WaterGAP Global Hydrology Model (WGHM), together with time series of daily climate variables (Müller Schmied et al., 2021). WGHM computes various water flows (e.g., evapotranspiration, groundwater recharge and Q) as well as water

storage variations in ten compartments: canopy, snow, soil, groundwater and the surface water bodies local and global wetlands, local and global lakes, global man-made reservoirs and rivers (boxes in Fig. 1). The term "local" means that the surface water bodies are fed only by the runoff produced in the same 0.5° cell, while "global" wetlands, lakes and reservoirs are also fed by inflows from the upstream cells. The runoff generated in a cell from the "vertical" water balance (Fig. 1) is transported through the groundwater and, if existing, through the various

types of surface water bodies before reaching the river. Outflow from the river compartment is Q. Glaciers are not simulated in this WaterGAP version; while there are some glaciers in the most upstream parts of the Arkansas and Missouri river basins, these are not expected to strongly impact mean TWSA of the large CDA units or streamflow at the outlet of the CDA units (Fig. 2). To calculate TWSA time series, the sum of all ten compartmental water storages is computed and normalized by its mean value over a reference period.





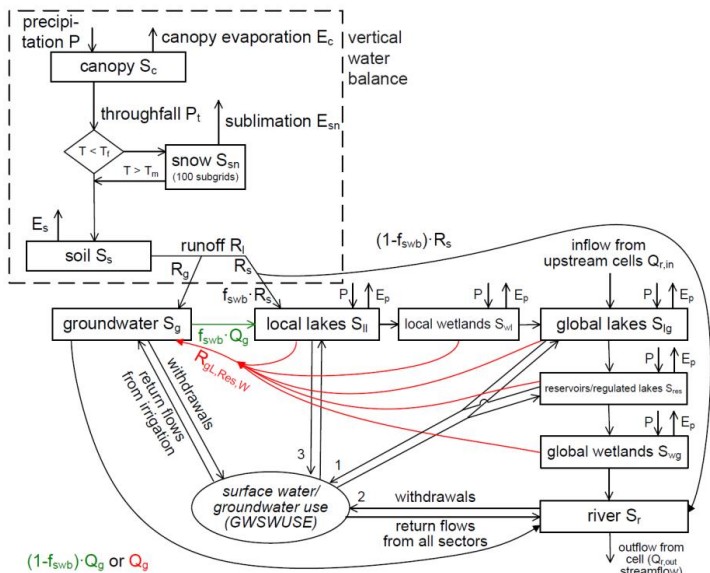


**Figure 1.** Schematic of WGHM in WaterGAP2.2d. For each 0.5° grid cell, daily water balances of a maximum of ten water storage compartments (boxes) are computed from their respective inflows and outflows (arrows) (Fig. 2 of Müller Schmied et al. (2021)). Green and red colors indicate processes that occur only in grid cells with humid and (semi)arid climate, respectively. $E_s$: soil evapotranspiration, $E_p$: potential evapotranspiration, $R_g$: groundwater recharge from soil, $R_s$: fast surface runoff and subsurface runoff, $R_{gL,Res,w}$: groundwater recharge from surface water bodies, $Q_g$: groundwater discharge to surface water bodies and the river, $F_{swb}$: area fraction of surface water bodies. Net groundwater abstracts are taken from the groundwater storage compartment, while net surface water abstractions are taken from global lakes or reservoirs in the cell (priority 1), the river (priority 2) or local lakes (priority 3).

In the ordinary differential equations describing the dynamics of the individual water storage compartment, outflows are parameterized as a function of compartmental water storage (Müller Schmied et al., 2021). Other important model parameters determine the maximum values of compartmental water storage, such as the maximum soil water storage in the effective rooting zone (soil compartment) or active lake depth, which defines the maximum height of the water table of local and global lakes above the outflow level. Parameters affecting potential evapotranspiration govern the simulated atmospheric demand for water. Temperature–related parameters are important for snow processes.

As a standard, WGHM is calibrated against observed mean annual Q by adjusting one model parameter, the runoff coefficient, and if necessary, two correction factors (Müller Schmied et al., 2021). In the equation that describes the soil water dynamics, the runoff coefficient determines, together with soil water saturation, the amount of runoff from the land $R_L$; it varies between 0.1 and 5. The larger the runoff coefficient, the smaller the runoff





becomes. If the adjustment of the runoff coefficient is not sufficient for not exceeding a maximum discrepancy
between simulated and observed mean annual Q of 10%, a multiplicative areal correction factor for runoff from

land is introduced that also corrects evapotranspiration (range of 0.5 to 1.5). If this is still not sufficient to match
observed Q within 10%, the Q in the grid cell where the gauging station is located is multiplied by a station
correction factor. This violates the mass balance but is done to avoid error propagation to the downstream basins.
In the standard WaterGAP, the calibration period was 1980-2009 if stream data are available for the station
otherwise the most recent earlier period. The runoff coefficient in basins without Q observations is determined by

a regression approach, where calibrated runoff coefficients are related to various characteristics of the drainage
basins (Müller Schmied et al., 2021). With this calibration and regionalization approach, a median Nash-Sutcliffe
efficiency of 0.52 and a median Kling-Gupta efficiency of 0.61 is achieved for the fit of the time series of monthly
Q at the 1319 calibration stations. The median correlation coefficient of 0.79 indicates an often poor simulation of
the timing of monthly Q both seasonally and inter-annually. WaterGAP 2.2d tends to underestimate the variability

of monthly Q in northern snow-dominated river basins (Müller Schmied et al., 2021). It underestimates the mean
annual TWSA amplitude in 66% of the 143 investigated river basins by more than 10%. TWSA trends, in particular
positive trends, are often underestimated (Müller Schmied et al., 2021; Scanlon et al., 2018).

### 3.2 Calibration setup for the Mississippi River basin

#### 3.2.1 Study period and CDA units

Due to TWSA and climate input data availability, the study period was limited to January 2003 to December 2016.
The study area excludes the most downstream part of the Mississippi River basin (MRB) due to a lack of Q
observations. The Q gauging station at Vicksburg in the lower MRB is the most downstream station with a long-
term record (Fig. 2). Hereafter, we refer to the upstream area of Vicksburg as the whole MRB. We study two
variants of the spatial configuration of CDA units, in which calibrated parameters were uniformly adjusted. Either

the whole MRB is treated as one CDA unit, or the MRB is subdivided into five CDA units. In the latter variant,
four of the five CDA units (Arkansas River basin, Missouri River basin, Upper MRB and Ohio River basin) are
upstream river basins that are defined as the drainage basin of four gauging stations for which data for the study
period 2003-2016 are available (Fig. 2). The fifth CDA unit is the Lower MRB, which receives inflow from the
four upstream CDA units. We divided our study period into a calibration period for parameter estimation from

2003 to 2012 and a validation period, in which the model is run with the estimated parameters, from 2013 to 2016.
Q is additionally validated at six gauging stations that were not used for calibration (Fig. 2).



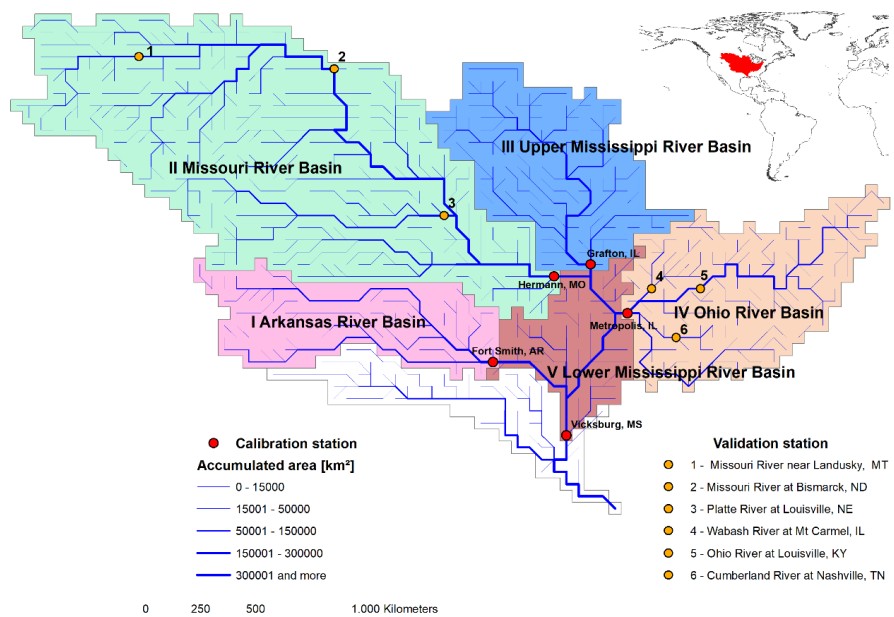

**Figure 2.** The Mississippi River basin as represented by the 0.5°x0.5° grid cells in WaterGAP, with delineation of
the five CDA units. The CDA units were defined as the upstream cells of the five indicated calibration stations
(streamflow gauging stations, shown in red). The stream network implemented in WaterGAP is shown, indicating
the upstream areas of each grid cell by the line width. In addition, the locations of the six streamflow validation
stations are plotted, shown in orange.

### 3.2.2 Observation data

Q data were obtained from the Global Runoff Data Centre (https://www.bafg.de/GRDC/) and the US Geological
Survey (https://maps.waterdata.usgs.gov/mapper/). For monthly Q observations, a random error of 10% is
assumed, based on the review of McMillan et al. (2012) and the study of Westerberg et al. (2016) for the UK, who
determined a median error for the mean flow of 12%. Actual percent errors are extremely variable, depending on
temporal aggregation, the Q value itself and various local conditions (Di Baldassarre and Montanari, 2009). In the
EnCDA approach, an additional error of 10% of the temporal average of the Q observation time series was applied
as this led to more stable EnCDA results.

To obtain TWSA observations for this study, level-2 GRACE data (spherical harmonic coefficients, SHC)
from TU Graz (ITSG Grace2018; Mayer-Gürr et al., 2018) were evaluated over the CDA units. These data
represent the Earth's time-variable gravity field as observed by the GRACE satellites via K-band ranging (KBR)
and GNSS tracking. We derived TWSA from SHCs up to degree and order 96, applying the DDK3 filter (Kusche
et al., 2009) and corrections for low-degree terms and effects such as glacial isostatic adjustment following



Gerdener et al. (2020). As the temporal mean value of GRACE-derived terrestrial water storage is unknown, it is a widely followed approach to normalize the monthly TWSA values relative to a constant mean over a certain reference period, here taken from 2003 to 2012. Uncertainties (1-sigma errors) were propagated to TWSA maps

based on the full variance-covariance matrix of the TU Graz data; this accounts for orbital effects and the generally meridional behavior of errors. To investigate the influence of different level-2 GRACE products, we compared the unit-averaged TWSA time series from ITSG-Grace2018 with TWSA derived from the Release-06 version of the Center for Space Research (CSR) and the Geoforschungszentrum (GFZ). For the whole MRB, 42% of the CSR and 35% of the GFZ monthly values were found within 1 standard deviation, and 76% of the CSR and 61% of the

GFZ monthly values were within 2 standard deviations of the TU Graz solution. Unexpectedly, the values are even higher for all sub-basin CDA units. Therefore, we decided to use ±2 standard deviations of the propagated GRACE uncertainties for quantifying the TWSA observation error in this study.

GRACE TWSA estimates for spatial units are affected by leakage errors that are caused by the need for spectral truncation and the need to filter the solutions, which, for averages of different spatial units, may lead to an under-

or overestimation of TWSA, thus affecting model calibration using GRACE TWSA. Therefore, when consistently comparing simulated to GRACE TWSA, it is advised to filter the simulated grid cell data with the same filter that was used to process the GRACE data (Döll et al., 2014). However, given the large number of simulations required in ensemble-based calibration, this approach is computationally impractical. To roughly estimate the leakage effect, a re-scaling factor for GRACE TWSA was estimated for each CDA unit using Eq.1 of Swenson and

Landerer (2012). The GRACE TWSA time series for CDA units can be multiplied with such a re-scaling factor to (ideally) reduce the leakage error and in this way make it better comparable to the simulated TWSA time series. First, the monthly time series of gridded TWSA as simulated by standard WaterGAP was filtered with the DDK3 filter, and then both the filtered and the unfiltered TWSA values were aggregated over all grid cells with a CDA unit. The re-scaling factor was then derived by minimizing the misfit between filtered and unfiltered TWSA time

series through a simple least square regression. The re-scaling factors are between 1.00 and 1.03 for the CDA units MRB, Missouri and Upper MRB. They are 0.90 and 0.93 for the Ohio and Arkansas River basins, respectively, and 1.41 for the Lower MRB. As the re-scaling factors are close to 1 in all CDA units except the Lower MRB and we suspect that the large re-scaling for the MRB is due to an overestimation of the TWSA trend in the Lower MRB by WaterGAP, we did not apply re-scaling factors to GRACE TWSA.

The GRACE mission relies on accelerometers to measure non-gravitational forces. However, since August 2007, battery cell failures onboard the GRACE satellites led to increasing power supply problems, especially during orbital eclipses. As a result, the thermal control of the accelerometers was deactivated in April 2011 such that thermal variations would directly increase the measurement noise. To mitigate this problem, thermal variations and their impact on the GRACE instruments are modeled during the processing at TU Graz and the accelerometer

data are calibrated (Klinger and Mayer-Gürr, 2016). This reduces the noise of the monthly gravity field solutions by an estimated 20-40% compared to solutions without accelerometer calibration (Klinger et al., 2016), but on balance, all GRACE solutions are deemed noisier from April 2011 onwards, the estimation of the noise floor is more uncertain, and the number of months without observations increases towards the end of the study period.


### 3.2.3 Climate forcing

Climate forcing required for both the irrigation water use model and WGHM encompasses time series of daily near-surface air temperature, total precipitation, downward shortwave radiation and downward longwave radiation. In this study, we applied the 0.5° GPCC-WFDEI data set where ERA-Interim reanalysis data of ECMWF have been bias-corrected by monthly precipitation time series of the Global Precipitation Climatology Centre and by other observations (Weedon et al., 2014). Monthly precipitation was corrected for wind-induced undercatch

(Weedon et al., 2014).

### 3.2.4 Calibration parameters

Many parameters in WaterGAP are spatially distributed, such as the parameter maximum soil water storage in the effective root zone $S_{max}$, which is computed as the product of soil water storage between field capacity and wilting point from a data set that provides a different value for each 0.5° cell and a rooting depth that is a fixed assigned

value for each class of land cover, with one dominant land cover per cell. Other parameters are set globally to the same value, e.g., the groundwater discharge coefficient. To enable an adjustment of the cell-specific value of a distributed parameter like $S_{max}$, one may choose to either adjust the land cover-specific rooting depth in each CDA unit or to introduce a multiplier of cell-specific $S_{max}$ as a calibration parameter. As the number of free (calibration) parameters should be limited given limited observations and equifinality, the second approach was chosen. For all

spatially distributed parameters, multipliers were introduced that serve as calibration parameters, while globally uniform parameters are directly calibrated.

In Table 2, information about the 24 potential calibration parameters that were investigated in this study is provided, including their estimated a-priori uncertainty range. They are ordered mainly according to the water storage compartment (Fig. 1) that they immediately impact due to inclusion in the respective water balance

equation. In addition, multipliers for precipitation and net radiation are included as calibration parameters, which were found to be the parameters that TWSA of the 33 largest river basins worldwide are most sensitive to (Schumacher et al., 2016b). The two multipliers for the net abstraction of groundwater and surface water are allowed to become negative as, e.g., an initially simulated positive net abstraction from groundwater (where water is removed from the ground due to pumping) may in reality be negative. The latter is the case if infiltration of

irrigation water that was taken from surface water sources dominates groundwater abstractions in the grid cell. For some parameters, the selected range was influenced by previous analyses of the WaterGAP model performance. Uniform distributions were assumed for all parameters.

The Q of larger rivers in the MRB is strongly impacted by the management of the many man-made reservoirs. The water balance of large (i.e. "global") reservoirs is simulated in WGHM with an algorithm that distinguishes

reservoirs with the main purpose of irrigation from others; different equations are used for reservoirs with a large storage capacity to mean annual Q ratio and those with a small ratio. With any globally applied algorithm, human decisions on reservoir management are very difficult to simulate, and adaptation of some parameters is not likely to lead to better simulation results unless each reservoir would be dealt with individually. Therefore, no parameter of the reservoir algorithm was adjusted in this study. This limits the ability of the calibrated model to achieve a



545 good fit to observations in river basins with many reservoirs such as the Missouri river basin (Fig. A1a in the appendix).

  From the potential calibration parameters, a small number of calibration parameters were selected for each CDA unit by a sensitivity analysis, to limit equifinality. The sensitivities of four output variables (simulated Q, TWSA, snow storage and water storage in local lakes) to all 24 parameters were analyzed separately for each of

550 the six CDA units, using the standard version of WGHM. For the sensitivity analysis, the Elementary Effect Test (EET) method of Morris (1991) was applied where the average of the elementary effects, i.e., the amount of change in the simulated variable due to a change in a parameter value, is used as the sensitivity measure or sensitivity index. The change in the variable is computed as the root mean square difference between a reference simulation and the simulation of the variable after deviating the parameter from its reference value. The EET method is

555 computationally inexpensive and recommended for parameter ranking and screening (Pianosi et al., 2016). 1000 random parameter sets were generated by Latin Hypercube Sampling and used as the reference parameter values. Then, each reference parameter set was perturbed One-at-A-Time (OAT) for each of the 24 parameters following a radial design proposed by Campolongo et al. (2011), which resulted in a total number of 25,000 (i.e., 1,000 x (1+24)) parameter sets. Parameters were ranked separately for each of the four output variables and the most

560 influential parameters for each variable were chosen. For each CDA unit, 8-10 calibration parameters were selected (Table 2). As a result, altogether 47 parameters were adjusted if the five sub-basin CDA units were used for model calibration.





**Table 2.** WGHM parameters, the range of assumed uniform a-priori distribution used for sensitivity analysis and
calibration as well as the CDA units in which parameter was adjusted in this study. The parameters are categorized
according to the processes or water storage compartments that they directly affect. P: precipitation, EP: potential
evapotranspiration, CA: canopy, SN: snow, SL: soil, GW: groundwater, SW: surface water, NA: net abstraction
of water by humans.

| Compart-ment | Parameter [units if not unitless] | Abbre-viation | Standard WGHM value | Range | Selected for adjustment in CDA units |
|---|---|---|---|---|---|
| P | Precipitation multiplier | P-PM | 1 | 0.5-2 | - |
| EP | Net radiation multiplier | EP-NM | 1 | 0.5-2 | - |
| EP | PT coeff. humid[1] | EP-PTh | 1.26 | 0.885-1.65 | All |
| EP | PT coeff. (semi)arid[2] | EP-PTa | 1.74 | 1.365-2.115 | - |
| CA | MCWH[3] [mm] | CA-MC | 0.3 | 0.1-1.4 | - |
| CA | LAI multiplier | CA-LAIM | 1 | 0.2-2.5 | - |
| SN | Snow freeze temp. [°C] | SN-FT | 0 | -1-3 | - |
| SN | Snow melt temp. [°C] | SN-MT | 0 | -3.75-3.75 | All |
| SN | Degree-day factor multiplier | SN-DM | 1 | 0.5-2 | - |
| SN | Temp. gradient [°C/m] | SN-TG | 0.006 | 0.001-0.01 | - |
| SL | $S_{max}$ multiplier[4] | SL-MSM | 1 | 0.5-3 | All |
| SL | Runoff coefficient | SL-RC | Variable | 0.3-3 | All |
| SL | Maximum EP (mm/d) | SL-MEP | 15 | 6-22 | I |
| GW | GW recharge factor mult.[5] | GW-RFM | 1 | 0.3-3 | V |
| GW | Max. GW recharge mult.[5] | GW-MM | 1 | 0.3-3 | I, III, IV |
| GW | Critical precip.[6] [mm/d] | GW-CP | 12.5 | 2.5-20 | - |
| GW | GW discharge coeff.[1/d] | GW-DC | 0.01 | 0.001-0.02 | IV |
| SW | River roughness coeff. mult. | SW-RRM | 3[7] | 1-5 | IV, V, MRB |
| SW | Active lake depth [m] | SW-LD | 5 | 1-20 | All |
| SW | Active wetland depth [m] | SW-WD | 2 | 1-20 | All |
| SW | SW discharge coeff.[8] [1/d] | SW-DC | 0.01 | 0.001-0.1 | All |
| SW | Evapo. red. factor mult.[9] | SW-ERM | 1 | 0.33-1.5 | - |
| NA | NA from GW multiplier[10] | NA-GM | 1 | -2-2 | I,II, V, MRB |
| NA | NA from SW multiplier[11] | NA-SM | 1 | -2-2 | II |

[1] Priestley-Taylor coefficient in humid grid cells
[2] Priestley-Taylor coefficient in (semi)arid grid cells
[3] Maximum water storage on canopy per Leaf Area Index (LAI)
[4] Multiplier for maximum soil water storage in the effective root zone
[5] Groundwater recharge is capped at 95% of total runoff from land $R_l$
[6] In (semi)arid grid cells, there is only GW recharge if daily precipitation exceeds the value of the parameter critical precipitation. Otherwise, the potential GW recharge remains in the soil
[7] For most river basins, including MRB
[8] For lakes and wetlands
[9] To take into account the impact of temporally varying areas of lakes, reservoirs, and wetlands on evaporation
[10] Multiplier for net abstraction from groundwater
[11] Multiplier for net abstraction from surface water (reservoirs, lakes, and rivers)

Seven parameters were selected as calibration parameters in all CDA units (Table 2). The precipitation
multiplier P-PM and the net radiation multiplier EP-NM can correct biases of the climate forcing. P-PM was
excluded from calibration even though it ranked 1st in the sensitivity analyses in all six basins for almost all four
test variables because the precipitation input is perturbed in EnCDA, and an additional multiplier would lead to a
double-counting of precipitation uncertainty. Potential evapotranspiration is a function of both net radiation and
the Priestley-Taylor coefficient. Even though EP-NM ranked somewhat higher in all CDA than the Priestley-
Taylor coefficient for humid areas EP-PTh units, we decided to adjust only EP-PTh (Table 2), as it is an actual


model parameter and not a climate forcing correction factor. The majority of the MRB is humid. Relatively small local lakes are distributed widely across the MRB (Fig. A1b), and active lake depth SW-LD and surface water

discharge coefficient SW-DC ranked highest regarding the storage dynamics of local lakes. Wetlands are abundant in the north of the MRB, forming wetland complexes where 25-50% of the land area may be covered by wetlands during wet periods (Fig. A1c); active wetland depth SW-WD was selected as TWSA was highly sensitive to it in half of the CDA units. Snow melt temperature SN-MT was selected because it was the snow-related parameter that was much more important than the other three snow parameters not only for snow storage but also for Q and

TWSA. The final two calibration parameters selected for all CDA units are the runoff coefficient SL-RC and the multiplier for maximum soil water storage SL-MSM. They are, after the three parameters P-PM, EP-NM and EP-PTh, the most influential parameters as they strongly affect Q, TWSA, and lake storage. For each CDA unit, an additional one to three calibration parameters were selected as they had a particularly high sensitivity rank due to the specific characteristics of the CDA unit. For example, the multiplier for net abstractions from groundwater

(Fig. A1d), NA-GM, was selected in four CDA units where groundwater withdrawals lead to groundwater depletion, which strongly affects TWSA while the multiplier for net abstractions from surface water (Fig. A1e), NA-SM, was only selected for the Missouri River basin. The maximum groundwater recharge multiplier GW-MM, which affects the soil texture-specific maximum amount of daily groundwater recharge, was selected in three CDA units, while the multiplier for the fraction of groundwater recharge GW-RFM was selected for one other

CDA unit. The calibration parameter maximum potential evapotranspiration SL-MEP, which limits actual evapotranspiration, was found to be influential in the driest CDA unit Arkansas River basin. Altogether, 14 out of the 24 parameters in Table 2 were selected as calibration parameters in the study on MRB.

### 3.3 Performance and uncertainty metrics

In this study, we only consider performance metrics for the simulated monthly time series of Q and TWSA as they

form the basis for calculating hydrological signatures such as drought or flow indicators that are used in global-scale water resources assessments. While the mean is an important characteristic in the case of Q, this is not true for TWSA, which is an anomaly with a zero temporal mean during the reference period. The Nash-Sutcliffe efficiency is a traditional performance metric in hydrological modeling. It provides an integrated measure of model performance concerning mean values and variability and is computed as

$$NSE = 1 - \frac{\sum_1^n (sim_{(t)} - obs_{(t)})^2}{\sum_1^n (obs_{(t)} - \mu_{obs})^2} \tag{2}$$

where $\mu_{obs}$ is the mean of observations; $sim_{(t)}$ and $obs_{(t)}$ refer to the simulated and observed values respectively at time-step $t$ of a total number of time steps $n$. The Kling-Gupta efficiency together with its three components enables distinguishing model performance regarding correlation, bias and variability (Kling et al. 2012), with

$$KGE = 1 - \sqrt{(CC - 1)^2 + (RBias - 1)^2 + (RVar - 1)^2} \tag{3}$$

where $CC$ is the correlation coefficient and

$$RBias = \frac{\mu_{sim}}{\mu_{obs}} \tag{4}$$

$$RVar = \frac{\sigma_{sim}/\mu_{sim}}{\sigma_{obs}/\mu_{obs}} \tag{5a}$$





where $\sigma$ is the standard deviation and $\mu$ is the mean; the subscript *sim* and *obs* refer to simulated variate and observations of that variate respectively. Expressing variability as the ratio of the coefficients of variation (Eq.5a)

ensures that bias and variability are not cross-correlated (Kling et al. 2012). In the case of TWSA, the bias is set to 1 in the computation of KGE, and

$$RVar = \frac{\sigma_{sim}}{\sigma_{obs}}. \hspace{3cm} (5b)$$

The optimal value of all the above performance metrics is one.

The uncertainty of model output as derived from the model output ensemble can be quantified by two

uncertainty metrics. In the case of Q, the average uncertainty bandwidth (AUBW) is expressed as a fraction of the ensemble mean (modified from Jin et al. 2010), with

$$AUBW_Q = \frac{1}{n}\sum_1^n \frac{UpperLimit(t) - LowerLimit(t)}{EnsembleMean(t)} \hspace{2cm} (6)$$

where *t* refers to the month and *n* is the total number of months. In the case of TWSA,

$$AUBW_{TWSA} = \frac{1}{n}\sum_1^n UpperLimit(t) - LowerLimit(t). \hspace{2cm} (7)$$

$AUBW_Q$ can be expressed in %, while the unit of $AUBW_{TWSA}$ is mm. Here, the highest and lowest values among all ensemble members (32 in the case of EnCDA, values for POC and GLUE listed in Table 6) are used as upper and lower limits in each month and make up the uncertainty bounds of the simulation. The metric "coverage of observations by model output" (CO) is calculated as the percentage of monthly observations including their uncertainty bounds (derived from observation errors described in Section 3.2.2) that are contained within the

uncertainty bounds of the model output. A large CO value and a small AUBW value indicate a low model output uncertainty.

### 3.4 Implementation of calibration approaches in this study

#### 3.4.1 POC

The state-of-the-art optimization algorithm Borg-MOEA (Borg Multiobjective Evolutionary Algorithm; Hadka

and Reed, 2013) was applied to search the parameter space to find Pareto-optimal parameter sets. Borg MOEA not only amalgamates search operators (i.e., algorithms to generate a new generation of solutions from their parents) and strategies from benchmark optimization algorithms like NSGA-II, ε-NSGA-II, ε-MOEA and GDE3 but also has the capability of exploiting these operators based on their performance of producing better off-springs for the optimization problem at hand. Apart from the auto-adaptive operator recombination strategy, Borg MOEA

includes a restart mechanism upon the occurrence of a search stagnation and strategies like population resizing and adaptive archive sizing. The NSE of monthly time series of Q and TWSA in the calibration period, $NSE_Q$ and $NSE_{TWSA,}$ were chosen as the two objective functions. For all CDA units, the initial population size was 400 and the improvement threshold ε (i.e., the side length of the ε-box) was set to 0.005 for all objectives. All other parameters of the algorithm were set to their recommended values (Hadka and Reed, 2013). Due to the high

computational demand of WHGM, we restricted each calibration to a maximum of 20,000 model runs. The POC application was run in parallel using openmpi-4.0.1 on 401 nodes of a Linux cluster machine with a Scientific Linux 7 environment.

All WHGM model runs for the six CDA units started in 1991. Calibration of the five sub-basin CDA units was done sequentially as follows. First, the four upstream CDA units (Fig. 2) were calibrated independently from each other. Q and total water storage in the downstream CDA unit V Lower MRB depends on inflow from the four upstream CDA units. For each upstream CDA unit, the parameter set resulting in the highest $NSE_Q$ at the respective calibration station was selected to transfer the best estimate of monthly Q to the downstream CDA unit. These parameter sets were then used in the calibration of the downstream CDA unit, which required running the model for the whole MRB.

### 3.4.2 GLUE

For each of the six CDA units, a random ensemble of 20,000 parameter sets was generated by Latin Hypercube Sampling (Campolongo et al., 2011), only varying the 8-10 influential parameters indicated in Table 2. Then, individual WGHM model runs were performed for the MRB and the four upstream CDA units (Fig. 2). Similar to the POC approach, all ensemble runs for the downstream CDA unit V Lower MRB were performed using, for each of the four upstream CDA units, the GLUE parameter sets that resulted in the highest $NSE_Q$ at the upstream calibration station. Like in the POC approach, all GLUE runs started in 1991 and were done in parallel on 401 nodes of a Linux cluster machine. Monthly time series of spatially averaged TWSA as well as Q at the calibration and validation stations during both the calibration and validation period were written as output, and the performance metrics (Section 3.3.) were computed. To identify behavioral and Pareto-optimal parameter sets as well as the compromise parameter sets (Eq. 1), $NSE_Q$ and $NSE_{TWSA}$ were used as likelihood measures.

To assess the impact of observation errors of Q and TWSA on model performance, the monthly time series of observed Q and TWSA were perturbed based on the observation errors described in Section 3.2.2. A uniform distribution of errors with the ranges of ±10% was assumed for Q and ±2 standard deviations of the computed GRACE error distribution for TWSA (see Section 3.2.2). 1,000 realizations of observations of Q and TWSA were generated. Then, $NSE_Q$ and $NSE_{TWSA}$ values for each of the 1,000 perturbed observation time series compared to each of the 20,000 WaterGAP time series were computed. Finally, the Pareto-optimal parameter sets for each of the 1000 realizations of observations were identified. This approach for taking into account observation uncertainty for selecting behavioral parameter sets is similar to the approach taken by Blazkova and Beven (2009).

### 3.4.3 EnCDA

EnCDA was performed by coupling the Parallel Data Assimilation Framework (PDAF; Nerger and Hiller, 2013), which implements an EnKF approach, to WGHM. 32 ensemble members were generated by perturbing forcing data and calibration parameters. Regarding the forcing data, an additive error of plus/minus 2°C for the temperature (with a triangular distribution around 0) and a multiplicative error of plus/minus 10% regarding the precipitation perturbation (with a triangular distribution around 1) (Eicker et al., 2014) was used. For each ensemble member, this error was set individually for each month and grid cell and applied to the daily forcing values. A spin-up phase run over 1991-2002 was performed to generate initial conditions for the calibration period. The EnKF is used to simultaneously update model parameters and storages during the calibration period 2003-2012 following Eicker





et al. (2014) and Schumacher et al. (2016a,b) but considering Q observations in addition to GRACE TWSA. For this, the state vector is augmented by CDA unit-specific calibration parameters. To avoid the system being

underdetermined, TWSA in 4° grid cells instead of TWSA averages over the CDA units were assimilated. Calibration parameters and water storages were adjusted with monthly time steps,

Simulations for the validation period 2013-2016 were done by continuing the 32 model runs of the calibration period with the 32 parameter sets estimated for December 2012, without any data assimilation. The ensemble mean of the simulated output variables of the 32 ensemble runs during the validation period is assumed to be the best

estimate of the time series of output variables. The EnCDA application was run in parallel using openmpi-3.1.4 on a Linux cluster machine with a Linux CentOS 7.9 environment and 70 nodes

In the case of the CDA unit covering the whole MRB, the EnCDA was performed by the parameters indicated in Table 2 while assimilating GRACE TWSA 4° grids over the whole basin as well as Q at the Vicksburg gauge station. For the sub-basin calibration, the EnCDA was applied separately to the four upstream CDA units first.

Then, the parameter sets of each ensemble member of the four upstream CDA units were set to the values obtained for December 2012. For calibrating the downstream CDA unit V by EnCDA, the 32 parameter sets in each of the four upstream CDA units were held constant, and states in these CDA units were not updated by DA. Parameters were perturbed independently per CDA unit without generating spatial correlations as different parameters are considered for the different CDA units (Table 2). An attempt to simultaneously calibrate all five CDA units was

not successful. Different from POC and GLUE, the performance metric NSE was not used to generate the calibrated parameter ensemble but only to determine behavioral parameter sets and the compromise parameter sets as well as for model output validation.

## 4 Results

Multi-objective parameter estimation may be aimed at determining 1) an optimal model parameter set that is

identified by weighting the multiple calibration objectives, e.g., the compromise solution (Eq. 1), 2) Pareto-optimal parameter sets or 3) an ensemble of behavioral parameter sets that leads to model output that fits reasonably well to observations given observation and other uncertainties. In any case, the calibrated parameter sets are specific to the applied model structure and input, including climate forcing, net abstractions of surface water and groundwater as well as physiographic characteristics such as the existence of surface water bodies or soil properties per grid

cell.

### 4. 1 Model performance during the calibration period 2003-2012

#### 4.1.1 Optimal parameter sets

*Differences between calibration approaches.* Table 3 and Fig. 3 show the performance of the (Pareto-)optimal parameter sets as measured by $NSE_Q$ and $NSE_{TWSA}$. Due to the applied search algorithm, the POC approach is

superior to the GLUE approach in identifying Pareto-optimal parameter sets. In all six CDA units, the POC parameter sets lead to higher NSE values than the GLUE parameter sets, for the compromise parameter set as well





as for the parameter sets that lead to either the highest $NSE_Q$ or the highest $NSE_{TWSA}$. In the case of GLUE, the 20,000 ensemble members are randomly distributed in the parameter space, while the evolutionary Borg-MOEA optimization algorithm applied in POC creates many more parameter sets that are close to the Pareto front while also requiring 20,000 model runs (Fig. S1 in the supplement). For the example of the CDA unit Arkansas River basin, the POC compromise parameter set leads NSE values of 0.74 and 0.85 for Q and TWSA, respectively, while the corresponding values in the case of GLUE are, with 0.69 and 0.83, slightly lower. Except in the Upper MRB, the performance of EnCDA-derived parameter sets is lower than of those derived by POC and GLUE. This is surprising as not only parameters but also water storages are modified each month during the calibration period to obtain a better fit to observed TWSA and Q. However, a weighted RMSE and not NSE is optimized in EnCDA, which may cause the lower NSE values. The weaker calibration success of EnCDA may also be due to the small ensemble size of only 32. The EnCDA compromise solution as well as the EnCDA ensemble mean perform better than POC and GLUE in the CDA unit that is characterized by many small wetlands, the Upper MRB, where WaterGAP shows the worst performance regarding TWSA. EnCDA for the MRB as one CDA unit leads to very poor results in particular regarding TWSA, with NSE values below 0.25 for both the compromise solution and the ensemble mean and even for the ensemble member leading to the largest $NSE_{TWSA}$.

**Table 3.** Performance of optimal parameter sets quantified by $NSE_Q$ and $NSE_{TWSA}$ in the different CDA units. NSE of parameter sets achieving the highest $NSE_Q$ or the highest $NSE_{TWSA}$, and of the compromise solution are listed as well as the NSE values of the EnCDA ensemble mean, the standard WaterGAP 2.2d model and an uncalibrated version of the WaterGAP 2.2d model. Results are provided for the calibration period 2003-2012. The compromise solutions were identified from Eq. 1 using p = 2. The best-performing calibration approach per CDA unit, with the highest average NSE, is indicated in bold. The 77 CDA units of the standard calibration are shown in Figs. S2 and S3.

| | $NSE_Q/NSE_{TWSA}$ | | | | | |
| | Arkansas | Missouri | Upper MRB | Ohio | Lower MRB | MRB |
|---|---|---|---|---|---|---|
| POC: highest $NSE_Q$ | 0.74/0.85 | 0.83/0.50 | 0.82/0.27 | 0.89/0.82 | 0.90/0.69 | 0.90/0.51 |
| POC: highest $NSE_{TWSA}$ | 0.63/0.89 | -0.82/0.81 | 0.14/0.65 | 0.73/0.90 | 0.85/0.93 | 0.28/0.84 |
| POC: compromise | **0.74/0.85** | **0.73/0.71** | 0.67/0.48 | **0.87/0.86** | **0.87/0.91** | **0.83/0.73** |
| GLUE: highest $NSE_Q$ | 0.70/0.79 | 0.77/0.21 | 0.78/0.18 | 0.88/0.81 | 0.87/0.26 | 0.88/0.19 |
| GLUE: highest $NSE_{TWSA}$ | 0.24/0.88 | -0.68/ 0.76 | 0.01/0.61 | 0.68/0.90 | 0.80/0.90 | 0.33/0.81 |
| GLUE: compromise | 0.69/0.83 | 0.65/0.71 | 0.61/0.46 | 0.86/0.84 | 0.84/0.89 | 0.85/0.65 |
| EnCDA: highest $NSE_Q$ | 0.61/0.51 | 0.69/0.59 | 0.70/0.49 | 0.79/0.91 | 0.83/0.88 | 0.54/0.13 |
| EnCDA: highest $NSE_{TWSA}$ | 0.59/0.84 | 0.40/0.66 | 0.07/0.67 | 0.63/0.94 | 0.74/0.91 | 0.44/0.23 |
| EnCDA: compromise | 0.59/0.84 | 0.62/0.65 | 0.68/0.60 | 0.79/0.91 | 0.83/0.88 | 0.51/0.19 |
| EnCDA: ensemble mean | 0.61/0.78 | 0.55/0.57 | **0.70/0.61** | 0.73/0.88 | 0.76/0.90 | 0.49/0.14 |
| Standard calibration[1] | 0.59/0.55 | 0.53/0.38 | 0.54/0.18 | 0.86/0.77 | 0.79/ -0.04 | 0.79/0.35 |
| Uncalibrated[2] | 0.18/0.67 | -1.02/0.38 | 0.56/0.17 | 0.85/0.72 | 0.71/0.06 | 0.71/0.38 |

[1] SL-RC and two correction factors are adjusted in 77 CDA units within the MRB, using observations of mean annual Q; calibration period 1980-2009
[2] SL-RC = 2, correction factors equal to 1



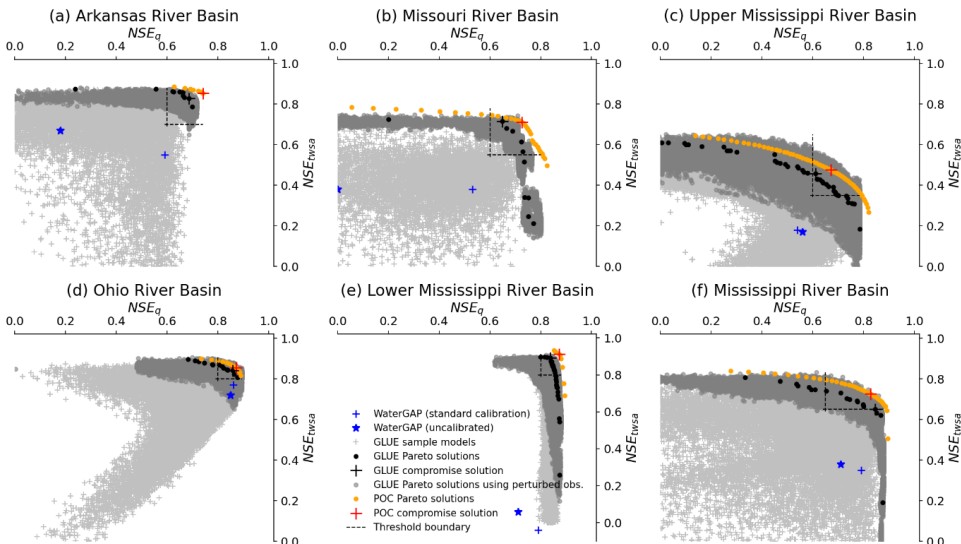

**Figure 3**. Performance of 1) Pareto-optimal solutions derived by an evolutionary optimization algorithm (POC) (orange dots), 2) the GLUE ensemble (light grey pluses), and 3) the Pareto-optimal subset of the GLUE ensemble (black dots), in all cases neglecting observation error when computing NSE. In addition, the performance of 4) the Pareto-optimal GLUE parameter subset for 1,000 realizations of perturbed observations are shown (dark grey dots), which shows the impact of observation errors on KGE. Compromise solutions of both POC and GLUE approaches are shown, too, together with the model performance after standard calibration and without calibration, consistent with Table 3. The thresholds for behavioral parameter sets (Table 6) are indicated by the grey dashed lines.

*Differences between CDA units.* Optimal performance strongly varies between the CDA units. The best performance with optimized parameter sets is achieved for the humid and hilly Ohio River basin and the downstream Lower MRB, with NSE values exceeding 0.85 for both Q and TWSA in the POC compromise solution (Table 3). Q in the Lower MRB is heavily determined by inflow from the four upstream CDA units. In the relatively dry Arkansas River basin, model performance regarding TWSA is similar to the two best-performing CDA units but, with 0.74, somewhat worse regarding Q. In the Missouri River basin and, in particular, in the Upper MRB, TSWA fit to GRACE observations is worse than in the other three sub-basins. Inadequate modeling of both man-made reservoirs and wetlands is suspected to cause the low performance regarding TWSA in both basins. The Missouri River basin is the basin that is most strongly impacted by man-made reservoirs (Fig. A1a). No parameters of the reservoir algorithm were calibrated (see Section 3.2.4). The northern parts of both basins (dark blue areas of Fig. A1c) are characterized by the existence of a high number of small wetlands whose location and extent are poorly quantified in WaterGAP. This stems from the classification of this whole area, in the Global Lakes and Wetland Database GLWD (Lehner and Döll, 2004), as a "wetland complex with a 25-50% coverage" with wetlands at maximum extent. This coarse information is included in WaterGAP by assigning a maximum





extent of local wetlands of 35% of the cell area (Döll et al., 2020). Thus, it is not only the WaterGAP algorithms
for simulating the water balance of wetlands but very likely also the poor localization of wetlands that prevent
parameter adjustment to result in good fits to observations. We speculate that for these conditions, modification of
water storages in EnCDA leads to an improved simulation of TWSA, and, to a smaller degree, of Q (Table 3). In
the case of CDA unit MRB, where all grid cells of the whole MRB are assigned the same value of the calibration
parameters (Table 3), $NSE_Q$ is, with a value of 0.83 for POC and GLUE, very similar to the two best-performing
sub-basins Ohio and Lower MRB. With a value of 0.73, $NSE_{TWSA}$ is within the range of the respective values of
all sub-basin CDA units.

*Benefits of multi-variable calibration.* The performance of the compromise solutions is compared to the
performance of the WaterGAP variant that is calibrated in the standard way (Sect. 3.1) and of an uncalibrated
WaterGAP variant. In the standard calibration, the runoff coefficient SL-RC and potentially two correction factors
are adjusted individually for each of 77 sub-basins (CDA units) using only observations of mean annual Q at the
sub-basin outlet (Figs. S2 and S3). In the uncalibrated variant, SL-RC is set to 2 and the correction factors to 1
throughout the MRB. For all CDA units, POC and GLUE compromise parameter sets result in higher NSE values
for both Q and TWSA as compared to both the uncalibrated and the standard model variant (Table 3 and Fig. 3).
This is also true for EnCDA except for the CDA unit MRB, where both $NSE_Q$ and $NSE_{TWSA}$ are worse than in
both the uncalibrated and standard WaterGAP variant, and the Ohio River basin where $NSE_{TWSA}$ is increased but
$NSE_Q$ decreased by EnCDA. In the case of the Ohio River basin, neither the standard calibration nor the
POC/GLUE compromise solutions achieve a significant improvement of the already high $NSE_Q$ of the uncalibrated
model, and even the improvement of TWSA simulation is rather small. As can be expected, the fit to observed
TWSA is improved more strongly in comparison to the standard calibration than the fit to observed Q, with the
strongest improvement in the small downstream Lower MRB.

Analysis of the KGE components CC, RBias and RVar (Eqs. 3-5) (Tables B1 and B2) shows that the improved
$NSE_Q$ and $NSE_{TWSA}$ of the compromise solutions of POC, GLUE and EnCDA as compared to the standard
WaterGAP results are, in all CDA units, mainly due to an improvement of the correlation (CC), the exception
being $NSE_Q$ in case of EnCDA. Standard calibration only improves the bias of Q compared to the uncalibrated
variant, mostly leading to an RBias value close to 1 (Table B1). The multi-variable approaches decrease the
overestimation of mean annual Q by the uncalibrated model except in the Upper MRB and the Ohio River basin,
where the overestimation by the uncalibrated model is already very small. However, as compared to the standard
and uncalibrated model variants, none of the three calibration approaches improves the strong underestimation of
Q variability by WaterGAP. Q variability in the compromise solutions becomes even more strongly
underestimated, in the Upper and Lower MRB and for the whole MRB. TWSA variability in the Arkansas and
Missouri River basins and the Lower MRB is improved as compared to the standard and uncalibrated WaterGAP
but worsened in the case of the wetland-rich Upper MRB (Table B2).

Overestimation of observed seasonal low flows prevails in all CDA units, not only in the compromise solutions
(Figs. 4 and S4) but also in the solutions showing the highest $NSE_Q$. The improved correlation but stronger
underestimation of Q variability as compared to the standard calibration can be seen in the hydrograph of observed
and simulated Q for the CDA Unit MRB, for POC and GLUE compromise solutions (Fig. 4a); the seasonal low
flows are better captured with the standard calibration than with the compromise solutions. Correlation of





simulated and observed TWSA is improved by achieving a small phase of seasonal dynamics shift (towards later in the year) by POC/GLUE, but in some years (e.g., 2008 and 2009), TWSA rise still occurs too early (Fig. 4b). In addition, the relatively high water storage at the end of the years 2010 and 2011 cannot be captured by any

simulation. These discrepancies in average TWSA over the MRB can be traced back to the Missouri and Upper MRB sub-basins where in many years, simulated TWSA increases too quickly and too much in the first half of the year (Figs. S4b, d).

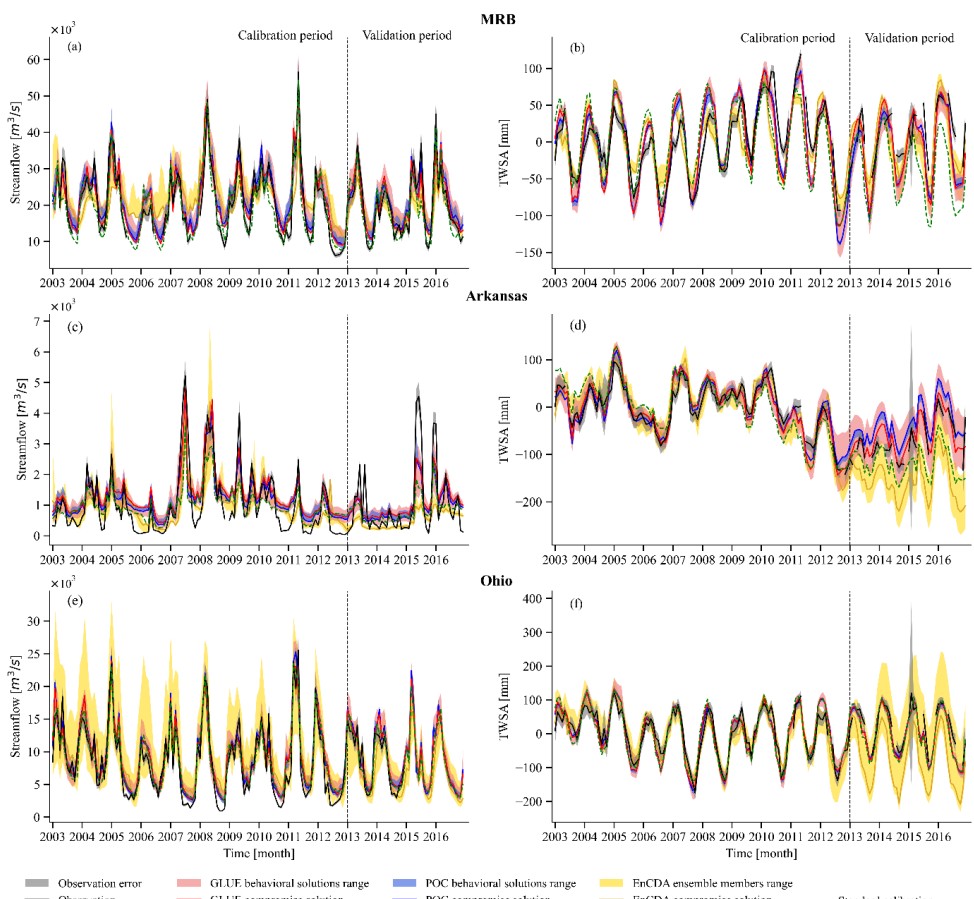

**Figure 4.** Monthly time series of simulated and observed Q (a, c, e) and TWSA (b, d, f) during calibration period

2003-2012 and validation period 2013-2016 for MRB (a, b), Arkansas River basin (c, d) and Ohio River basin (e, f). Observations and their assumed errors are shown together with simulated GLUE, POC, and EnCDA compromise solution, with the range of GLUE and POC behavioral solutions (maximum and minimum monthly values of the behavioral solutions, Table 6) and the range of all 32 EnCDA ensemble members, as well as with the WaterGAP variant with standard calibration.




Q is temporally more variable in the Arkansas River basin than in the MRB (Fig. 4c). The seasonal low flows in the Arkansas River basin are extremely overestimated by all WaterGAP model variants, with EnCDA sometimes reaching observed low values but not simulating the temporal dynamics and POC achieving a slightly lower overestimation than GLUE. Simulation of high flows was improved by multi-variable calibration (Fig. 4c). TWSA

performance of the compromise solutions is much better than that of the standard WaterGAP (Fig. 4d). The Ohio River basin is the CDA unit with the best model performance and little change due to any calibration, except a slight improvement of TWSA correlation (Fig. 4e, f). However, also here an overestimation of seasonal low flows in about half of the calibration years cannot be improved by parameter adjustment (Fig. 4e). Altogether, the visual inspection of the hydrographs of all six CDA units reveals that even if multi-variable calibration leads to improved

performance metrics, fit to observations can only be slightly improved as compared to the standard calibration (Figs. 4 and S4), except for the much-improved fit to TWSA in the Lower MRB (Fig. S4f).

*Trade-offs between optimal fit to Q and TWSA*. Trade-offs are large for all three calibration approaches, as quantified by the NSE values for the model runs achieving the highest $NSE_Q$ and $NSE_{TWSA}$, except in the two CDA units with an already satisfactory $NSE_{TWSA}$ in the uncalibrated model variant (Arkansas and Ohio River basins).

The optimal fit to observed TWSA then results in very poor fits to observed Q, in particular for the Missouri River basin and the Upper MRB (Table 3). Considering POC, optimal TWSA performance leads to a stronger overestimation of mean Q of 27-73% as compared to 1-18% in the case of optimal Q performance (excluding the downstream Lower MRB) (Table B1). While the ratio of simulated to observed variability of TWSA decreases and thus improves, the corresponding ratio for Q decreases, too, but thus becomes worse. $RVar_Q$ ranges from 0.80

to 0.88 in the case of maximum $NSE_Q$ and decreases to the range of 0.53-0.84 in the case of maximum $NSE_Q$ (except for the Arkansas River basin). Considering POC in the Missouri River basin as an example, the parameter set with the best fit to observed TWSA results in $NSE_{TWSA}$ of 0.81 but a negative $NSE_Q$; the parameter set with the best fit to Q achieves an $NSE_Q$ of 0.83 but $NSE_{TWSA}$ deteriorates to 0.50 (Table 3). The parameter set with optimal fit to TWSA leads to an even higher overestimation of mean Q (RBias = 1.73) and an even higher

underestimation of Q variability (RVar = 0.61) as compared to the ensemble member with the best fit to observed Q (RBias = 1.08, RVar = 0.80), while correlation slightly decreases (Table B1). KGE components regarding TWSA for the same CDA unit reveal that the correlation of observed and simulated TWSA strongly decreases from 0.91 to 0.77 if optimization is done for Q instead of TWSA, while variability is overestimated somewhat more (RVar = 1.09 instead of 1.03) (Table B2). Similar patterns are observed for the CDA units MRB and Upper

MRB. In the case of the Arkansas River basin and the Lower MRB, trade-offs between optimal fits to Q and TWSA observations identified by POC are lower than those identified by GLUE, which shows the advantage of the search algorithm applied in POC.

### 4.1.2 Behavioral parameter sets

We identified behavioral parameter sets using thresholds for minimum acceptable performance in terms of $NSE_Q$

and $NSE_{TWSA}$, taking into account the observation uncertainties of Q and TWSA. To do this, we evaluated the performance of the 20,000 simulated GLUE ensemble members with respect to uncertainty-perturbed observations (Fig. 3 and S1), as described in Sect. 3.4.2. For GLUE and EnCDA, all parameter sets within the thresholds were





selected as behavioral, while for POC, the behavioral parameter sets are the subset of Pareto-optimal parameter sets above the thresholds. The Pareto-optimal GLUE model runs for 1,000 perturbed observation time series (dark grey dots in Fig. 3) served to assess the impact of observation uncertainty on performance. Not each dark grey dot represents a different parameter set because the NSE for the same parameter set varies with the perturbed observation time series. The width of the band of the Pareto-optimal model runs in case of perturbed observations close to the compromise solution helped to identify the thresholds for $NSE_Q$ and $NSE_{TWSA}$. In the case of the poorly simulated Upper MRB, we decided to keep the thresholds above those indicated by the observation error analysis to avoid calling very poorly performing parameter ensembles behavioral (Fig. 3). We chose the compromise solution as the point of departure as we wish to give equal weight to performance for Q and TWSA. Thresholds for behavioral parameter sets vary between the CDA units due to the different optimal performances that can be achieved, in the different CDA units, by varying parameters, given a fixed model structure and the model input. The selected thresholds for behavioral solutions are indicated in Fig. 3 and Table 4, while Table 4 also provides the number of behavioral POC and GLUE parameter sets as well as of the behavioral EnCDA ensemble members.

**Table 4:** Number of identified behavioral parameter sets (or ensemble members) for each CDA unit that result in simulation results that exceed both the $NSE_Q$ and $NSE_{TWSA}$ thresholds. Listed are the number of behavioral parameter sets in the GLUE approach (out of 20,000 per CDA unit), the number of behavior Pareto-optimal parameter sets in the POC approach (out of 20,000) and the number of behavioral EnCDA ensemble members (out of 32).

|  | Thresholds for behavioral ensemble members NSE [Q, TWSA] | Number of behavioral GLUE parameter sets | Number of behavioral Pareto-optimal POC parameter sets | Number of behavioral EnCDA ensemble members |
|---|---|---|---|---|
| I Arkansas | [0.60, 0.70] | 668 | 8 | 5 |
| II Missouri | [0.60, 0.55] | 72 | 24 | 3 |
| III Upper MRB | [0.60, 0.35] | 156 | 30 | 19 |
| IV Ohio | [0.80, 0.80] | 196 | 11 | 0 |
| V Lower MRB | [0.80, 0.80] | 1517 | 7 | 6 |
| IV MRB | [0.65, 0.65] | 138 | 26 | 0 |

In the case of POC and GLUE, an uncertainty band is delineated by the minima and maxima of monthly Q or TWSA values when considering all behavioral parameter sets (Figs. 4 and S4). For EnCDA, these figures show the range of all 32 ensemble members, also because there are no behavioral EnCDA members in the case of CDA units Ohio and MRB. AUBW and coverage of observations (including their uncertainty) by the uncertainty band of the model output can be expected to correlate (Section 3.3). Both AUBW and the coverage are smaller for POC and EnCDA than for GLUE (Table 5) due to their smaller number of behavioral ensemble members. When extending the considered EnCDA ensemble members to the whole ensemble of 32 members, the coverage increases slightly, but at the same time, the width of the uncertainty bands increases strongly (Table 5). Comparing the six CDA units, neither AUBW nor coverage correlates with the number of behavioral ensemble members.



**Table 5.** Coverage of monthly observations by model output (CO), in % of monthly observations contained in uncertainty bound of observations, and average uncertainty bandwidth AUBW during the calibration period 2003-

2012 for both Q and TWSA, considering only the behavioral parameter sets (Table 4). In the case of EnCDA, also the values for the whole ensemble of 32 members are shown in parentheses. AUBW for Q is listed in %, AUBW for TWSA in mm.

| | Q/TWSA | | | | | |
| --- | --- | --- | --- | --- | --- | --- |
| | Arkansas | Missouri | Upper MRB | Ohio | Lower MRB | MRB |
| POC: Coverage | 24/70 | 55/40 | 29/42 | 49/67 | 48/90 | 52/37 |
| GLUE: Coverage | 46/94 | 72/57 | 45/61 | 72/87 | 58/95 | 58/59 |
| EnCDA: Coverage | 15/63 | 36/45 | 44/75 | -/-[1] | 57/67 | -/-[1] |
| | (25/67) | (37/48) | (53/74) | (55/91) | (60/65) | (36/35) |
| POC: AUBW | 22/6 | 26/12 | 16/8 | 17/10 | 7/23 | 19/8 |
| GLUE: AUBW | 60/49 | 41/28 | 35/29 | 43/43 | 21/82 | 32/26 |
| EnCDA: AUBW | 20/19 | 17/10 | 51/50 | -/-[1] | 16/27 | -/-[1] |
| | (63/49) | (60/38) | (78/56) | (96/63) | (48/37) | (24/18) |

[1]No behavioral parameter sets identified

For POC and GLUE, the average width of the uncertainty bands for Q in the six CDA units is 7-26% and 21-60% of the ensemble mean of monthly Q, respectively. For GLUE, the lowest AUBW occurs in the downstream Lower MRB and the highest in the Arkansas River basin (Table 7). However, even the wider GLUE bands do not cover most of the observed seasonal low flows (including the rather small observation error bands) in all CDA units, while high flow months are covered more often (Figs. 4 and S4). Coverage in the GLUE approach ranges

from 46% to 72% of the observed Q values among the six CDA units, with the lowest values for the two CDA units with the highest underestimation of Q variability, Arkansas and Upper MRB, even though the Arkansas River basin has the widest uncertainty band.

Coverage of observations including their error range by the uncertainty band is, in the case of GLUE and POC, higher for TWSA than for Q except for Missouri and MRB (Table 7). In the case of GLUE, TWSA coverage

ranges from 59% to 95%. The Arkansas River basin has a low Q coverage but a very high TWSA coverage, while the Missouri River basin has the highest Q coverage and the lowest TWSA coverage even though for the Missouri River basin, the Q performance of the compromise solution is relatively poor (Tables 3 and 4). The TWSA time series for the Arkansas River basin differs from those of the other CDA units by its high ratio of interannual to seasonal variability (Fig. 4).

**4.2 Model performance during the validation period 2013-2016**

Model performance of both the POC and GLUE compromise solutions in the validation periods is similar to the calibration periods regarding Q but much worse regarding TWSA (compare Table 6 to Table 3 for NSE values). For most CDA units and calibration approaches, the performance loss regarding TWSA between the calibration and the validation period is similarly high for the ensemble members that were identified as having the best fit to

TWSA. We suspect that the poor fit of simulated TWSA to observed TWSA in the last years of the GRACE mission, where there is also a large fraction of missing monthly GRACE data (Figs. 4 and S4), is related to increased observational errors (compare Sect. 3.2.2). This suspicion is supported by the fact that $NSE_{TWSA}$ of the





uncalibrated model is lower for the validation period than for the calibration period, which is not the case for $NSE_Q$ in all CDA units except the Arkansas River basin.


**Table 6.** Model performance during the validation period 2013-2016 indicated by $NSE_Q$ and $NSE_{TWSA}$, as achieved by the three calibration approaches POC, GLUE, and EnCDA as well as by the standard WaterGAP 2.2d and the uncalibrated WaterGAP 2.2d models. The best-performing calibration approach per CDA unit, with the highest average NSE, is indicated in bold. The indication "highest $NSE_{TWSA}$" refers to the parameter with the best performance during the calibration period. The values in parenthesis in the line "EnCDA compromise" are $NSE_{TWSA}$ values that are computed after normalizing TWSA during the validation period by the mean TWSA of the validation period.


|  | $NSE_Q/NSE_{TWSA}$ | | | | | |
|---|---|---|---|---|---|---|
|  | Arkansas | Missouri | Upper MRB | Ohio | Lower MRB | MRB |
| POC: compromise solution | 0.59/-0.04 | **0.72/-2.76** | **0.79/-0.05** | **0.85/0.75** | **0.87/0.80** | **0.85/0.31** |
| POC: ensemble mean[1] | 0.62/0.17 | 0.73/-3.18 | 0.81/-0.09 | **0.84/0.76** | **0.86/0.81** | 0.83/0.32 |
| GLUE: compromise solution | **0.61/0.66** | 0.68/-3.44 | 0.74/0.02 | 0.86/0.72 | 0.84/0.77 | 0.84/0.11 |
| GLUE: ensemble mean[2] | 0.49/0.36 | 0.65/-2.00 | 0.71/0.02 | 0.81/0.70 | 0.83/0.75 | 0.73/0.28 |
| EnCDA: compromise | 0.07/-3.99 (0.11) | 0.02/-0.30 (-0.30) | 0.68/-0.07 (-0.07) | 0.74/-2.60 (0.20) | 0.76/-0.66 (0.43) | 0.61/-1.72 (-1.00) |
| EnCDA: ensemble mean[3] | 0.07/-2.90 | -2.71/-0.94 | 0.62/-0.04 | 0.75/0.18 | 0.67/-0.44 | 0.61/-2.14 |
| POC: highest $NSE_{TWSA}$ | 0.64/0.36 | -0.45/-1.99 | 0.53/0.13 | 0.58/0.80 | 0.85/0.82 | 0.31/0.45 |
| GLUE: highest $NSE_{TWSA}$ | 0.45/-0.02 | -0.35/-0.77 | 0.46/0.15 | 0.50/0.80 | 0.81/0.82 | 0.38/0.36 |
| EnCDA: highest $NSE_{TWSA}$ | 0.07/-3.99 | -14.08/-10.60 | 0.63/0.20 | 0.75/-0.08 | 0.66/-1.08 | 0.56/-2.87 |
| Standard calibration | 0.44/-0.85 | 0.60/-3.70 | 0.47/-0.40 | 0.85/0.62 | 0.76/-6.24 | 0.76/-2.38 |
| Uncalibrated | 0.56/0.22 | -0.80/-2.2 | 0.59/-0.39 | 0.82/0.52 | 0.75/-5.60 | 0.75/-1.58 |

[1]Computed by running WGHM with the ensemble of behavioral Pareto-optimal parameter sets identified using POC (Table 6)

[2]Computed by running WGHM with the ensemble of behavioral parameter sets identified using GLUE (Table 6)
[3]Computed by running WGHM with the ensemble of 32 parameter sets identified using EnCDA (Section 4.1.3)

All compromise solutions perform somewhat better than the WaterGAP standard variant, except for EnCDA in the CDA units Missouri, Ohio and MRB (Table 6). Performances of the ensemble mean of the behavioral GLUE

parameter sets, of the ensemble mean of the behavioral Pareto-optimal POC parameter sets and of the EnCDA ensemble mean are similar to their respective compromise solutions (Table 6). In all CDA units, POC and GLUE perform better than EnCDA regarding both Q and TWSA. POC results are slightly better than GLUE results, the exception being the Arkansas River basin where POC performance regarding TWSA degrades from its high level during the calibration period due to overestimating mean TWSA (Fig. 4).



The temporal mean value of GRACE-derived TWSA is generally unknown. The standard approach taken in
this study of normalizing TWSA values to a constant mean over the reference period, here 2003-2012, may be
problematic as it assumes that the mean derived over longer periods than the reference period (here 11 years)
remains at the reference period value, which need not be true. Therefore, we additionally calculated, for the
example of the EnCDA compromise solution, the $NSE_{TWSA}$ after reducing the TWSA time series by its temporal
mean of the validation period instead of the mean of the calibration period. The resulting $NSE_{TWSA}$ values are, for
most CDA units, somewhat improved (Table 6).

### 4.3 Characterization of estimated parameter sets

POC and GLUE identify parameter sets that are assumed to be temporally constant. Here, we compare these two
ensembles of estimated parameter sets. Starting with the CDA unit MRB, we first characterize the parameter sets
of the POC compromise solution and the parameter sets leading to the best fit to either Q or TWSA. We compare
the parameter set of the GLUE compromise solution to the parameter set of the POC compromise solution. Then,
we describe the POC behavioral Pareto-optimal parameter sets as well as the GLUE behavioral parameter sets,
including parameter correlations. Finally, we highlight the most interesting results for the five sub-basin CDA
units. The EnCDA parameter sets are not considered as the EnCDA approach leads to a lower model performance
than POC and GLUE.

### 4.3.1 CDA unit MRB

*Parameter set of the POC compromise solution.* In the compromise solution, the runoff coefficient SL-RC is close
to the maximum value of 3, minimizing runoff at a given soil water saturation (Fig. 5f). This SL-RC is in line with
the values obtained by the standard calibration where calibrated SL-RC are also very high (Fig. S3b). While in the
standard calibration, one or two correction factors are needed in most standard calibration CDA units to decrease
mean annual runoff to the observed values, this is achieved in this study by a high value of SL-MSM, the multiplier
for the standard maximum soil water storage, which is adjusted in the POC compromise solution to a high value
of 2.5. A "deeper soil" with higher water storage capacity leads to decreased soil saturation and lower runoff, and
at the same time to higher variability of soil water storage and thus TWSA. EP-PTh, affecting potential
evapotranspiration, is reduced from its standard value of 1.26 to 1.02, which seems to contradict the adjustment of
both SL-RC and SL-MSM as this should lead to a reduction of actual evapotranspiration and thus an increase in
runoff, in particular at high soil saturation values (Eq. 17 in Müller Schmied et al., 2021).

In addition to SL-MSM, three other parameters are adjusted by the calibration in a way that water retention is
increased (improving correlation with both observed Q and TWSA), while at the same time a higher TWSA
variability results (decreasing or at least not improving fit to observed Q and TWSA). Both maximum wetland
(SW-WD) and lake depths (SW-LD) are increased by calibration, from 2 to 5.7 m in the case of wetlands and from
5 m to 8 m in the case of lakes, and the lake and wetland discharge coefficient SW-DC is adjusted to its minimum
value of 0.001/d. In contrast, the adjustment of the river roughness coefficient multiplier (SW-RRM) to 1.5, i.e.,
to half of the value in the uncalibrated model, leads to a doubling of flow velocity in the river as compared to the

standard value and thus lower water retention (reducing correlation with observed Q and TWSA), a higher variability of Q (improving the fit to observations) and a higher variability of TWSA (worsening the fit to observations). In addition, the net abstraction from groundwater is decreased by 80% (NA-GM = 0.2). Snow melt temperature SN-MT is lowered from the standard value of 0 °C to -2.6 °C with POC. Overall, most parameters are adjusted to increase the correlation between observed and simulated TWSA (except SW-RRM) and reduce mean

runoff (except EP-PTh). Unfortunately, the adjusted parameters increase TWSA variability (except SW-RRM), leading to an even stronger overestimation than the uncalibrated and standard calibrated variants (Table B2) and a worse underestimation of Q variability (Table B1).

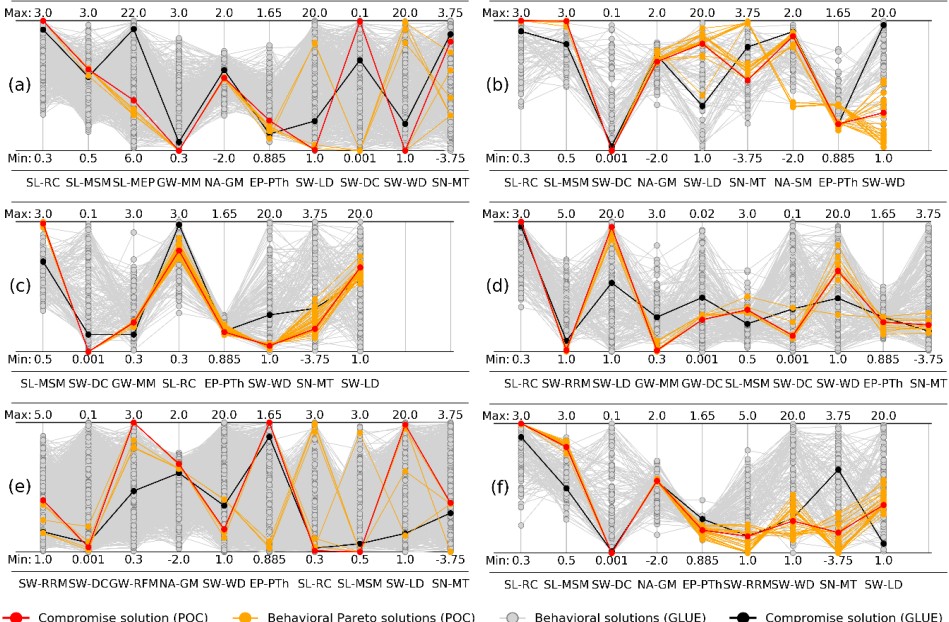

**Figure 5.** Parameter sets determined by POC and GLUE calibration approaches as depicted by parallel coordinate
plots for CDA units (a) Arkansas, (b) Missouri, (c) Upper MRB, (d) Ohio, (e) Lower MRB, and (f) MRB. The parameter abbreviations are given at the bottom of each plot, where the order was selected to show interesting relations between parameter values. The numbers at the top and bottom of the plots indicate the a-priori range of the calibration parameters listed in Table 2. The number of behavioral solutions is given in Table 6. GLUE
behavioral solutions are shown in greys, GLUE compromise solution in black, POC Pareto behavioral solutions in oranges and POC compromise solution in red.

*Parameter sets with optimal fit to Q or TWSA for POC.* Regarding trade-offs, the POC parameter set that leads to the best fit to observed Q is characterized by a higher SW-RRM (2.2 instead of 1.5 in the compromise parameter
set), a two-third reduction of SW-LD, a higher SN-MT, and a value of NA-GM of approximately 1. The latter shows that the net groundwater abstractions estimated by the water use models of WaterGAP lead to a good fit to





the monthly Q time series. In the POC parameter set leading to the best fit to observed TWSA, SL-MSM reaches 3 (the maximum value), while SW-WD attains a value of more than 12 m. This parameter set includes an SW-RRM value of only 1 (the lower bound, leading to a minimum flow velocity) and a slightly negative NA-GM. The latter parameter value means that the net water abstractions from groundwater, which are dominantly positive in the MRB (Fig. A1e), i.e. more water is withdrawn from the groundwater than recharged by return flows, are not only decreased but become mostly net groundwater recharge by the parameter adjustment. This could be caused, for example, by an original overestimation of the fraction of the total water abstraction that stems from groundwater and not surface water. Return flow from irrigation with surface water can lead to a net abstraction from groundwater that is negative, i.e., is an artificial groundwater recharge. However, it might also be caused by an underestimation of groundwater recharge, such that groundwater storage loss and the decrease of groundwater outflow to rivers by net groundwater abstractions would be overestimated if NA-GM was not adjusted from its standard value of 1.

*Parameter set of the GLUE compromise solution.* 6 out of the 9 parameters in the GLUE compromise solution are very similar to those of the POC compromise solution (Fig. 5f). The GLUE compromise solution has a slightly higher NSE$_Q$ but a considerably lower NSE$_{TWSA}$ (due to a lower correlation but a similar performance of variability) due to a lower soil moisture capacity and a very minimum lake water storage. In addition, the snow melt temperature is much higher.

*Behavioral Pareto-optimal POC parameter sets.* The 26 behavioral Pareto-optimal parameter sets derived by POC coincide in the four parameters SL-RC, SL-MSM, SW-DC, and NA-GM (Figs. 5f and 6, and Excel file in the supplement). The parameter values of the other five parameters diverge somewhat, indicating conflicts between a good fit to observed Q and TWSA. The fit to Q decreases and the fit to TWSA increases with decreasing EP-PTh, SN-MT, and SW-RRM and with increasing SW-WD. A negative correlation is visible between the values for SW-WD (wetland depth) and the values for SW-LD (lake depth) (see also Fig. S5f); this indicates that the same impact on Q and TWSA is achieved by either a large wetland depth or a large lake depth. The negative correlation between SW-WD and the three parameters EP-PTh, SW-RRM, and SN-MT is not easily interpretable (Fig. S5f).

*Behavioral GLUE parameter sets.* Behavioral GLUE parameter sets are much more diverse than behavioral Pareto-optimal parameter sets (Figs. 5f and 6). The GLUE parameter sets take into account, in an approximate manner, the uncertainty of performance indicators that stems from observation errors (Sections 3.2.2 and 3.4.2), in addition to the conflicting goals of achieving a good fit to observed Q and observed TWSA that is also reflected by the Pareto-optimal parameter sets. The 138 behavioral GLUE parameter sets, which all result in NSE values > 0.65, vary widely and for some parameters cover the whole parameter range (Figs. 5f and 6). In most behavioral sets, the SL-RC values are larger than 2, but there is even a set with a value below 1. SL-MSM ranges between 1 and 2.7, while the parameter value of the POC compromise solution is at the upper end of this range. Different from the Pareto-optimal POC solutions, SW-RRM values do not encompass very small values close to 1 but tend to be higher, mostly between 2 and 3 (Fig. 6). SN-MT as well as the three parameters related to lakes and wetlands, SW-DC, SW-LD, and SW-WD, are not constrained at all by the calibration (Fig. 5f and Fig. 6). Parameter correlations are very low, except negative correlations of EP-PTh with SL-RC, NA-GM and SW-DC (Fig. S5f).



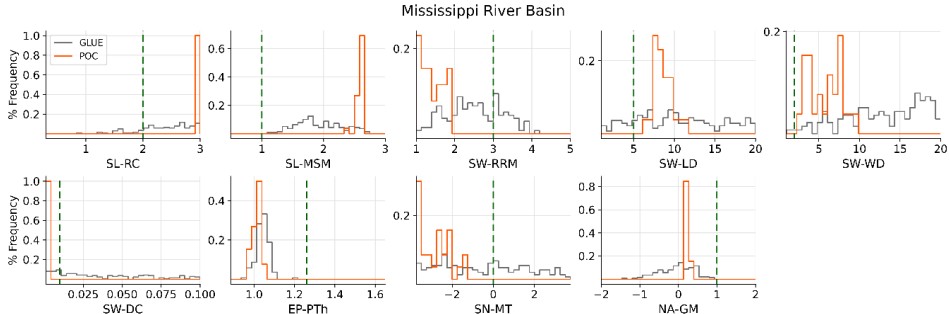

**Figure 6.** Histogram of parameter values in calibrated parameter sets according to POC and GLUE for the MRB (CDA unit VI). All behavioral parameter sets are considered for GLUE, while the smaller ensemble of behavioral Pareto-optimal parameter sets is shown for POC. The total number of parameters set for POC and GLUE is listed in Table 6. The y-axis shows the ratio of the number of parameter values in class to the total number of parameter sets, while the x-axis shows the a-priori parameter range listed in Table 2. The green dashed line indicates the parameter values of the uncalibrated WaterGAP model.

### 4.3.2 The five sub-basin CDA units

For all five sub-basins except the downstream Lower MRB (with SL-RC = 0.33), calibrated SL-RC is close to the maximum value of 3 in the POC compromise solution (Fig. 5). SL-MSM is at its lower bound in the Lower MRB, but increases maximum soil water storage in all other CDA units; the multiplier is almost at its maximum value of 3 for the Missouri River basin and the Upper MRB, about 2 for the Arkansas River basin and 1.3 for the Ohio River basin, which is the basin with the best performance of the uncalibrated model. In all CDA units but the Arkansas River basin, SW-LD reaches very high values between 10 and 20 m, and SW-WD is also higher than the uncalibrated values in all CDA units except Arkansas and Upper MRB. The SW-DC is at its minimum value in the Missouri River basin and the Upper MRB, close to its uncalibrated value in the Arkansas and Ohio River basins and in between in the Lower MRB. Calibrated SN-MT varies strongly among the CDA units. NA-GM is always below 1 to increase groundwater retention. The Lower MRB is the only CDA unit where optimal EP-PTh was high (1.65) while in all other CDA units, the calibrated value was close to 1.

Overall, there is a particularly high equifinality of parameter sets in the Lower MRB, with strong negative correlations between parameters of the Pareto-optimal POC solutions (Fig. S5e). Among the POC solutions in the Arkansas River basin, the parameters wetland depth (SW-WD) and surface water discharge coefficient SW-DC (Fig. 5a, compare POC compromise solution with POC behavioral solutions, and Fig. S5a) are so negatively correlated that the parameters alternatively take values at the opposite limits of the parameter ranges. A high value of maximum storage in surface water bodies has a similar effect on Q and TWSA dynamics as a low surface water discharge coefficient that keeps water in storage. Parameters may also show very strong correlations within a very small parameter space as in the case of EP-PTh and SW-WD in Upper MRB (Figs. 5c and S5c).





The GLUE behavioral parameter sets cover an even larger range in the Lower MRB and the Arkansas River
basin as compared to the MRB (Figs. 5 and S7). Correlations between parameters are generally low (Fig. S6),
except for high negative correlations between EP-PTh and SL-RC in the Missouri River basin and between EP-
PTh and SL-MSM in the Ohio River basin. However, low correlations between the calibrated parameters do not
indicate a low equifinality.

### 4.4 Added value of spatially more resolved CDA units

An important decision in model parameter estimation is the choice of CDA units, i.e., the selection of the group
of grid cells for which calibration parameters are assumed to be the same. A higher number of CDA units within
the same geographic domain leads to the adjustment of more parameters, causes a higher computational effort and
is expected to lead to an improved representation of reality. We performed two analyses to evaluate the added
value of dividing the MRB into five sub-basin CDA units.

In the first analysis, we used the compromise solutions obtained for the CDA unit VI (MRB), where the same
calibration parameter values are assigned to all grid cells in the whole MRB, to compute Q and TWSA for each of
the five sub-basin CDA units. Model performance of this calibration variant ("whole basin calibration") is
compared to the performance that is achieved in the sub-basins if each sub-basin is calibrated individually, i.e., if
five CDA units are used to cover the whole MRB. Analysis for both the calibration period (Table 7) and the
validation period (Table S1) clearly shows the added value of distinguishing five sub-basin CDA units (calibration
variant "sub-basin calibration") as overall model performance improves in each of the five sub-basins as compared
to the calibration variant "whole basin calibration. Due to the specific search algorithm, performance gains are
more pronounced with POC than with GLUE. Performance gains are very high in the case of EnCDA due to the
poor performance of the whole-basin calibration. Considering POC and regarding Q, the added value of more
CDA units is highest for the Missouri River basin, followed by the Arkansas River basin and the Upper MRB.
However, for these three sub-basins, there is no added value regarding TWSA. In the always best performing Ohio
River basin, there is a small added value for both Q and TWSA, while in the downstream Lower MRB, where Q
is dominated by inflow from the four upstream sub-basins, Q performance remains essentially unchanged while
TWSA performance improves with more CDA units.





**Table 7.** Comparison of model performance in the five sub-basins of the MRB between the calibration of MRB as a whole (one CDA unit VI) and calibration of the individual sub-basins (five CDA units I–V). In addition, the performance of the model with standard calibration of 77 CDA units but adjusting only up to three parameters based on observed mean annual Q is shown. Model performance is indicated by $NSE_Q$ and $NSE_{TWSA}$ during the calibration period 2003-2012 as achieved by the compromise solutions of the three calibration approaches POC, GLUE, and EnCDA. The sub-basin calibration NSE values are identical to those in Table 3, except for MRB (see footnote 1).

| | $NSE_Q/NSE_{TWSA}$ | | | | | |
| | Arkansas | Missouri | Upper MRB | Ohio | Lower MRB | MRB |
| --- | --- | --- | --- | --- | --- | --- |
| POC: whole basin calibration | 0.65/0.83 | 0.38/0.71 | 0.57/0.48 | 0.82/0.77 | 0.83/0.69 | 0.83/0.73 |
| POC: sub-basin calibration | 0.74/0.85 | 0.73/0.71 | 0.67/0.48 | 0.87/0.86 | 0.81/0.90 | 0.81/0.79[1] |
| GLUE: whole basin calibration | 0.67/0.84 | 0.49/0.64 | 0.64/0.33 | 0.85/0.75 | 0.85/0.74 | 0.85/0.65 |
| GLUE: sub-basin calibration | 0.69/0.83 | 0.65/0.71 | 0.61/0.46 | 0.86/0.84 | 0.77/0.89 | 0.77/0.77[1] |
| EnCDA: whole basin calibration | -0.41/0.60 | -1.69/0.51 | 0.36/0.26 | 0.57/0.55 | 0.51/0.60 | 0.51/0.19 |
| EnCDA: sub-basin basin calibration | 0.59/0.84 | 0.62/0.65 | 0.68/0.60 | 0.79/0.91 | 0.83/0.88 | 0.83/-0.31[1] |

[1]based on Q at Vicksburg and TSWA averaged over the whole MRB computed by a WaterGAP run, in which the calibration parameters in the five sub-basins (CDA units I-V) were set to their respective compromise solution values.

An evaluation of the performance regarding the mean of TWSA over the entire MRB using the individual parameter sets of the five sub-basin CDA units shows a small added value of using sub-basin CDA units in the case of POC and GLUE while in the case of EnCDA the already poor fit to TWSA in the whole basin variant is further degraded (column MRB in Table 7). However, EnCDA estimation of Q at Vicksburg is much improved with five CDA units and reaches the high values of GLUE and POC, both of which show a slight degradation of the Q simulation at Vicksburg as compared to the whole-basin calibration.

In the second analysis, we evaluated the ability of the different calibration variants to simulate Q at six Q gauging stations that were not used for model calibration in this study; three are located in the Missouri River basin and three in the Ohio River basin (Fig. 2). Differences between the stations are larger than between the calibration approaches. Good $NSE_Q$ values are only achieved at two stations, Mt. Carmel and Louisville in the Ohio River basin. The best performance at Mt. Carmel is achieved with the whole-basin GLUE approach (NSE = 0.77), while the POC sub-basin approach achieves the optimal performance at Louisville, with NSE = 0.91. (Table 8). Sub-basin calibration strongly improves NSE as compared to whole basin calibration in the case of the Platte River station at Louisville for both POC and GLUE, by reducing the bias (RBias) but decreasing correlation (CC) and the fit to observed Q variability (RVar) (Table 9) but not during the validation period (Table S2). There is some added value of sub-basin calibration regarding Q simulation at the Louisville station on the Ohio River for both the calibration and the validation period. At this station, sub-basin calibration also leads to higher NSE values as compared to the standard calibration and uncalibrated WaterGAP variants. For the other four stations, however, sub-basin calibration leads to worse performance than whole-basin calibration during the calibration period. For





the station on the Cumberland, which is not a calibration station in the standard calibration, the standard calibration even leads to a better performance than all the ensemble-based calibrations for both the calibration and validation

period. At the Bismarck station on the Missouri River, where model performance is similarly poor as on the Cumberland, even the uncalibrated WaterGAP variant performs better or similar to the calibrated variants due to the highest correlation. During the validation period, the performance of all three calibration approaches becomes very low at the three stations in the Missouri River basin (Table S2), while it remains constant or even improves for the three stations in the Ohio River basin. No calibration approach performs consistently better than any other

approach; performance rather depends on the period and the station. Overall, calibration using Q observations on downstream stations only leads to apparently random changes in Q simulation at upstream stations that have not been used in the calibration.

**Table 8.** Comparison of model performance at the six Q validation stations in the Missouri and Ohio sub-basins

of the MRB (Fig. 2) between the calibration of MRB as a whole (CDA unit VI) or calibration of the individual sub-basins (CDA units I–V). Model performance is indicated by $NSE_Q$ and the three KGE components during the calibration period 2003-2012 as achieved by compromise solutions of the three calibration approaches POC, GLUE, and EnCDA. In the case of CDA, the performance metrics for the 2003-2012 CDA run are shown and not of a run with the parameter set of December 2012. The best-performing calibration variant for each station is

shown in bold. In addition, performances of the standard and uncalibrated WaterGAP model variants are shown.

| | $NSE_Q$/CC/RBias/RVar | | | | | |
|---|---|---|---|---|---|---|
| | Missouri near Landusky | Missouri at Bismarck[1] | Platte at Louisville[1] | Wabash at Mt Carmel[1] | Ohio at Louisville | Cumberland at Nashville |
| POC: whole basin calibration | 0.30/0.73/ 0.67/1.10 | -0.04/0.38/ 0.68/0.29 | -0.56/0.79/ 1.61/0.95 | 0.74/**0.91**/ 1.24/0.74 | 0.78/0.91/ 1.11/0.68 | 0.37/**0.86**/ 1.59/0.46 |
| POC: sub-basin calibration | 0.23/0.78/ 0.58/1.45 | -0.38/0.41/ 0.41/0.51 | 0.54/**0.83**/ 0.96/1.26 | 0.65/0.87/ 1.24/0.81 | **0.91/0.96**/ 1.08/0.84 | 0.32/0.84/ 1.62/0.54 |
| GLUE: whole basin calibration | **0.50**/0.80/ 0.80/1.30 | -0.03/0.32/ 0.69/0.40 | -0.55/0.76/ 1.57/**0.98** | **0.77/0.91**/ 1.20/0.80 | 0.83/0.92/ 1.05/0.80 | 0.42/0.85/ 1.53/0.49 |
| GLUE: sub-basin calibration | 0.41/0.77/ 0.72/1.25 | -0.15/0.39/ 0.56/0.41 | **0.58**/0.80/ **1.00**/1.05 | 0.67/0.87/ 1.25/0.76 | 0.87/0.94/ 1.07/0.81 | 0.31/0.84/ 1.62/0.48 |
| EnCDA: whole basin calibration | 0.20/0.56/ **1.00**/0.89 | -1.09/-0.32/ 1.42/0.45 | -8.98/0.57/ 3.15/0.60 | 0.66/0.85/ **1.09**/0.58 | 0.41/0.71/ 0.86/0.55 | 0.24/0.59/ **1.30**/0.36 |
| EnCDA: sub-basin calibration | 0.48/0.74/ 1.12/0.83 | 0.46/0.89/ 1.47/0.53 | -1.8/0.42/ 1.79/0.77 | 0.59/0.89/ 1.38/0.67 | 0.65/0.82/ 1.00/0.64 | 0.13/0.69/ 1.58/0.51 |
| Standard calibration | 0.37/**0.81**/ 1.19/**1.06** | 0.36/0.64/ **1.03**/0.40 | 0.04/0.71/ 0.99/1.40 | 0.70/0.87/ **1.09**/0.97 | 0.78/0.89/ 1.03/0.79 | **0.52**/0.84/ 1.42/**0.56** |
| Uncalibrated | 0.40/0.77/ 1.11/1.08 | **0.47/0.82**/ 1.38/**0.62** | -6.30/0.69/ 2.26/1.21 | 0.69/0.89/ 1.20/0.88 | 0.78/0.88/ **0.98/0.86** | 0.40/0.84/ 1.54/0.48 |

[1]Calibration station of standard calibration





**5 Discussion and conclusions**

In this study, three ensemble-based methods for estimating optimal and behavioral parameter sets for the global hydrological model WaterGAP using observations of streamflow (Q) and total water storage anomaly (TWSA) are presented and evaluated for the Mississippi River basin. Two spatial calibration set-ups were tested. The whole basin down to the Q observation station Vicksburg was either treated as one CDA unit with spatially uniform calibration parameters, or the basin was divided into five sub-basins that were treated as individual CDA units. In each case, observations are monthly time series of Q at the most downstream grid cell of each CDA unit and of monthly time series of TWSA spatially averaged over the CDA unit (POC and GLUE) or 4° boxes (EnCDA) during the period 2003-2012. Based on a sensitivity analysis for each CDA unit individually, 8-10 calibration parameters were adjusted in each CDA unit. For each CDA unit and calibration approach, an "optimal" compromise parameter set could be determined. A method for taking into account the uncertainty of Q and TWSA observations in the selection of behavioral ensemble members was developed. This method is based on the GLUE ensemble, and the derived performance thresholds, which define which parameter sets can be regarded as acceptable and informative given the observation uncertainties, can be applied to all three calibration approaches.

**5.1 Advantages and disadvantages of the three multi-variable calibration approaches**

The applicability of the EnCDA approach is strongly limited by its high computational burden which restricted the number of ensemble members in our study to 32, as compared to 20,000 in the case of POC and GLUE. This very small ensemble is likely the reason for the generally lower performance of the EnCDA results as compared to POC and GLUE both during the calibration and validation periods. In the case of EnCDA, calibration and data assimilation using only one CDA unit for the whole MRB resulted in a worse performance regarding both Q and TWSA than that of the uncalibrated WaterGAP during the calibration period 2003-2012. Regarding the five sub-basin CDA units, performance for all but one CDA unit was worse than that of POC and GLUE even though not only parameters but also water storages are adjusted in EnCDA. Performance was, however, improved over the standard and uncalibrated model variants for 4 out of 6 CDA units, in particular regarding TWSA. Q simulation by EnCDA during the calibration period might be improved by using log Q instead of Q (Clark et al., 2008; Paiva et al., 2013), and in the case of dry world regions, by censoring no flow observations (Wang et al., 2020). Note that we define performance generally in terms of NSE, while EnCDA unlike POC and GLUE does not optimize NSE but rather the RMS of model-observation differences.

During the validation period 2013-2016, where EnCDA uses the 32 parameter sets obtained at the end of the calibration period (December 2012) to compute Q and TWSA without any update of water storages, TWSA and to a lesser extent Q "drifted off" from the observations, resulting in very poor fits. This may be explained by the fact that the monthly parameter updates in EnCDA absorb model misrepresentations that generate seasonally varying errors such that the December 2012 parameter sets were not able to lead to a reasonable simulation during the whole four years of the validation period. Unlike EnCDA, POC and GLUE decide on optimal parameter sets based on the overall behavior during the calibration period, based on all simulated and observed calibration variables.



POC and GLUE show similar performances. However, the search algorithm in POC leads to better
identification of (Pareto-)optimal parameter sets than GLUE, for the same computational burden, even though the
difference between the time series of Q and TWSA computed by POC and GLUE is small compared to the
discrepancies to the observations (Figs. 4 and S4). Therefore, POC is preferable to GLUE if only one optimal
parameter set, e.g., the compromise parameters set, or Pareto-optimal parameter sets are to be identified.
Application of GLUE is required for the identification of behavioral parameter sets and thus an estimation of the
uncertainty of simulated WaterGAP output.

**5.2 Added value of multi-variable calibration as compared to the standard WaterGAP calibration for identifying one "optimal" parameter set**

The compromise parameter sets identified by multi-variable calibration result in better simulations of both Q and
TWSA during the calibration period as compared to the standard WaterGAP for all six CDA units, except for
EnCDA in the case of the whole MRB (Table 3). However, the added value of any calibration is very low in the
humid and hilly Ohio basin where the performance of the uncalibrated model is already good. As can be expected,
the improvement of TWSA simulations is more pronounced than the improvement of Q. Higher $NSE_Q$ values are
mostly caused by improved correlation, while Q variability is still underestimated in all CDA units, and in three
CDA units even more strongly than by the standard and uncalibrated WaterGAP variants. In two CDA units, the
mean Q is overestimated by more than 10% (Table B1). The much higher $NSE_{TWSA}$ values of the compromise
solutions as compared to standard WaterGAP are also mainly caused by much-improved correlations, with
improved or worsened TWSA variabilities depending on the CDA unit (Table B2). The analysis of model
performance at the same observation locations for the validation period 2013-2016 confirms the added value of
POC and GLUE.

However, visual inspection of the hydrographs for both the calibration and validation periods reveals that the
fit to observations can only be improved slightly by the multi-variable calibration. An exception is the simulation
of TWSA in the Lower MRB, which is affected by intensive irrigation in the Mississippi Embayment. There,
standard WaterGAP simulates a declining TWSA trend due to groundwater depletion, which does not occur
anymore with the three multi-variable approaches that make use of observed TWSA (Fig. S4f). The
underestimation of seasonal low flows in all six CDA units remains after calibration, not only in the compromise
solutions (Figs. 4 and S4) but also in the POC and GLUE runs with the highest $NSE_Q$.

An advantage of POC and GLUE over the standard WaterGAP calibration is that by adjusting 8-10 parameters
per CDA unit, it is possible to achieve higher NSE values for Q without having to use any correction factors. In
the standard calibration, both areal and station correction factors are necessary for many CDA units in the western
part of the MRB to reduce simulated mean annual Q to observed values (Calibration status CS3 and CS4 in Fig.
S3a). It is particularly beneficial that station correction factors (Fig. S3d) are avoided by the new calibration
approach as they lead to abrupt changes in Q and destroy mass conservation (Müller Schmied et al., 2021). Even
by adjusting only 9 parameters homogeneously in the whole MRB using monthly time series of observed Q and
TWSA, improved model performance is achieved, compared to adjusting more than 100 parameters in 77 CDA
units in the standard WaterGAP calibration, except for the Ohio River basin and Q in the Missouri basin. This



statement only relates to the Q observations considered in this study, not to the Q at all 77 standard calibration stations.

There appears to be almost no added value of the multi-variable calibration approaches for the simulation of Q at upstream locations within the calibrated CDA unit where Q observations were not used for calibration (Table

8). This may be due to the very large and heterogeneous CDA units; the CDA unit MRB covers almost 3 million km$^2$, while the largest sub-basin CDA, the Missouri River basin covers 1.35 million km$^2$ and the smallest, the lower MRB, still 0.25 million km$^2$.

### 5.3 Estimation of WaterGAP output uncertainty

Both GLUE and EnCDA aim at estimating model output uncertainty, but the small ensemble size of EnCDA

prevents a meaningful estimation. The presented application of the GLUE approach aims at quantifying the uncertainty of WaterGAP output due to parameter uncertainty, taking into account the uncertainty of model output observations. The estimated uncertainty bands underestimate the model uncertainty. The identified 72 to 1517 behavioral parameter sets (Table 4), out of a total of 20,000, result in uncertainty bands that are small compared to the variability of Q and TWSA, in particular seasonal variability (Figs. 4 and S4). Unfortunately, only 46-72%

of the monthly Q estimates of the GLUE behavioral model runs fall into the uncertainty band of observed Q, depending on the CDA unit (Table 5). With 59-95%, TWSA coverage is higher, except in the Missouri River basin.

Low coverage values indicate that the model suffers from errors in either model input or model structure. An explanation for the overestimated low flows might be that WaterGAP, like most hydrological models, is not able

to simulate water loss from the river into the groundwater, while a recent study has found strong indications for extensive losing river conditions in the MRB (Jasechko et al., 2021). Further model uncertainties that appear particularly relevant for the limited performance of WaterGAP in the different CDA units are related to the modeling of man-made reservoirs, which may be particularly relevant for the Missouri River basin, and the poor specification of location and extent of small wetlands (Prairie potholes) in the Missouri River basin and the Upper

MRB.

The relatively thin uncertainty bands indicate equifinality of the very diverse behavioral parameter sets (Fig. 5) for the study period. The widths of the uncertainty bands of POC and GLUE do not change appreciably between the calibration and the validation period (Figs. 4 and S4), which indicates that calibrated parameter sets are transferable between the two periods. The exceptions are the TWSA uncertainty bands in the Arkansas river basin

(Fig. 4) and Lower MRB (Fig. S4) which, for unknown reasons, are wider in the validation period, indicating that parameter sets that lead to similar model output in the calibration period might result in more discrepant model output. under changed climatic conditions.

### 5.4 Trade-offs between optimal simulation of Q and TWSA

Trade-offs between the optimal simulation of Q and TWSA are relevant in all CDA units. POC trade-offs are only

slightly smaller than GLUE trade-offs when considering the NSE values achieved by parameter sets that lead to



either maximum $NSE_Q$ or maximum $NSE_{TWSA}$ (Table 3). There are particularly large trade-offs between a good fit to Q and TWSA in two sub-basin CDA units with many surface water bodies, i.e., in the Missouri River basin (reservoirs, wetlands and lakes) and the Upper MRB (wetlands and lakes) (Fig. A1) and, accordingly, also in the CDA unit MRB. In the Missouri River basin, for example, the POC parameter set resulting in an optimal fit to

observed Q has an $NSE_Q$ of 0.83 but an $NSE_{TWSA}$ of only 0.5, while the POC parameter set resulting in an optimal fit to observed TWSA improves $NSE_{TWSA}$ to 0.81 but degrades $NSE_Q$ to the very poor value of -0.82 (Table 3). We suspect that the poor knowledge of the location and extent of small wetlands and the difficulty of simulating the operation of man-made reservoirs (without adjustment of parameters) are the main reasons for the strong trade-offs. In most CDA units, an optimal TWSA fit leads to a strong overestimation of mean Q and an even stronger

underestimation of Q variability (Table B1), while a good fit to Q leads to an overestimation of TWSA variability, in different degrees depending on the CDA unit (Table B2). We speculate that this trade-off cannot be explained by potential errors of the used GRACE TWSA time series due to leakage effects, the impacts of which are not included in the values of GRACE TWSA used in this analysis (see Section 3.2.2). For the Lower MRB, the multiplicative leakage re-scaling factor of 1.41 (see Section 3.2.2) matches the overestimation of TWSA variability

(RVar = 1.42 for the POC parameter set with the best fit to Q, Table B2), but this may be by chance. Besides, the estimated re-scaling factor may be biased by an overestimated negative TWSA trend in the standard WaterGAP run that was used to compute it.

Much smaller trade-offs between the optimal fits to observed Q and TWSA were found with another hydrological model in a calibration study for 83 European river basins, where both Q and TWSA observations

were used for adjusting up to 53 parameters in a basin-specific manner (Rakovec et al., 2016). When TWSA was considered in addition to Q in the calibration objective, the correlation of observed and simulated Q decreased slightly while bias and variability remained almost unchanged. However, TWSA correlations that were achieved by calibration were extremely low, with a median r of 0.56 if only Q observations were used in the calibration, increasing to only 0.67 if in addition TWSA observations were included. In our study, TWSA correlations are

much higher; for the calibration period, they vary between the six CDA units from 0.80-0.95 in the case of the POC compromise solution. Even the uncalibrated WaterGAP variant leads to CC values in the range of 0.76-0.93.

Accessible Q observation data are rare in many parts of the globe, while GRACE TWSA observations cover the whole globe and are freely available. The use of GRACE TWSA observations strongly increased the fit of simulated to observed TWSA during the calibration and validation periods. But is the calibration of WaterGAP

against TWSA observations only beneficial for the estimating Q? This question can be answered by analyzing the performance metrics of the GLUE a-priori ensemble of 20,000 parameter sets. For both the calibration and the validation period, Q simulation degrades in three of the six CDA units in the variants "highest $NSE_{TWSA}$" as compared to the uncalibrated WaterGAP. This is the case in the Upper MRB where WaterGAP struggles with uncertain information regarding the location and extent of small wetlands, in the Ohio River basin in which already

the uncalibrated WaterGAP variant simulates Q well and in the MRB (Tables 3 and 6). In the Missouri River basin, Q simulation by the uncalibrated model is very poor and remains so after calibration against GRACE TWSA only. In the Murray-Darling basin, EnCDA using GRACE TWSA only resulted in Q overestimation (Schumacher et al., 2018). Thus, a calibration against GRACE TWSA only may degrade or not the Q simulation as compared to an uncalibrated model run, and it is difficult to estimate where such degradation could occur. Further studies are



needed to understand under which circumstances calibration against GRACE TWSA does not degrade the
        simulation of Q.

### 5.5 Added value of sub-basin CDAs instead of one basin CDA

        Considering Q performance at the outlet of the five sub-basin CDA unit and the aggregated TWSA, overall model
        performance is somewhat improved if calibration is done individually for the five CDA units instead of adjusting
parameters homogeneously over the whole MRB, for both the calibration (Table 7) and the validation periods
        (Table S1). The added value is smaller for the validation period. However, in three CDA units, TWSA performance
        during the calibration period is not affected by the higher number of CDA units, while Q at the calibration stations
        was improved if the station was used in the calibration. In addition, Q performance at gauging stations inside the
        two sub-basin CDA units was not improved by sub-basin calibration (Table 8 and Table S2). Q performance at
these gauging stations appears to be unrelated to the type of calibration done (including no calibration) as the best-
        performing calibration approach varies randomly among CDA units and periods. Therefore, to increase the quality
        of Q simulations with WaterGAP, we suggest using CDA units that are smaller than the Mississippi River sub-
        basins selected for this study, i.e., smaller than about 400,000 km². This is also supported by the study of Mizukami
        et al. (2017) who selected 531 CDA units for the continental US. Alternatively, the joint calibration against
multiple Q observations within a CDA unit should be tested (Xie et al., 2012; Wanders et al., 2014).

### 5.6 Characteristics of identified (Pareto-)optimal and behavioral parameter sets

        In (Pareto)-optimal parameter sets, the optimized runoff coefficient SL-RC obtains values very close to its upper
        bound in all CDA units except the downstream Lower MRB where Q is dominated not by runoff generation within
        the CDA unit but by inflows from upstream CDA units. High SL-RC values, which tend to decrease runoff, are
also obtained by the standard WaterGAP calibration (Fig. S3). Further reduction of runoff is achieved in this study,
        except for the downstream Lower MRB, by increasing maximum cell-specific soil water storage by multiplication
        with optimized SL-MSM values that are larger than 1, ranging between 1.3 for the best-simulated Ohio River basin
        to almost 3 for the Missouri River basin and the Upper MRB. A larger maximum soil water storage leads to
        decreased soil saturation and lower runoff, and at the same time to higher variability of soil water storage and thus
TWSA. It is surprising that EP-PTh, a factor in the equation of potential evapotranspiration, is reduced in all CDA
        units (except in the Lower MRB) from its standard value of 1.26 to values around 1, which leads to reduced actual
        evapotranspiration and thus increased runoff. The multipliers adjusting grid cell values of human net water
        abstraction from groundwater (adjusted in four CDA units) tend to be less than zero, indicating an overestimation
        of net groundwater abstractions by the standard model variant. Water abstraction from surface water bodies
(adjusted only in the Missouri River basin) might be underestimated. The optimal values of the other calibration
        parameters can differ strongly between POC and GLUE compromise solutions or between the Pareto-optimal POC
        solutions (Fig. 5). The correlations between calibration parameters can be high and differ strongly between the
        CDA units; general patterns cannot be seen (Fig. S5). For example, SL-RC can correlate positively or negatively
        with SL-MSM and EP-PTh.



Regarding the behavioral POC parameter sets, the better the fit to TWSA is, the higher the maximum wetland depth SW-WD, ranging between 1.5 m and 11 m. The same is true for other parameter values that increase water storage capacities, such as SL-MSM, SW-WD, or SW-LD.

The ranges of most parameters in the behavioral GLUE parameter sets, which take into account the impact of observation uncertainty on optimized parameter sets, are only slightly narrower than the a-priori parameter ranges
(Figs. 5, 6 and S7). This is the case even though behavioral parameter sets are only a very small fraction of 0.04% - 0.76% of the 20,000 a-priori parameter sets of GLUE. We found larger equifinality in TWSA simulation than in Q simulations except for the downstream Lower MRB where simulated Q is dominated by the inflow from upstream as quantified by the compromise parameters sets of the four upstream CDA units (Excel file in supplement). That TWSA observations constrain parameter sets less than Q observations was also discovered in
the multi-variable calibration study for ten large basins in Sub-Sahara Africa by Xie et al. (2012).

Likely due to the fixed and dominant inflow from upstream, there was hardly any narrowing of the a-priori parameter range for the downstream Lower MRB (Figs. 5 and S7). EP-PTh is the only parameter whose distribution shows a peak (except for the Lower MRB, Figs. 6 and S7). Small peaks are often seen for SL-MSM and the net abstraction multipliers. SL-RC mostly shows a low frequency of values below 1 and increasing
frequencies towards the upper parameter bound. Uniform distributions without any significant narrowing are seen for SN-MT, SW-DC, SW-WD, and SW-LD. Parameter correlations among the behavioral parameter sets are mostly low, except for negative correlations that exist, depending on the CDA unit, between EP-PTh and parameters such as SL-RC, SL-MSM, and SN-MT (Fig. S6).

Multi-variable calibration did not lead to improved identifiability of parameters, i.e., the determination of a
small range of suitable parameter values, except for the three parameters SL-RC, SL-MSM and EP-PTh. This makes the application of the "optimal" compromise parameters set derived by POC problematic for estimating, e.g., groundwater recharge, groundwater abstractions and surface water abstractions. Examples are the multipliers for net groundwater and surface water abstractions in the Missouri basin, where the POC compromise solution suggests that net groundwater abstractions are 25% lower and net surface water abstractions are 50% higher than
estimated without parameter adjustment (Fig. 5b and Excel file in supplement). Even the behavioral Pareto-optimal parameter sets, which are obtained by optimizing the fit to observations that are assumed to be error-free, include severe decreases but also slight increases of net groundwater abstraction as compared to the standard value as well as strong increases of net surface water abstractions but also a reversal from net abstractions to net additions of water to surface water bodies by large return flows from groundwater-sourced water to surface water bodies (Fig.
5b and Excel file in supplement). In the Arkansas basin, the POC compromise solution suggests a strong decrease of both groundwater recharge and net abstractions from groundwater, to 30% and 24% of the standard values, respectively, but very similar performances regarding the assumely error-free observations can be obtained if both values are decreased much less or even if groundwater recharge is increased (Fig. 5a and Excel file in supplement). The remaining equifinality of the parameter sets of our study even with using two different
observation variables is in accordance with the results of a calibration study for flood design in Sweden (Harlin and Kung, 1992). In that study, a large number of sets of twelve parameters were identified by model calibration using a Monte-Carlo approach, and, like in our study, it was for most parameters not the value of the individual


parameter that determined if the simulation of Q was behavioral but the combination of the parameter values within each parameter set.

### 5.7 Outlook

Based on this pilot study, we suggest that the multi-variable POC approach, combined with the described sensitivity analysis, is best suited for estimating a CDA unit-specific optimal parameter set for the global hydrological model WaterGAP that leads to a better fit to observations of both Q and TWSA than the simple standard WaterGAP calibration approach. Such basin-specific POC compromise parameter sets can then be used to simulate the best estimate of past and future water flows and storages, in particular, if various future scenarios, e.g., driven by the output of multiple climate models, or hydrological seasonal ensemble forecasts are computed by WaterGAP. Multi-variable model calibration against both Q and TWSA allowed constraining three model parameters while a large range of values of all other calibration parameters can lead to equally good fits to the observations, even if the uncertainty of observations is neglected, as in the case of Pareto-optimal POC parameter sets. We suggest that the parameter interdependence of Pareto-optimal parameter sets should be analyzed, The identified POC compromise parameter set should be applied with caution.

As we found that Q can be simulated reasonably well only at locations where Q observations have been used in the calibration, the selection of rather small CDA units that are similar in size to the calibration units in the standard WaterGAP calibration will be pursued for global-scale multi-variable calibration. Optimal ways of addressing GRACE TWSA leakage errors need to be identified to increase the commensurability of simulated and observed TWSA. Additional observation variables such as snow cover and water storage variations in lakes and man-made reservoirs will be taken into account to further reduce equifinality. Unfortunately, no information on groundwater levels is available at the global scale, which is likely required to constrain the parameters related to surface water and groundwater abstraction (Hosseini-Moghari et al., 2020).

The study results show that, currently, only the GLUE approach enables us to compute, with WaterGAP, values of water flows and storages that are informed by Q and TWSA observations and their uncertainties. The derived model output uncertainty bands approximately represent the impact of equifinality of parameter sets on model output and take into account the uncertainty of the observations. Such model output uncertainty bands can be readily computed by the presented methods but will underestimate the total model output uncertainty significantly, as they do not include the impact of model input and structure uncertainties. Likely because of the very small ensemble size that was feasible in EnCDA due to its severe computational burden, parameter sets and model output uncertainties could not be estimated by EnCDA in this study, neither for the calibration period nor for projections for periods without observation data. However, the EnCDA approach has the potential to lead to improved parameter estimates or model structure by, e.g., detecting a seasonal variation of model parameters. Further studies should investigate the feasibility of including NSE as an optimization metric in the EnCDA approach.

We recommend including, in future freshwater-related climate change impact studies, a behavioral ensemble of parameter sets as determined by the GLUE approach even though this will require a significant computational effort. This should reduce the underestimation of modeling uncertainty by traditional multi-model studies. As shown in the multi-model/multi-parameter study for the Colorado River basin by Mendoza et al.


(2016), parameter sets with a similar performance during the calibration period may provide very different projections of climate change hazards, and the impact of parameter uncertainty is similar to the impact of hydrological model selection.

**Appendix A: Surface water bodies and human water abstractions in the CDA units of the Mississippi River basin**

To understand the sensitivity of model output to parameters, the spatial distribution of storages and flows that they affect is required. Water balances of reservoirs (Fig. A1a) are not directly impacted by the calibration parameters in Table 2, while lake dynamics (Fig. A1b) are directly impacted by active lake depth (SW-LD) and wetlands (Fig. A1c, d) by active wetland depth (SW-WD). Please note that knowledge about the wetlands in the northern parts of the CDA units Missouri River basin and the Upper MRB as well as in the southern part of the Lower MRB is

restricted to the information in these areas that generally 25-50% of the land area are covered by wetlands in the wet season. In WaterGAP, this information was translated into a maximum extent of local wetlands of 35%. The surface water discharge coefficient (SW-DC) affects both lakes and wetlands. Potential net abstractions from groundwater (Fig. A1e) and surface water (Fig. A1f) are simulated with a monthly time step for each grid cell, and multipliers for each of them (NA-GM and NA-SM) affect model output differently in the various CDA units due

to different net abstraction patterns.



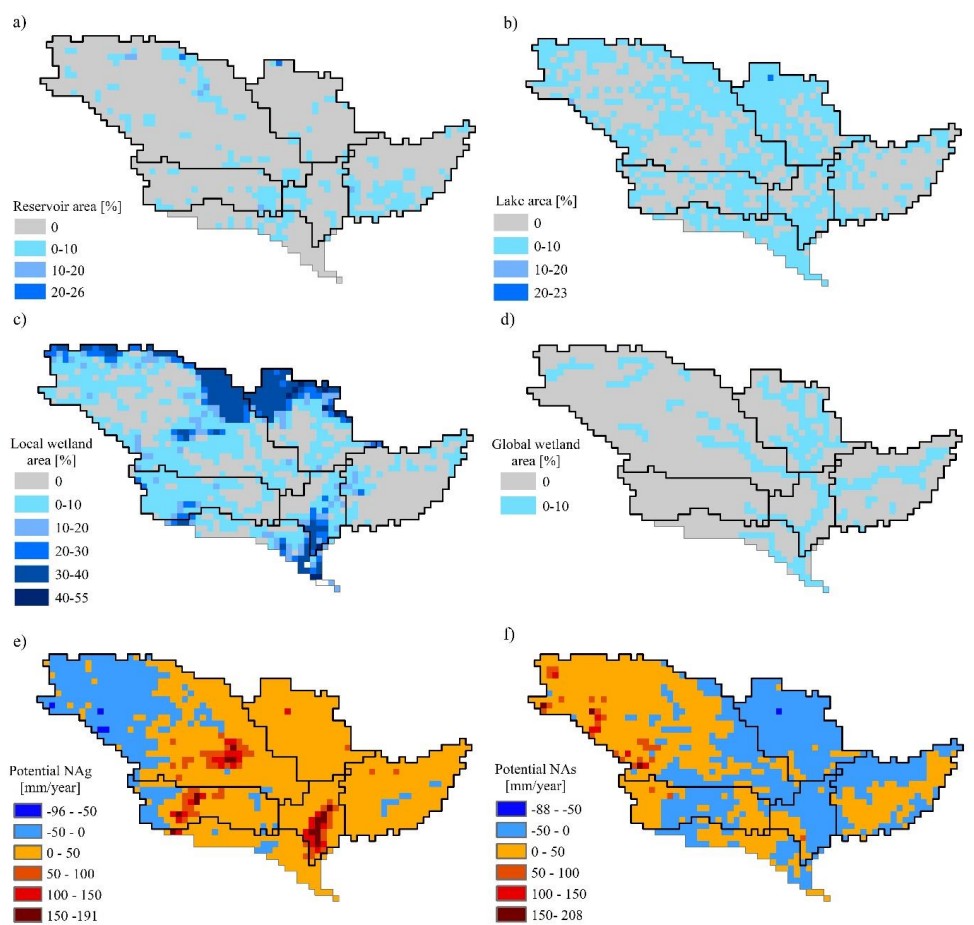

**Figure A1**. Man-made reservoirs (a), lakes (b), local wetlands (c), global wetlands (d) as well as potential net abstractions from groundwater (e) and from surface water (f) in the CDA units of MRB, as taken into account in WaterGAP. Maximum areal extents of the surface bodies in percent of the 0.5° cell area are shown, while potential net abstractions in mm/yr are provided for the period 2003-2012.





**Appendix B Performance of different calibration methods during the calibration period 2003-2012: Components of the KGE performance metric for both Q and TWSA**

**Table B1.** KGE components of model runs of Table 3 regarding Q for the calibration period 2003-2012.

| | CC/RBias/RVar | | | | | |
| --- | --- | --- | --- | --- | --- | --- |
| | Arkansas | Missouri | Upper MRB | Ohio | Lower MRB | MRB |
| POC: highest $NSE_Q$ | 0.90/1.18/0.59 | 0.92/1.08/0.80 | 0.91/1.03/0.82 | 0.95/1.07/0.89 | 0.95/1.04/0.88 | 0.95/1.01/0.88 |
| POC: highest $NSE_{TWSA}$ | 0.91/1.42/0.52 | 0.87/1.73/0.61 | 0.84/1.47/0.53 | 0.95/1.27/0.71 | 0.94/1.07/0.84 | 0.92/1.34/0.62 |
| POC: compromise | 0.90/1.18/0.59 | 0.86/1.06/0.80 | 0.89/1.20/0.66 | 0.95/1.11/0.83 | 0.95/1.06/0.82 | 0.94/1.09/0.78 |
| GLUE: highest $NSE_Q$ | 0.90/1.21/0.52 | 0.91/1.14/0.71 | 0.89/1.01/0.82 | 0.94/1.07/0.88 | 0.93/1.02/0.86 | 0.94/1.02/0.82 |
| GLUE: highest $NSE_{TWSA}$ | 0.91/1.74/0.38 | 0.88/1.70/0.60 | 0.84/1.51/0.43 | 0.95/1.30/0.63 | 0.94/1.12/0.75 | 0.94/1.33/0.64 |
| GLUE: compromise | 0.90/1.31/0.55 | 0.86/1.18/0.68 | 0.88/1.23/0.57 | 0.94/1.11/0.79 | 0.93/1.07/0.79 | 0.93/1.05/0.84 |
| EnCDA: highest $NSE_Q$ | 0.81/1.18/0.76 | 0.86/1.14/0.76 | 0.86/1.11/0.77 | 0.92/1.13/0.69 | 0.92/1.04/0.78 | 0.75/1.06/0.60 |
| EnCDA: highest $NSE_{TWSA}$ | 0.77/1.04/0.66 | 0.82/1.29/0.77 | 0.73/1.34/0.86 | 0.89/1.23/0.55 | 0.92/1.13/0.66 | 0.71/1.10/0.58 |
| EnCDA: compromise | 0.77/1.04/0.66 | 0.84/1.15/0.65 | 0.83/1.07/0.76 | 0.92/1.13/0.69 | 0.92/1.04/0.78 | 0.73/1.05/0.60 |
| EnCDA: ensemble mean | 0.79/0.97/0.67 | 0.82/1.21/0.68/ | 0.86/1.12/0.76 | 0.91/1.17/0.61 | 0.94/1.13/0.68 | 0.75/1.11/0.58 |
| Standard calibration | 0.83/0.98/0.53 | 0.74/1.05/0.73 | 0.73/0.64/0.76 | 0.93/1.05/0.82 | 0.89/0.97/0.95 | 0.89/0.97/0.95 |
| Uncalibrated | 0.87/1.76/0.55 | 0.84/1.72/0.77 | 0.76/1.07/0.73 | 0.93/1.07/0.87 | 0.91/1.15/0.92 | 0.91/1.15/0.92 |

**Table B2.** KGE components of model runs of Table 3 regarding TWSA for the calibration period 2003-2012.

| | CC/RVar | | | | | |
| --- | --- | --- | --- | --- | --- | --- |
| | Arkansas | Missouri | Upper MRB | Ohio | Lower MRB | MRB |
| POC: highest $NSE_Q$ | 0.93/1.06 | 0.77/1.09 | 0.80/1.42 | 0.95/1.22 | 0.95/1.42 | 0.85/1.30 |
| POC: highest $NSE_{TWSA}$ | 0.95/1.04 | 0.91/1.03 | 0.84/1.07 | 0.95/1.05 | 0.97/1.05 | 0.93/1.10 |
| POC: compromise | 0.93/1.06 | 0.87/1.10 | 0.82/1.26 | 0.94/1.14 | 0.96/1.01 | 0.91/1.22 |
| GLUE: highest $NSE_Q$ | 0.89/1.01 | 0.57/0.91 | 0.80/1.47 | 0.92/1.14 | 0.92/1.67 | 0.75/1.37 |
| GLUE: highest $NSE_{TWSA}$ | 0.94/0.97 | 0.89/1.05 | 0.82/1.05 | 0.96/1.08 | 0.95/1.07 | 0.91/1.07 |
| GLUE: compromise | 0.94/1.16 | 0.87/1.07 | 0.81/1.26 | 0.94/1.13 | 0.94/0.93 | 0.87/1.20 |
| EnCDA: highest $NSE_Q$ | 0.83/1.27 | 0.78/0.65 | 0.77/1.11 | 0.95/0.98 | 0.94/1.04 | 0.44/0.71 |
| EnCDA: highest $NSE_{TWSA}$ | 0.94/1.14 | 0.83/0.65 | 0.83/0.98 | 0.97/0.90 | 0.96/1.00 | 0.52/0.72 |
| EnCDA: compromise | 0.94/1.14 | 0.82/0.67 | 0.80/1.03 | 0.95/0.98 | 0.94/1.04 | 0.49/0.72 |
| EnCDA: ensemble mean | 0.92/1.16 | 0.76/0.62 | 0.80/1.00 | 0.94/1.02 | 0.95/0.98 | 0.45/0.69 |



| Standard calibration | 0.83/1.19 | 0.63/0.76 | 0.63/1.09 | 0.91/1.13 | 0.80/1.63 | 0.71/1.11 |
|---|---|---|---|---|---|---|
| Uncalibrated | 0.85/1.09 | 0.63/0.76 | 0.62/1.10 | 0.90/1.21 | 0.82/1.60 | 0.73/1.11 |

*Code availability:* The WaterGAP 2.2d code is accessible at https://doi.org/10.5281/zenodo.6902110.

*Data availability.* All optimal and behavioral parameter sets obtained by the three calibration approaches for the six CDA units together with the resulting performance metrics are listed in an Excel file that is part of the supplement.


*Supplement.* The supplement related to this article (Excel file) is available online at URL???

*Author contributions.* PD designed the study, with contributions from HMMH, KS, SMHM, SS, AG and JK. HMMH, KS and HG performed calibration and data analyses. HMMH and SMHM produced the figures. SA and
HMS improved the WaterGAP code. LB and HG processed and analyzed GRACE TWSA data. PD wrote the original draft of the manuscript. All authors contributed to the final draft.

*Competing interests.* The authors declare that they have no conflict of interest.

*Acknowledgments.* This study was enabled by the financial support of the German Research Foundation for the research unit "Understanding the global freshwater system by combining geodetic and remote sensing information with modelling using a calibration/data assimilation approach (GlobalCDA)". The authors thank Olga Sydak (née Engels) for first analyses and discussions and Christoph Niemann for contributing to the generation of figures.

*Review statement.* This paper was edited by xxx and reviewed by xxx anonymous referees.

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
