# Peer review of "Leveraging multi-variable observations to reduce and quantify the output uncertainty of a global hydrological model: Evaluation of three ensemble-based approaches for the Mississippi River basin"

_Hydrology and Earth System Sciences, 2023_

## Author Response (AR1)

We thank both reviewers and the editor very much for their helpful comments and constructive suggestions for improving the manuscript. Below, each editor's and reviewer's comment is followed by our answer (indicated by "AC"). The new text in the revised manuscript is written in bold.

**Editor Ryan Teuling**

**E**: First of all, I would like to apologize for the time it has taken to receive sufficient reviewers for this submission, and to reach an initial decision. As you have seen, the two anonymous referees, whom I both consider experts in the field of regional-scale hydrological modeling and uncertainty analysis, both see your work as interesting and potentially suitable for HESS. However they also identified a considerable number of issues, several of which were shared between the reports. Given the nature of these issues, it seems that a considerable amount of rewriting, as well as some new analysis, might be needed, and this translates into major revisions. When preparing a revised version, you can generally follow the approaches outlined in your replies posted in the online discussion. I want to ask you to pay specific attention to the scientific contribution and wider relevance of your results. HESS generally does not publish case studies on a particular basin or using a particular model. Hence, your findings for the Mississippi using WaterGAP should be used to provide a more generic insight.

**AC**: The intention of the study is not to provide new knowledge about the Mississippi River basin or the WaterGAP model but to test the three ensemble-based calibration and uncertainty approaches with the aim to find out how they can be applied globally, also by other global hydrological models, for utilizing observations of multiple model output variables to reduce and quantify the model output uncertainty. We agree that in the original manuscript we did not formulate clearly the relevance of our study results for global hydrological modeling in general group. We are thankful to you and the reviewers for making us aware of this. To better show the scientific contribution and the wider relevance of our results for other (global) hydrological modelers, we completely rewrote the abstract, Section 5.1 and the conclusions (also including a figure that summarizes our recommendations but also caveats, see our answer to you third comment below. We also revised the introduction thoroughly. For example, we reformulated the objective of the study as follows:

**The objective of this paper is to analyze how the uncertainty of the output of GHMs can be reduced and quantified by parameter estimation that utilizes observations of multiple output variables and their uncertainties. For the example of the Mississippi River basin (MRB), the paper shows how Q and TWSA observations can be utilized to obtain one optimal parameters set (the "compromise solution") as well as ensembles of Pareto-optimal and behavioral parameter sets for the GHM WaterGAP, by evaluating the applicability of the three multi-variable calibration approaches POC, GLUE and EnCDA. It presents a method for defining performance thresholds for behavioral parameter sets based on observations and their uncertainties as well as the initial GLUE ensemble. In each approach, model parameters of all grid cells within so-called calibration-data assimilation (CDA) units, either the whole MRB or five sub-basin CDAs, were uniformly adjusted. We derive conclusions for multi-variable parameter estimation and quantification of model output uncertainty in global-scale hydrological modeling, answering the following research questions:**

**E**: Also, I do not agree with your reply on the issue of using NSE vs KGE. These are not independent metrics, and the mathematics behind the dependency are discussed extensively in

Gupta et al. (2009). Following your reasoning, I could see the value of optimizing against one of the components of KGE, and evaluating on the other(s), but given the nature of the components i do not expect expect them to contain much mutual information on hydrologic behavior.

**AC**: We changed the reply on the issue NSE vs KGE, which now reads:

"NSE is used for parameter estimation only in the case of POC and GLUE, but not in the case of EnCDA (where the objective function cannot be freely set, see Table A1, but is a weighted RMSE), and we evaluated the results using both NSE and the three components of KGE (not KGE itself). We think it is particularly informative to evaluate the result of a parameter set optimization using a certain objective function/performance metric by another performance metric, in particular the three KGE components as these are well interpretable (correlation: temporal shifts; bias: difference in means; variability ratio: difference in temporal variability). In the revised version, we have added a sentence to section 4.1 to indicate how the improvement in the correlation coefficient by the calibration can be interpreted: **Thus, calibration mainly leads to improved timing of monthly streamflow and TWSA.**"

**E**: In addition, I would like to ask you to consider optimizing the display items (figures and tables). There is a lot of information being presented, but key figures that condense some of the findings in clear concepts are missing. This makes the manuscript currently read perhaps more like a scientific report than a focused manuscript. I believe condensing some of the results, and using the space for 1-2 new figures summarizing the main results, could help in addressing several of the referees' comments.

**AC**: Table 1 was removed from the main text and the text in sections 1, 2, 4 and 5 was slightly condensed. We added a figure (Figure 7) to the new Section 6 Conclusions that summarizes the main results of the study.

**E**: I am looking forward to receiving a revised version of your work. Given the nature of the comments, this new version will be returned to the referees for consultation

**Reviewer 1**

**RC:** The study by Doell et al. compares three different strategies to reduce parameter uncertainty for the global hydrological model WaterGap. The methods used are BORG, GLUE, and an ensemble Kalman filter, which the authors apply in a pilot study to the Mississippi basin. How we best estimate global water models is an interesting and relevant question to which the authors contribute. I do like the study and what the authors do and show, but I have some critical comments regarding how the work is currently presented and discussed. I outline my main comments below.

**AC**: Thank you for the positive feedback.

**RC**: [0] The authors' use of sensitivity analysis is very nice and interesting, but the results are hardly discussed. I would have liked to see more detail on these results. For example, the precipitation multiplier is not slected as important. Interesting, given that this parameter is often very relevant. Is rthis due to the monthly time step? The authors study a huge domain. How did sensitivity to the parameters vary across this domain? A lot of insights to be gained from this analysis, but they are not discussed. I think this would be worth including rather than some other parts as suggested below.

**AC**: In the revised version, we added a paragraph to section 3.2.4 in which we discuss the new table below, which was added to the supplement as Table S2. In this and the following paragraph, we show in more detail which output variables are sensitive to which parameters and how this differs among the CDA units.

**Table S2. The most influential parameters for streamflow, TWSA, snow cover and local lake storage, covering together at least 50% of the total effect.**

| CDA Unit | Streamflow | TWSA | Snow cover | Local lake storage |
|---|---|---|---|---|
| **I Arkansas** | **SL-RC, SL-MSM, EP-PTh, SL-MEP, GW-MM** | **SL-RC, SL-MSM, NA-GM** | **SN-MT** | **SW-LD, SW-DC** |
| **II Missouri** | **SL-RC, SL-MSM, EP-PTh, SN-MT, NA-SM** | **SL-RC, SL-MSM, SW-WD, EP-PTh, NA-GM** | **SN-MT** | **SW-LD, SW-DC, NA-SM** |
| **III Upper MRB** | **SL-RC, SL-MSM, EP-PTh, SN-MT, GW-MM** | **SL-RC, SL-MSM, SW-WD, SW-DC, EP-PTh** | **SN-MT** | **SW-LD, SW-DC** |
| **IV Ohio** | **SL-RC, SL-MSM, SW-RRM, EP-PTh, GW-MM** | **SL-RC, SL-MSM, EP-PTh, GW-DC** | **SN-MT** | **SW-LD, SW-DC** |
| **V Lower MRB** | **SL-RC, SL-MSM, SW-RRM, EP-PTh, SN-MT** | **SL-MSM, GW-RFM, NA-GM** | **SN-MT** | **SW-LD, SW-DC** |
| **MRB** | **SL-RC, SL-MSM, SW-RRM, EP-PTh** | **SL-RC, SL-MSM, EP-PTh, NA-GM** | **SN-MT** | **SW-LD, SW-DC** |

**Note that although SW-WD was not selected in unit I, IV, V, MRB, we decided to select the parameter for all units due to effect on groundwater recharge from surface water bodies**

Regarding the precipitation multiplier P-PM, we write in line 583 "P-PM was excluded from calibration even though it ranked 1st in the sensitivity analyses in all six basins for almost all four test variables because the precipitation input is perturbed in EnCDA, and an additional multiplier would lead to a double-counting of precipitation uncertainty." So one reason for not including P-PM in POC and GLUE was that we wanted to compare all three calibration methods. The other reason was that different from other basins such as the Amazon or the Ganges-Brahmaputra basins, precipitation in the Mississippi River Basin is expected to be rather well represented by the climate data used as input to WaterGAP. The mean annual precipitation in the CDA units that was used to drive WaterGAP does not differ much from the values derived from the high-resolution (4 km) PRISM dataset for the USA. In the revised version, we extended the explanation for why we did not use P-PM as calibration parameters and referred to the new Table S1 below that was added to the supplement. The new text in section 3.2.4 reads:

**P-PM was excluded from calibration even though it ranked first in all six CDA units for almost all four test variables, for various reasons. First, the precipitation input is perturbed in EnCDA, and an additional multiplier would lead to a double-counting of precipitation uncertainty. Second, mean annual precipitation in the CDA units of WaterGAP climate forcing does not differ much from the values derived from the high-resolution (4 km) PRISM dataset for the USA (Table S1).**

**Table S1. Comparison of mean annual precipitation in the CDA units for the calibration period 2003-2012 between GPCC-WFDEI used to drive WaterGAP and the high-resolution (4 km) PRISM\* dataset for the USA [mm/yr]**

| CDA unit | GPCC-WFDEI | PRISM | Ratio PRISM/GPCC-WFDEI (potential P-PM) |
|---|---|---|---|
| I Arkansas | 705 | 667 | 0.95 |
| II Missouri | 595 | 622 | 1.04 |
| III Upper MRB | 951 | 878 | 0.92 |
| IV Ohio | 1313 | 1242 | 0.95 |
| V Lower MRB | 1286 | 1254 | 0.97 |
| MRB | 839 | 829 | 0.99 |

**\*https://climatedataguide.ucar.edu/climate-data/prism-high-resolution-spatial-climate-data-united-states-maxmin-temp-dewpoint**

**RC**: [1] This is a very long paper with a lot of details on the model and the data that, at least to me as a reader, seems excessive and not needed to understand the main story presented. It makes reading the paper a bit tedious because most readers will not run WaterGap and they might not even be interested in the extensive background information on the data (as part of the main story).

For example, lines 500-508 discuss problems with the GRACE data and how others have gone about reducing them. Is this really something I need to know to follow the story? I think text like this can go into the supplemental material without reducing the strength of the story told. On the contrary, it would make it better because I do not have to read through this background information unless I want to.

Lines 466-508 discuss details of the GRACE data and their uncertainties in (excessive) detail. At the same time, the authors spent one sentence on stating that two studies considered streamflow errors of about 10%, while the next sentence states that this is maybe a possible average but the variability is very large. The authors spend over 60 lines discussing GRACE and 6 (ok 7) lines to discuss the other variable they use. I do not understand why the authors do not present a more balanced discussion given that both variables suffer from significant and potentially complex uncertainties.

**AC**: Regarding the description of the WaterGAP model in section 3.1, we constrained the information to the information that is necessary to understand 1) the meaning and importance of parameters that are to be estimated by the multi-variable calibration and 2) the differences between the multi-variable calibration presented in the manuscript and the (very simple) standard calibration of WaterGAP. Thus, we think that it is not beneficial to shorten the model description or move it to the supplement.

Regarding the description of the GRACE TWSA data, we agree with the reviewer that there is excessive detail in the main text. We moved the text on leakage problems and other aspects related to the uncertainty of GRACE TWSA data (lines 483 to 508 in version 1 of the manuscript) to the supplement as Text S1. To increase the readability by decreasing the length

of the main text, we also moved section 2.4 "Comparison of the three calibration approaches" (lines 276-377, including Table 1) to the Appendix.

**RC**: [2] Starting from the back, i.e. the Outlook section, I wonder what transferrable knowledge the authors contribute that is unrelated to using WaterGap (and potentially the traditional approach to calibrating WaterGap)?

My impression is that most of the conclusions are rather specific to the use of WaterGap. I do not think that this is a problem per se, but it would be good if the authors would be clearer about general outcomes and those specific to WaterGap. One problem in this context is that Discussion and Conclusions are jointly discussed and that this section is 7 pages long. I think these sections can be joined if this part of the paper is short, but here it is very long. A long discussion followed by a very short conclusions and outlook section would make it much easier for the reader. There the authors could also easily separate specific and general conclusions.

**AC**: We have followed your suggestion to split the "Discussion and Conclusions" section and organize the last part of the manuscript as follows:

5 Discussion (with sections 5.1 to 5.6)

6 Conclusions (which includes, in revised form, what was 5.7 Outlook).

We completely rewrote the conclusions (and partly the abstract), where we now focus on clearly saying what other (global) hydrological modelers can learn from our study, regarding the application of the three alternaive calibration approaches and caveats, and added a figure to the conclusions that concisely represents the proposed approach (**Figure 7: The proposed approach for reducing and quantifying model output uncertainty of GHM by multi-variable parameter estimation and main recommendations and caveats for applying the approach in global hydrological modeling.**)

**RC**: [3] The final recommendation to include uncertainty in climate change impact projections related to freshwater is good, but this is already widely done (see below). Can the authors be more specific regarding their recommendation? They could for example discuss this issue much more in the context of global models and the specific implications this has.

Just a few random examples from a quick online search:

https://www.nature.com/articles/s41598-019-41334-7

https://hess.copernicus.org/articles/21/4245/2017/

https://agupubs.onlinelibrary.wiley.com/doi/full/10.1029/2011WR010602

**AC**: Thank you for making us rethink our recommendation on including parameter uncertainty in a multi-model ensemble of impact models to project climate change hazards. Focusing on global-scale climate change impact studies, we will replace the last paragraph of the main text

"We recommend including, in future freshwater-related climate change impact studies, a behavioral ensemble of parameter sets as determined by the GLUE approach even though this will require a significant computational effort. This should reduce the underestimation of modeling uncertainty by traditional multi-model studies. As shown in the multi-model/multi-parameter study for the Colorado River basin by Mendoza et al. (2016), parameter sets with a similar performance during the calibration period may provide very different projections of climate change hazards, and the impact of parameter uncertainty is similar to the impact of hydrological model selection. "

by the following:

**Climate change impact studies for individual river basins have shown that parameter sets with a similar performance during the calibration period may provide very different projections of climate change hazards, and that the impact of parameter uncertainty can be similar to the impact of the selected climate or hydrological model selection (Mendoza et al., 2016; Her et al., 2019). Therefore, consideration of parameter uncertainty by running the hydrological model with a number of behavioral parameter sets helps to reduce the underestimation of the uncertainty of potential climate change impact. However, producing a global-scale ensemble of potential future changes in hydrological variables by combining not only multiple greenhouse gas emissions scenarios, global climate models and global hydrological models (as is currently done in ISIMIP) but also model-specific behavioral parameter sets is currently infeasible. The main reason is that behavioral (or even optimal) parameter sets have not yet been determined for any global hydrological model in a spatially explicit manner at the global scale. In addition, the computational effort for such a multi-model/multi-parameter ensemble is likely prohibitive.**

RC: [4] The connection to existing literature is in places very extensive and in others very brief. All methods used here have been previously assessed widely. Maybe not exactly in this combination, but certainly individually or in combination with other methods. I would therefore have expected that the authors help the reader to start from a more informed level.

For example, the (poor) ability of GLUE to identify the best parameter set has been explored in the past (see link below) and thus is what should be expected. The issue now is rather what relevance this has for the study at hand.

https://backend.orbit.dtu.dk/ws/portalfiles/portal/9729153/MR2007_305.pdf

AC: With the help of both reviews, we noticed that we have not clearly described the role of GLUE (with a random ensemble of parameter sets) as compared to the role of POC (in which a search algorithm derives an ensemble of Pareto-optimal parameter sets). In the revised version of the manuscript, we revised the introduction and developed the storyline of the paper differently. Before formulating the study objective, we now state (citing Blasone et al. 2008) that optimal parameter sets are best identified using a search algorithm, as used in POC, while GLUE serves, in the face of equifinality, to identify behavioral parameter sets and thus to quantify the model output uncertainty. Connected to this, we changed the title of the manuscript from "Multi-variable parameter estimation for a global hydrological model: Comparison and evaluation of three ensemble-based calibration methods for the Mississippi River basin" to

**Leveraging multi-variable observations to reduce and quantify the output uncertainty of a global hydrological model: Evaluation of three ensemble-based approaches for the Mississippi River basin**

We also revised the conclusions accordingly.

**RC**: [5] While I possibly sound rather critical, I think this is an interesting and relevant study. My comments are simply meant to help the authors communicate their work with the readers. Shortening the paper, being clearer about specific and general contributions, and a better connection with existing literature would make it much easier for readers to understand the study and its relevance

**AC**: We will direct our revision in this way.

**Reviewer 2**

**RC:** Döll et al. exploit three different approaches to identify parameters in a WaterGAP model of the Mississippi River Basin. This should provide insights for the calibration of global hydrological models. Although as a reviewer I aim to be constructive and to provide concrete recommendations, I have to admit that I found this difficult for the presented study. I hope I can make my points clear and that this provides enough guidance for the authors to search for other directions.

**AC**: Thank you for your critical feedback that will help us to better communicate the objectives, results and conclusions of our study.

**RC**: The long, quite unfocused, introduction seems to give the goal of this study (line 70), where the complex and long research question is already a preparation for the reader on what is coming. My summary of the goal of the study, if I understood correctly, would be to explore how global hydrological models can be calibrated in order to make better use of available observations.

**AC**: The manuscript does aim at showing how to make (better) use of available observations in global hydrological modeling beyond streamflow but it is not only about calibration in the sense of finding optimal parameter sets but also about estimation of model output uncertainty. In the revised version, we will therefore change the title of the manuscript from "Multi-variable parameter estimation for a global hydrological model: Comparison and evaluation of three ensemble-based calibration methods for the Mississippi River basin" to

**Leveraging multi-variable observations to reduce and quantify the output uncertainty of a global hydrological model: Evaluation of three ensemble-based approaches for the Mississippi River basin**

We have condensed and restructured the introduction to better fit to the revised paper title and to clarify early on that GLUE is not applied to see whether is as efficient or better than POC (with an optimization algorithm) but to be able to determine behavioral parameter sets given the uncertainty of the observations and thus quantify model output uncertainty. Most importantly, we have changed the formulation of the research objective. We have replaced

"The objective of this paper is to assess the suitability of the three multi-variable calibration approaches POC, GLUE and EnCDA for identifying ensembles of optimal and behavioral parameter sets of the GHM WaterGAP by model calibration against observations of Q and TWSA, taking into account observation uncertainties. In addition, an approach for taking into account the observation errors for the definition of performance thresholds for behavioral parameter sets is presented. In each calibration approach, model parameters of all WaterGAP grid cells within so-called calibration-data assimilation (CDA) units were uniformly adjusted. Based on calibration exercises either for the whole Mississippi River basin (MRB) as one CDA unit or for its five sub-basins (four upstream basins and one downstream basin) as alternative CDA units, we will answer the following research questions:"

by

**The objective of this paper is to analyze how the uncertainty of the output of GHMs can be reduced and quantified by parameter estimation that utilizes observations of multiple output variables and their uncertainties. For the example of the Mississippi River basin (MRB), the paper shows how Q and TWSA observations can be utilized to obtain one optimal parameters set (the "compromise solution") as well as ensembles of Pareto-optimal and behavioral parameter sets for the GHM WaterGAP, by evaluating the applicability of the three multi-variable calibration approaches POC, GLUE and EnCDA. It presents a method for defining performance thresholds for behavioral parameter sets based on observations and their uncertainties as well as the initial GLUE ensemble. In each approach, model parameters of all grid cells within so-called calibration-data assimilation (CDA) units, either the whole MRB or five sub-basin CDAs, were uniformly adjusted. We derive conclusions for multi-variable parameter estimation and quantification of model output uncertainty in global-scale hydrological modeling, answering the following research questions:**

**RC**: In my understanding and experience, one of the reasons why GHMs currently are not thoroughly or automatically calibrated is mainly because of computational demand, besides model complexity (leading to non-uniqueness). The argument of computational demand has, surprisingly, not been taken into account in any way in selecting calibration approaches for this study. Expensive algorithms and approaches, such as Borg-MOEA and EnKF are explored, already going towards computational limits for the basin explored here. How is this ever going to translate to a global application then?

**AC:** We wanted to explore whether it is possible to benefit from the advantages of EnKF (EnCDA), which - different from typical calibration of hydrological models - simultaneously adjusts system states (water storages) and model parameters. We hypothesize that this property can be of advantage in situations where a model has structural deficiencies that cannot be "absorbed" via parameter calibration. The goal of the study was to explore whether EnKF can be used to assimilate not only observations of total water storage anomalies, as has already been shown to be feasible and successful at the global scale (e.g., Gerdener et al. 2023) but also streamflow observations (which had not yet been demonstrated), while at the same time adjusting parameters (which also had not yet been demonstrated in this context). We hypothesized that taking into account, in EnKF, both the uncertainties of the climate input and 2) adapting storages, parameter estimates would stabilize towards the end of the calibration period; these parameters could then be used for periods without observation data. Our study has shown that in the current setting of this study, the EnKF approach was less successful in these aspects compared to POC and GLUE, and was thus not found as applicable for reducing and quantifying the uncertainty of output of WaterGAP. We would

like to point out that compared to an uncalibrated run, in the present setting, EnCDA does show improvements. Increasing the numerical efficiency of the framework even with very large state vectors, as it could be the case for a global EnCDA, is still under development. After submitting this paper, the run time of the assimilation setup was already strongly improved by avoiding reading in and writing to the hard disc, reducing the run time of a global GRACE assimilation by 75%. It is well-known that EnKF performance relies very much on the proper representation of model state and, in this case, parameter correlations, and this in turn depends on ensemble size. Our EnKF may improve in the given setting for larger ensembles, but this is indeed computationally very demanding at the global scale. This has been expressed in section 5.7 (now section 6 Conclusions), where we write "Likely because of the very small ensemble size that was feasible in EnCDA due to its severe computational burden, parameter sets and model output uncertainties could not be estimated by EnCDA in this study, neither for the calibration period nor for projections for periods without observation data."

We are convinced that both GLUE and Borg-MOEA (for POC) are not too expensive to be applied in global hydrological modeling. With 20,000 ensemble members, the run times for six CDA units in our study were 72 hours and 53 hours for POC and GLUE, respectively, and, for 32 ensemble members in the case of EnCDA, 72 hours, in the parallel computing environments described in the manuscript (section 3.4). We are currently setting up a global POC for 712 calibration units (drainage basins) covering and based on runs with a small number of calibration units we estimate the total runtime in case of 20,000 ensemble members to be 15-20 days. In times of high-performance computing, computational demand for global-scale multi-variable parameter estimating is very high but not prohibitive. We have added the information about runtimes to the revised version of the manuscript, in sections 3.4.1, 3.4.2, and 3.4.3.

**RC**: But besides, there were other reasons to be surprised by the selected methods and approaches. The three methods seem to be presented as calibration strategies, but I would argue they are not. GLUE is presented as an optimization technique, while it is merely a way of evaluating a sample. Therefore, it should not come as a surprise that Borg-MOEA outperforms GLUE; the authors already write themselves that the search algorithm searches in the region of interest, while GLUE is just a sample across the whole parameter space. This conclusion, therefore, could have been drawn without doing all the computations. The same holds true for the EnCDA. An implementation of EnKF is used as a way of calibrating, but EnKF has never been developed to serve as a calibration algorithm. It is useful for real-time applications, it is useful to identify model structural errors, but it never claims a convergence towards an optimal parameter set. Therefore, no surprise that results drifted off in the validation period!

**AC**: We would argue that both Borg-MOEA and GLUE are calibration strategies; Borg-MOEA is a technique for identifying (Pareto-) optimal parameter sets, while GLUE is a technique for identifying behavioral parameter sets but can be also used to determine (in a sub-optimal way compared to POC) optimal (i.e., best-behaving) parameters sets. In this way, both are calibration techniques. GLUE approaches were called calibration, for example, in Marmy et al. (2016) and Wu and Jansson (2013).

With the help of both reviews, we noticed that we did not clearly formulate the role of GLUE (with a random ensemble of parameter sets) as compared to the role of POC (in which a search algorithm derives an ensemble of Pareto-optimal parameter sets). In the revised manuscript, we therefore modified the storyline and improved the insufficiently clear

presentation of the roles of Borg-MOEA and GLUE. Before formulating the study objective, we now state (citing the additional reference Blasone et al. 2008) that optimal parameter sets are best identified using a search algorithm, as used in POC, while GLUE serves, in the face of equifinality, to identify behavioral parameter sets and thus to quantify the model output uncertainty. We also revised the conclusions accordingly.

Regarding EnKF, it is true that EnKF has never been developed as a calibration algorithm, and our research investigated whether EnKF can serve to estimate parameters of a global hydrological model using observations of streamflow and TWSA. EnKF has been demonstrated in various studies to improve the realism of global hydrological model simulations when compared to various observations, and this includes our own EnKF implementation at global scale with WaterGAP, which however, only takes into account TWSA observations (Gerdener et al., 2023). It is one of the standard techniques when multiple data sets, at different spatial scales and with possibly differing temporal or spatial coverage are to be combined, such as in meteorological or hydrological reanalyses. Various papers (e.g., Wanders et al., 2014, cited in the manuscript) have shown, typically in regional or local settings, that the EnKF variants are capable of estimating model parameters along with model states. Therefore, we believe it is perfectly reasonable to ask whether EnKF is able, at the same time, to estimate model parameters albeit maybe not as efficient as a dedicated calibration approach. Some of the reasons why EnKF has the potential for improved parameter estimation are provided in section 2 of the manuscript and our response to the previous comment.

We agree that from a parameter calibration perspective, deriving an optimal parameter set from EnKF seems complicated. POC and GLUE generate constant parameter sets. However, we hypothesize that the updates of the water storages could stabilize the parameters and compensate for model structure deficiencies and climate input uncertainties. EnKF generates a time series of estimates of parameter sets, which is often misunderstood as generating time-variable parameters. This time series which in the ideal cases converges may include typical seasonal signals, and such signals point towards model errors and are difficult to interpret. In this study, we decided to apply the parameter estimates of the last month of the calibration phase during the validation phase, to be able to compare the different ensemble-based approaches for reducing and quantifying uncertainty — which is the aim of this study. Future studies will investigate how seasonal signals in the parameter estimates can be used to (1) trace back model errors and (2) develop empirical error models, which can include parameterizations depending on the season.

RC: Besides these methodological issues, the study is hard to read and follow. Only at page 20 (!) I felt that I got a more concrete picture of what was done. And even then, it read a lot like a diary. For instance, first I was very very surprised at line 520 that also a multiplication factor for precipitation and net radiation were included as calibration parameters. Then I was not so surprised to find out that the multiplication factor for precipitation came out as most sensitive (l. 582), to then I was surprised again to learn that it was still left out of the calibration (l. 585). I know that there is an argument for documenting failures etc., but I don't think this is helpful at all at this level: just leave this kind of stuff out, don't bother the reader with it. Furthermore, there is some kind of strange mixed use of NSE and KGE. The NSE is optimized, but the KGE components are evaluated. Why not directly optimizing the KGE then? That would lead to different results compared to the NSE. Figure 3 shows NSE's if I read the axes, but the caption refers to some kind of KGE.

**AC**: To increase the readability, we decreased the length of the main text. We have moved section 2.4 "Comparison of the three calibration approaches" (lines 276-377, including Table 1) to the Appendix. Regarding the description of the GRACE TWSA data, we moved the text on leakage (lines 483 to 508) to the supplement.

Regarding the reviewer's comment on the method descriptions reading as a diary, our goal was to make transparent to the reader the many decisions that need to be taken in parameter estimation. Regarding the process of deciding on whether to include the precipitation multiplier P-PM as a calibration parameter, it has nothing to do with documenting a failure but with explicating why it was excluded even though model results are sensitive. While we could remove this from the manuscript, reviewer 1 wanted to get a deeper discussion on the selection of calibration parameters and is interested in a more detailed explanation for the exclusion (R1 comment 0). We therefore revised the part on the precipitation multiplier as follows (section 3.2.4):

**P-PM was excluded from calibration even though it ranked first in all six CDA units for almost all four test variables, for various reasons. First, the precipitation input is perturbed in EnCDA, and an additional multiplier would lead to a double-counting of precipitation uncertainty. Second, mean annual precipitation in the CDA units of WaterGAP climate forcing does not differ much from the values derived from the high-resolution (4 km) PRISM dataset for the USA (Table S1).**

**Table S1. Comparison of mean annual precipitation in the CDA units for the calibration period 2003-2012 between GPCC-WFDEI used to drive WaterGAP and the high-resolution (4 km) PRISM\* dataset for the USA [mm/yr]**

| CDA unit | GPCC-WFDEI | PRISM | Ratio PRISM/GPCC-WFDEI (potential P-PM) |
|---|---|---|---|
| **I Arkansas** | 705 | 667 | 0.95 |
| **II Missouri** | 595 | 622 | 1.04 |
| **III Upper MRB** | 951 | 878 | 0.92 |
| **IV Ohio** | 1313 | 1242 | 0.95 |
| **V Lower MRB** | 1286 | 1254 | 0.97 |
| **MRB** | 839 | 829 | 0.99 |

**\*https://climatedataguide.ucar.edu/climate-data/prism-high-resolution-spatial-climate-data-united-states-maxmin-temp-dewpoint**

Regarding the mixed use of NSE and KGE: NSE is used for parameter estimation only in the case of POC and GLUE, but not in the case of EnCDA (where the objective function cannot be freely set, see Table A1, but is a weighted RMSE), and we evaluated the results using both NSE and the three components of KGE (not KGE itself). We think it is particularly informative to evaluate the result of a parameter set optimization using a certain objective function/performance metric by another performance metric, in particular the three KGE components as these are well interpretable (correlation: temporal shifts; bias: difference in means; variability ratio: difference in temporal variability). In the revised version, we have added a sentence to section 4.1 to indicate how the improvement in the correlation coefficient by the calibration can be interpreted: **Thus, calibration mainly leads to improved timing of monthly streamflow and TWSA.**

We have corrected the typo (KGE) in the caption of Figure 3, it should read NSE.

**RC**: Finally, there is no conclusion-section, just a very extensive "Discussion and conclusion", which is already indicative that there are too many separate aspects that are aimed to be tackled in this study. This study aimed to serve the GHM community, but the kind of strategies and questions explored here have already been extensively addressed and investigated by regional scale models – with the same conclusions as this study. Now the challenge remains how to translate this to models applied to larger areal extends, and this study does not seem to contribute to that.

**AC**: We will follow your suggestion (and that of the other reviewer) to split the "Discussion and Conclusions" section and organize the last part of the manuscript as follows:

5 Discussion (with sections 5.1 to 5.6)

6 Conclusions (which includes, in revised form, what is now 5.7 Outlook).

To better show the scientific contribution and the wider relevance of our results for other (global) hydrological modelers, we have completely rewritten the conclusions, focusing on what was learned regarding methods for global-scale reduction and quantification of the output uncertainty of global hydrological models by the three approaches for multi-variable parameter estimation POC, GLUE and EnCDA. We also concluded that based on the experiences in our study, run times for a global-scale application for, e.g., 1000 basins are not prohibitive.

However, due to the high computational demand, those many behavioral parameter sets could not be used in climate change impact studies, neither if just WaterGAP were applied and certainly not in a multi-model ensemble with various global hydrological models. Regarding the consideration of parameter ensembles in global-scale climate change impact studies, we have therefore changed our conclusion and replaced the last paragraph of the main text

"We recommend including, in future freshwater-related climate change impact studies, a behavioral ensemble of parameter sets as determined by the GLUE approach even though this will require a significant computational effort. This should reduce the underestimation of modeling uncertainty by traditional multi-model studies. As shown in the multi-model/multi-parameter study for the Colorado River basin by Mendoza et al. (2016), parameter sets with a similar performance during the calibration period may provide very different projections of climate change hazards, and the impact of parameter uncertainty is similar to the impact of hydrological model selection. "

by the following:

**Climate change impact studies for individual river basins have shown that parameter sets with a similar performance during the calibration period may provide very different projections of climate change hazards, and that the impact of parameter uncertainty can be similar to the impact of the selected climate or hydrological model selection (Mendoza et al., 2016; Her et al., 2019). Therefore, consideration of parameter uncertainty by running the hydrological model with a number of behavioral parameter sets helps to reduce the underestimation of the uncertainty of potential climate change impact. However, producing a global-scale ensemble of potential future changes in**

**hydrological variables by combining not only multiple greenhouse gas emissions scenarios, global climate models and global hydrological models (as is currently done in ISIMIP) but also model-specific behavioral parameter sets is currently infeasible. The main reason is that behavioral (or even optimal) parameter sets have not yet been determined for any global hydrological model in a spatially explicit manner at the global scale. In addition, the computational effort for such a multi-model/multi-parameter ensemble is likely prohibitive.**

RC: Overall, the methods seem to be not in line with the goals that this study aims to achieve, and the written presentation requires substantial improvement.

AC: As described above, in the revised version we have changed the title and framed the three methods and study goals more clearly. And we have improved the presentation, mainly by reformulating, restructuring and shortening; we have fully rewritten the conclusions and strongly revised the abstract, the introduction, section 5.1 and the conclusions.

**References**

Gerdener, H., Kusche, J., Schulze, K., Döll, P., Klos, A. (2023): The Global Land Water Storage Data Set Release 2 (GLWS2.0) derived via assimilating GRACE and GRACE-FO data into a global hydrological model. J. Geodesy, 97, 73. https://doi.org/10.1007/s00190-023-01763-9.

Marmy et al. (2016): Semi-automated calibration method for modelling of mountain permafrost evolution in Switzerland. The Cryosphere, 10, 2693–2719. doi:10.5194/tc-10-2693-2016

Wu, S.H., Jansson, P.-E. (2013): Modelling soil temperature and moisture and corresponding seasonality of photosynthesis and transpiration in a boreal spruce ecosystem. Hydrol. Earth Syst. Sci., 17, 735–749. doi:10.5194/hess-17-735-2013

---

## Author Response (AR2)

HESS-2023-18: Leveraging multi-variable observations to reduce and quantify the output uncertainty of a global hydrological model: Evaluation of three ensemble-based approaches for the Mississippi River basin

P. Döll et al.

We thank both referees and the editor very much for their helpful comments and constructive suggestions for improving the manuscript. Below, each editor's and reviewer's comment is followed by our answer (indicated by "AC"). The new text in the revised manuscript is written in bold.

**Editor Ryan Teuling**

Two referees (of which one has also reviewed the initial submission) have now submitted their reports on the revision of your manuscript. As you will see, both assessments largely agree in that they recognize the improvements that you have made, but also point out a number of issues that need improvement. Specifically, both reviewers identify the need for a more thorough discussion, with a reflection on the literature mentioned in the Introduction. Anonymous referee #1 also lists a number of other issues that you will need to address. I think both reports will help you to further improve your manuscript. I am looking forward to receiving a revised version that addresses the issues and remaining concerns from the referees.

AC: We have further improved the manuscript by following the advice of the two referees. In addition to modifying the discussion, we revised the abstract, and to clarify the role of EnCDA (Referee 1), modifications were made in the introduction and methods section as well as in the conclusion. Below please find our responses to the comments of the referees.

**Referee 1**

RC: The authors significantly revised the manuscript. I like that some sections have been moved to the supplemental material or shortened. And I appreciate that some parts are better explained now. I still have some issues with the (new) discussion section and how the contributions are assessed. My main concern is that the discussion section makes hardly any effort to place the study's findings into the context of existing literature. Please see my detailed comments below.

AC: We are glad that with the previous resubmission we achieved an improvement in clarity and readability. We have now followed your advice and revised the discussion section. For details, please see our responses below.

(1) RC: You now state that you perturb the precip multiplier in the EnCDA runs, but not in the POC and GLUE runs. Is this a fair comparison then? It is surprising that EnCDA does not perform well if you perturb multiplier, parameters, and states, while you only calibrate the parameters with the other strategies. I am unclear why EnCDA worked well in previous studies where TWSA was assimilated, but not here. Would the required ensembles not be known from previous work by the authors?

AC: Thank you for this comment. The comparison is certainly not a fair one, but that would be extremely difficult to facilitate and it also may not reflect reality.

We conduct a specific model calibration exercise here, and we test two methods (POC and GLUE) that were specifically developed for model calibration, i.e. parameter estimation, together with a third one (EnKF) that was specifically developed for a different purpose, i.e. state estimation. The EnKF is slightly modified here to facilitate parameter estimation, by extending the state vector, which is new at least in this setting (global hydrology model calibration from GRACE and streamflow data), at least to our knowledge. We find here that the EnKF actually works quite well during the assimilation phase, and this is not at all in contrast with other studies and also not with our own work on assimilating just TWSA (Eicker et al., 2014; Schumacher et al., 2018; Gerdener et al., 2023. But it did not perform equally well in estimating model parameters with which we then run a free (uninformed by GRACE and streamflow) simulation in the validation period. We do not know of any comparable studies. EnKF is first of all developed to optimized state estimates (different from POC and GLUE) and here we try to see how far it can compete with dedicated calibration approaches, in a setting where POC and GLUE are expected to perform well. Put in other word, it would have been a big surprise if EnKF had been on par in the experiment performed here, even though we had hoped that due to e.g. balancing precipitation input uncertainties by state adjustment, parameter estimation might be improved. No study is known to us that had systematically compared model parameter estimation from TWSA data assimilation to other more dedicated calibration approaches, and thus certainly no study that would even compare multi-variable EnCDA to calibration approaches.

The other issue that we would like to recall is that EnCDA, as a variant of EnF, optimizes a weighted RMSE of observations, and this is in contrast to the POC and GLUE method, whereas "performance" in this paper is mainly understood in terms of NSE which is the metric optimized commonly in model calibration approaches (and in POC and GLUE here). That might be another reason why the EnKF does not compare so well.

To clarify the EnCDA approach, we revised the manuscript in numerous places, in the abstract, the introduction (e.g. lines 131-148 and extension of research question 1), Section 2.3 on the EnCDA method in general, Section 3.4.3 on the specific EnCDA implementation (adding one reference), the discussion sections 5.1 (adding two references) and 5.4 and the conclusions (Section 6, in particular lines 1287-1298).

(2) RC: You conclude (in the abstract) that POC is better than GLUE for finding optimal parameters, which has been shown many times before. More interesting is the issue that EnCDA did not work well. Why that was the case might warrant more space in the abstract than mentioning accepted knowledge.

AC: Please see our answer to RC (1). There may be multiple reasons here, and one of them is certainly the ensemble size. The ensemble size for Ensemble Kalman filters is generally much smaller as compared to those in model calibration approaches simply since one does not only perturb the parameters but the states and the forcing. Given limited compute time, we could increase the parameter ensemble in EnCDA greatly and likely obtain much better parameter estimates, if we would remove forcing perturbation and limit or even remove the state perturbation, but then we would simply not be running an EnKF anymore. The experiment might be considered more "fair" in the comparison with POC and GLUE, but that is simply not the real situation where we work with data assimilation. There are other reasons – the EnKF does not allow one to look at all at a Pareto front in hindsight, since one has to specify the weighting between GRACE and streamflow data right from the start. And the optimization functional of the EnKF is also different from many of the metrics considered

here. Then, we know that an Ensemble Kalman smoother (EnKS) would make better use of all data for providing the final state and parameter estimates, but the numerical expense would be twice as with the EnKF. Lastly, the EnKF and EnKS represent unbiased estimators only under limited conditions which are here clearly not met, in particular for streamflow simulation.

We have revised the section on the performance of the three approaches, which now reads:

**The GLUE approach is almost as successful as POC in enhancing WaterGAP performance and also allows, with a comparable computational effort, the estimation of model output uncertainties that are due to the equifinality of parameter sets given the observation uncertainties. Our experiment reveals that the EnCDA approach performs similarly in most CDA units during the assimilation phase but is not yet competitive for calibrating global hydrological models; its potential advantages remain unrealized, likely due to its high computational burden, which severely limited the ensemble size, and the intrinsic nonlinearity in simulating Q.**

(3) RC: The discussion of why EnCDA is so poor – given that it has more degrees of freedom than the other methods, is rather brief. (a) I would have assumed that the authors create a smaller testcase to see whether the problem is simply due to insufficient ensemble members. Does the problem go away if more samples are used? (b) Is the statement that Q simulations would have been better with transformed data or by ignoring zero flows correct. You would have obtained a better NSE value, because you reduce peak errors or include only a subset of the data. The simulations stay the same, you just have a different statistical metric that gives a higher value.

AC: Regarding (a): With increasing the ensemble gradually, e.g. to 64 or 100 members, results in EnCDA gradually improve but in this way the method would become unfeasible if one wants to compete with POC or GLUE in terms of ensemble size (several ten thousands). The problem does not go away. The smaller ensemble and the related loss of estimation power is the unavoidable price that we have to pay for using a recursive state estimator in contrast to the dedicated calibration method. The ensemble choice that we made here (n=32) corresponds to our earlier work and also to other studies (Zaitchik et al., 2018; Girotto et al., 2016; Kumar et al., 2016; Getirana et al, 2020a/b). With gradually increasing the size the estimators increase gradually in accuracy, but it will be never possible to use ensemble sizes as common in POC. We have added the above discussion in Section 5.1.

Regarding (b): The reviewers comment "The simulations stay the same, you just have a different statistical metric that gives a higher value" is not correct for the EnKF, at least not if we would have stayed with the usual least squares metric within the EnKF optimization. function. During the simulation the state and parameter updates would be different and therefore the simulation (after these updates) would be different. But we feel we cannot add this to the already very long manuscript – it has to wait until we publish an update.

(4) RC: The authors state that "However, the added value of any calibration is very low in the humid and hilly Ohio basin where the performance of the uncalibrated model is already good". While this is interesting, it would be good to place this section into the context of existing literature. For example (just doing a quick scan across the literature), van Werkhoven et al. (2009, AWR, https://doi.org/10.1016/j.advwatres.2009.03.002 ) made similar conclusions regarding the high performance of a priori parameters for humid US regions when using the SAC-SMA model. Considerations of climate regimes for global water models calibration was studied in Yoshida et al. (2022, WRR, https://doi.org/10.1029/2021WR030660 ) and Kupzig et al. (2023, ERL, DOI 10.1088/1748-

9326/acdae8) . So, there seems to be a context of past studies in which this new result could be placed. The wide range of co-authors could easily add some relevant reference for discussion.

AC: Regarding the sentence cited by the referee, we did not think that it would add sufficient value to add a reference to back this rather logical and straightforward result that we do not consider to be very central for our study. Following the advice of the reviewer, we now added a reference to the study of Troy et al. (2008) as they found, in the calibration of the VIC model to streamflow only, the same behavior for the Ohio River (their Figure 3). The revised sentence in Section 5.2 now reads:

**However, the added value of any calibration is very low in the humid and hilly Ohio basin where the performance of the uncalibrated model is already good; in their study on calibrating the VIC model for the USA using observed Q only, Troy et al. (2008) also found that modeled streamflow that fit well to observations before calibration, as was the case for the Ohio River, continued to do so.**

We do not want to discuss this result in the context of climate (i.e. that hydrological models often work best in humid climates), because the point is just that rather obviously where an uncalibrated model is already doing a quite good job, calibration cannot help much. In fact, hydrological models also perform poorly in humid climates if the impact of surface water bodies on streamflow and TWSA is high. This is why we do not think that it would be useful to cite Yoshida et al. (2022). The papers of Van Werkhoven et al. and Kupzig et al. deal with sensitivity analyses to select the most relevant parameters from the whole model parameter set, while we chose not to discuss our selection of the most sensitive parameters (which was the same for all three compared approaches) to not further increase the length of the already very long manuscript. In addition, both papers use only streamflow observations and not TWSA for their sensitivity analysis, while Section 5.2 discusses the value of multi-variable calibration. In the revised version, the paper of Kupzig et al. (2023) is now cited in Section 5.6.

In Section 5.2, we also put our overestimation of summer low flows in the context of the Troy et al. study by adding:

**In the study of Troy et al. (2008), the overestimation of summer low flows in the Arkansas basin, the basin that is affected most by this behavior in our study (Fig. 4), is reduced but not removed by the calibration.**

In addition, we inserted text on the added value of multi-variable calibration at the end of Section 5.2:

**The added value of multi-variable Pareto-optimal calibration of WaterGAP for 28 very large globally distributed basins using monthly time series of Q and TWSA was investigated by Werth and Güntner (2010). They found that improved simulations of TWSA and Q were achieved for most basins after calibration, but calibrated Q was still poor compared to the observed values in some basins; a better fit to GRACE TWSA did not necessarily lead to a better fit of simulated to observed Q. For the Mississippi basin, the relative RMSE was reduced by calibration by about 20% for both Q and TWSA. A multi-variable model calibration for the Lake Urumia basin (Iran) showed that satellite observations of time series of TWSA and irrigated area led to a good fit to observed TWSA and a reduction in the Q bias, but additional in-situ observations of Q were necessary to estimate parameter sets that lead to a good fit (Hosseini-Moghari et al., 2020). Both studies underline that model calibration should be based on both Q and TWSA observations.**

(5) RC: The discussion of possible reasons for output uncertainty does also not contain any

meaningful link to existing literature. Only one paper on groundwater losses is mentioned, but nothing else. For example, other large-scale models such as VIC etc have been applied to the same domain (given runs for the US) (e.g. Troy et al. 2008 WRR, https://doi.org/10.1029/2007WR006513 ). Did VIC show comparable results (NSE values)? Given that this models also does (did?) not include transmission losses. I think the authors should make a bit more effort to provide context for their results. Zajac et al. (2017, JoH, https://doi.org/10.1016/j.jhydrol.2017.03.022 ) for example explored the influence of reservoirs/lakes on global streamflow simulations, incl. uncertainty – How does this link to the findings here in which the authors seem to explore/experience similar issues in their GLUE simulations?

AC: The study of Troy et al. does not provide comparable NSE values. However, we related our results qualitatively to their result in Section 5.2 (see our answer to RC comment 4). We do not see a link to the study of Zajac et al. (2017). This study just compared the difference in Q fit to observations by a model that does not take into account lakes and reservoirs at all and a model version in which major lakes and reservoirs are included. This is not comparable to our study with a model that, different to the model used in Zajac et al. (2017), its standard version already includes lakes and reservoirs (and wetlands) and for which parameters are to be optimized. To clarify the difficulty of simulating water flows and storages in the Prairie Pothole Region, we added a sentence (and reference) to the second but last paragraph in Section 5.3:
**The Prairie Pothole Region contains between 5-60 wetlands per km$^2$, and their hydrological modeling relies on accurately characterized depth-volume relationships derived from detailed topographic surveys (Minke et al., 2010).**

6) RC: The authors later discuss how their parameter equifinality with that of Harlin and Kung (1992). While I appreciate that a comparison is made, does the very different set-up (floods versus monthly flows, very different regions, …) make this a useful comparison? The huge number of existing GLUE might provide an example which is much more like the study performed here.

AC: While there is a large number of GLUE studies, most use only streamflow observations. Both Harlin and Kung (1992) and the study of Jost et al. (2012), which is now also referred to in Section 5.6, are examples of rare multivariable GLUE studies that also provide some information on the estimated parameter sets. In the revised version, we added the following sentence at the end of Section 5.6:
**A multi-variable parameter estimation of a hydrological model for the upper Columbia River basin in Canada, which used observations of Q and glacier volume change, identified 23 rather different behavioral parameter sets that all led to very high NSE values for daily streamflow of at least 0.92 (Jost et al., 2012).**

(7) RC: Because I do not want to discuss the same issue again and again, let me make a wider statement here. The discussion section significantly lacks references to other literature. You need to convince that the results and findings are not so specific to WaterGap that other users do not benefit. One way to do so is by showing how your results add knowledge to previous studies. The authors hardly do this, and several discussion sections contain no references at all. Let me stress that my point here is not for the authors to simply include the few references I mention, but to rather make some effort to scan the literature so that their results are placed in context. Also, there are some references mentioned in the introduction (e.g.

Scanlon et al. 2018/2019) but they do not come back in the discussion section, which I think should be the case.

AC: To our knowledge, this is the first time that a GHM has been calibrated using both streamflow and total water storage anomaly except for Werth and Güntner (2010) and Hosseini et al. (2020); and the focus of the discussion section is show/discuss what we have learned about to an appropriate methodology for doing multi-variable parameter and uncertainty estimation (using TWSA and Q) in global hydrological modeling, clarifying aspects that are of value for other large-scale to global hydrological modeling. This was the reason for not focusing our discussion on a comparison to studies presented in the literature that did not do multi-variable parameter estimation.

In the revised discussion section, we followed your advice and came back to some references from the introduction in the discussion and also tried to place our results more strongly in the context of (additional) publications. We added the following to the discussion Sections 5.1 to 5.6:

- 5.1 Advantages and disadvantages of the three ensemble-based multi-variable calibration approaches: We refer to two additional references regarding the ensemble size of EnKF data assimilation
- 5.2 Added value of multi-variable calibration as compared to the standard WaterGAP calibration for identifying one optimal parameter set: See our answer to (4) RC. And we included two sentences relating our study to Scanlon et al. (2018, 2019).
  **Thus, where GHMs incorrectly simulate TWSA trends (Scanlon et al., 2018), multi-variable model calibration is likely to lead to more realistic simulated trends. However, at least for our CDA units, variability and probably also seasonality of simulated TWSA are not necessarily improved by such a calibration (Scanlon et al., 2019).**
- 5.3 Estimation of output uncertainty: See our answer to (5) RC.
- 5.4 Trade-offs between optimal simulation of Q and TWSA: Nothing was added, as the study results had already been related to two other studies (Rakovec et al., 2016 and Schumacher et al., 2018).
- 5.5 Added value of sub-basin CDAs instead of one basin CDA: Nothing was added, as the study results had already been related to five other studies.
- 5.6 Characteristics of identified (Pareto-)optimal and behavioral parameter sets: See our answer to (6) RC. In addition, a sentence citing Kupzig et al. (2023) was added.
  **SL-RC and SL-MSM, which affect the release of water from the soil and determine the maximum amount of water that can be stored in the soil, respectively, were found to be the most influential parameters for a number of Q metrics of the evaluated 347 global river basins (Kupzig et al. 2023).**
-

**References**

Zaitchik, B. F., et al. (2008). Assimilation of GRACE terrestrial water storage data into a land surface model: Results for the Mississippi River basin. Journal of Hydrometeorology, 9(3), 535-548.

Girotto, M., et al. (2016). Assimilation of gridded terrestrial water storage observations from GRACE into a land surface model. Water Resources Research, 52(5), 4164-4183.

Kumar, S. V., et al. (2016). Assimilation of gridded GRACE terrestrial water storage estimates in the North American Land Data Assimilation System. Journal of Hydrometeorology, 17(7), 1951-1972.

Getirana, A., et al. (2020a). Satellite gravimetry improves seasonal streamflow forecast initialization in Africa. Water Resources Research, 56(2), e2019WR026259.

Getirana, A., et al. (2020b). GRACE improves seasonal groundwater forecast initialization over the United States. Journal of hydrometeorology, 21(1), 59-71.

**Referee 3**

RC: The authors have made a number of revisions based on the comments from the reviewers and the editor. While I agree most aspects pointed out have been properly addressed I think a few minor revisions are needed.
AC: Thank you very much for your careful reading and detailed suggestions.

Abstract:
RC: L23-24: "…ensemble-based multi-variable calibration approaches…" -> ensemble-based approaches (for brevity and consistency with the title)
AC: It is important to keep the "multi-variable" here, as it is the essential feature. In the title, this is covered by "multi-variable observations".

RC: L29, L353, L1157, and L1283: observation data -> observational data
AC: done

RC: L31-32: Delete ", which utilizes the Borg multi-objective evolutionary search algorithm to find Pareto-optimal parameter sets, "
AC: done

RC: L39-40: Delete ", in which both parameter sets and water storages are updated,"
AC: done

RC: L43 and the rest of the manuscript: validation -> evaluation.
AC: While we agree that the term validation is problematic and might be replaced by e.g. "testing" or "evaluation", we prefer the term "validation period" to the term "evaluation period", because it is widely in the hydrological literature to refer to the period outside the calibration period for which the model is run with the calibrated parameters. We also evaluate model performance during the calibration period, so the term evaluation period is also not perfect.

RC:L50: GHM abbreviation not defined in the abstract
AC: We changed the sentence and do not use the abbreviation anymore.

Introduction:
RC: L64: "… uncertainty due to uncertain …" -> uncertainty due to
AC: done

RC: L90-91: Delete "(some missing months before 2016 and a gap until the start of GRACE-Follow-on mission in May 2018)"
AC: done

RC: L144: "How and how well…" -> How and to which extent
AC: done

Section 2:
RC: L206-217: This last paragraph reads more as discussion than a description of the

approach, which is the intention as stated in L182. Thus, I suggest this paragraph to be removed from section 2.1.
AC: done

RC: L251-259: Same as above. I suggest to continue from L250 directly into "To achieve plausible and stable EnCDA results…"
AC: done

Section 3:
RC: L334: The sentence from L90-91 could come here to elaborate.
AC: not included for brevity.

RC: L451: "…, for various reasons." -> for two main reasons.
AC: done

RC: L522, L523, L527, L609: Is it Borg-MOEA or Borg MOEA? Pick one and use it consistently.
AC: Borg MOEA

RC: L549-550: in which environment did the GLUE runs take place? It is not clear/explicit from the text if the same environment described in section 3.4.1 was used or not.
AC: It is made explicit in the sentence before, which reads "All GLUE runs started in 1991 and were done on the same Linux cluster machine as the POC runs."

RC: L565-566: plus/minus -> ±
AC: done

RC: L576: "… grids over …" -> grid cells over
AC: done

Section 4:
RC: Figure 4: I suggest increasing the legend font size. Also, in the caption, Table 36?
AC: Done, also for Figure S4

RC:
L741: Sect. -> Section
L770: correlates -> correlate
L791: Table 32?
L856: 5 m to 8 m -> 5 to 8 m
L893: "6 out of the 9 …" -> "Six out of the nine ..."
L965: quotation marks are opened but never closed. ("whole …)
AC: all done

Section 5
RC:
L1026: multi-variable calibration -> ensemble-based
L1061: RMS? Did you mean RMSE?
L1097: 9 -> nine
L1166: .. -> .
AC: done

Section 6

RC: L1280-1285: Is it possible to draw such conclusions if the EnCDA was limited to 32 (vs 20,000 for POC and GLUE) ensemble members?. As pointed out by the authors in L1051-1052, results might be an artifact of the small ensemble size.

AC: We had already written in the following sentence "Potential reasons are the severe computational burden of the EnCDA approach that only allowed setting up a very small ensemble and the intrinsic nonlinearity in simulating Q." For clarity, we revised the sentence to the following:

**Possibly due to the severe computational burden of the EnCDA approach that only allowed setting up a very small ensemble, the multi-variable EnCDA approach that we followed in our pilot study is not suitable for application for GHM parameter estimation using Q and TWSA, as 1) its performance is lower during the calibration period than that of POC and GLUE, or for the large CDA unit MRB even lower than that of the uncalibrated WaterGAP and 2) its application during the validation period (without observational data) led to spurious results (Figs. 4 and S4). The intrinsic nonlinearity in simulating Q makes a multi-variable EnCDA that includes Q observations more difficult than an EnCDA that only includes TWSA or TWSA and other storage observations.**

---

## Author Response (AR3)

hess-2023-18

Dear editor, dear editorial team,

thank you very much for the thorough review process that definitely helped to improve the manuscript significantly.

Regarding the suitability for color-blind readers, we have adapted and tested Figures 2 and S2. Unfortunately, we could not yet adapt Figures S5 and S6 in the supplement as the co-authors who made the figures has been on parental leave in his home country Bangladesh where he seems to be disconnected from the Internet. He is expected to return to the working place in Germany next week, I hope that then we can submit the supplement with the two adapted figures.

Best regards,

Petra Döll